# Micro𝕊plit: semantic unmixing of fluorescent microscopy data

Ashesh Ashesh [1], Federico Carrara [1,2], Igor Zubarev[1], Vera Galinova[1], Melisande Croft[1], Melissa Pezzotti[3], Daozheng Gong [4], Francesca Casagrande [1], Elisa Colombo[1], Stefania Giussani[1], Elena Restelli[1], Eugenia Cammarota[1], Juan Manuel Battagliotti[1], Nikolai Klena [1], Moises Di Sante [3], Raghabendra Adhikari [5], Daniel Feliciano [5], Gaia Pigino [1], Elena Taverna[1], Oliver Harschnitz[1], Nicola Maghelli [1], Norbert Scherer [4], Damian Edward Dalle Nogare[1], Joran Deschamps [1], Francesco Pasqualini [3] & Florian Jug [1]✉

Fluorescence microscopy is constrained by optical limits, fluorophore chemistry and finite photon budgets, imposing trade-offs between imaging speed, resolution and phototoxicity. Here we introduce Micro𝕊plit, a deep learning-based computational multiplexing method that enables multiple cellular structures to be imaged simultaneously in a single fluorescent channel and then computationally unmixed. We show that Micro𝕊plit separates up to four superimposed noisy structures into distinct, denoised image channels, enabling faster and more photon-efficient imaging. Built on Variational Splitting Encoder-Decoder networks, Micro𝕊plit models a posterior distribution over solutions, allowing uncertainty-aware predictions and the estimation of spatially resolved prediction errors from posterior variability. We demonstrate robust performance across diverse datasets, noise levels and imaging conditions, and show that Micro𝕊plit improves downstream analysis while reducing photon exposure. All methods, data and trained models are released as open resources, enabling immediate adoption of computational multiplexing in biological imaging.

Fluorescence microscopy is an essential tool for exploring structures and dynamics within cells, tissues, and organisms. Multiplexed acquisition techniques involving multiple fluorophores, each tagged to a different cellular structure, are used to acquire multichannel image data; however, overlap in fluorophore excitation spectra imposes practical limits on the number of fluorophores that can be used in a biological sample (Fig. 1a) without problems such as crosstalk or bleedthrough[1]. Even if a suitable set of fluorescent labels have been chosen, the sequential nature of multiplexed acquisitions requires multiple exposures of the sample, which reduces the photon budget available for other purposes[2,3] and also limits the maximum sampling frequency in live-cell imaging scenarios. A limited photon budget can

be addressed by reducing the photon exposure per acquisition, which in turn will lead to more noisy images that will be harder to analyze[3–5].

In this work, we describe a computational multiplexing approach, Micro𝕊plit, which addresses the limitations outlined above. We propose labeling and imaging multiple biological structures in a single fluorescent channel and then employing our proposed method to split superimposed structures from within this multi-structure channel computationally into separate unmixed image channels (Fig. 1b). We recently presented the underlying methodological components that can enable such computational multiplexing in theory[6,7] and have now created a practical method that (1) combines the benefits of both approaches into a single framework, (2) enables the processing of

[1]Fondazione Human Technopole, Milan, Italy. [2]Universitá Campus Bio-Medico di Roma, Rome, Italy. [3]University of Pavia, Pavia, Italy. [4]University of Chicago, Chicago, IL, USA. [5]HHMI/Janelia Research Campus, Ashburn, VA, USA. ✉e-mail: florian.jug@fht.org

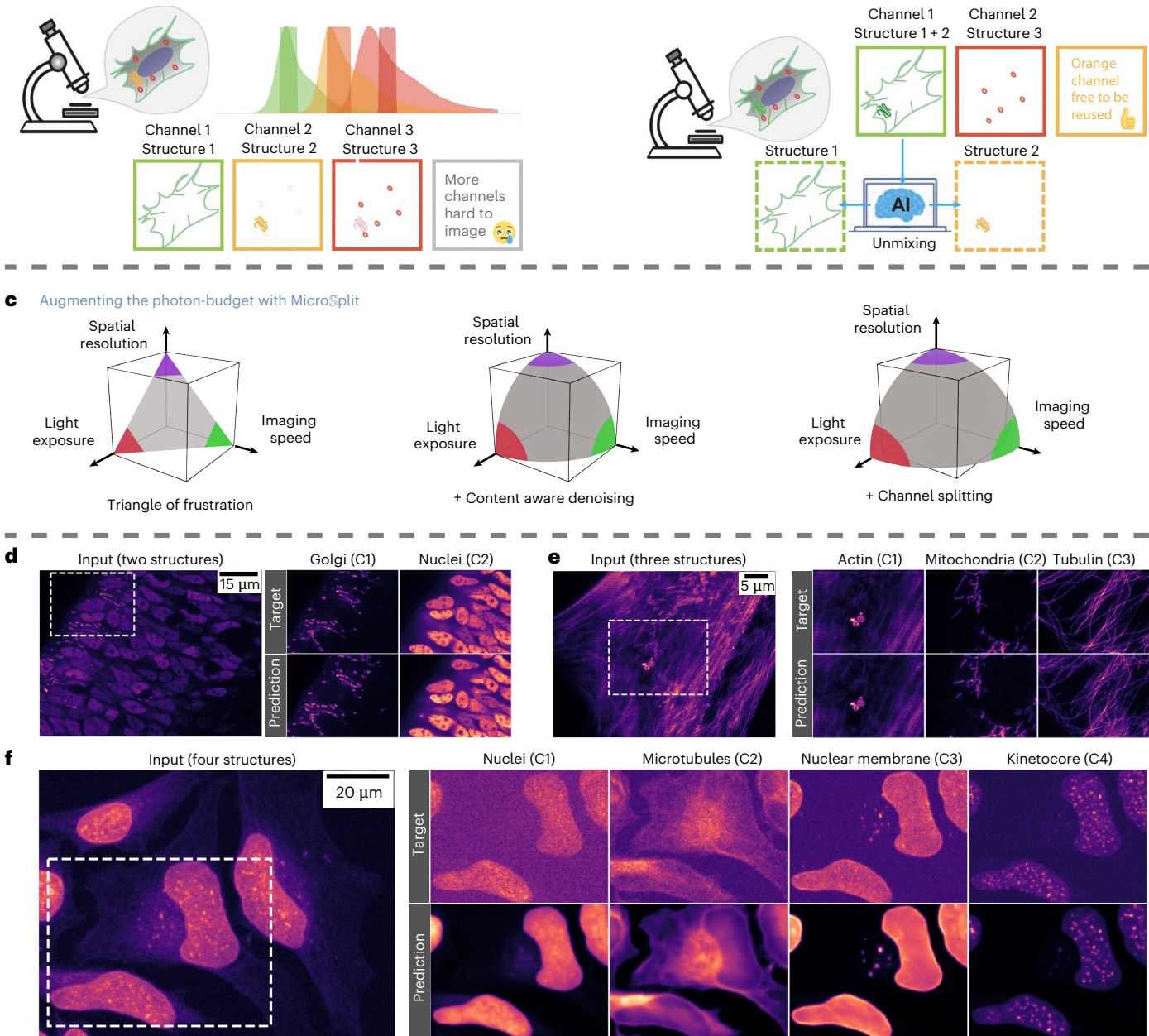

**Fig. 1 | Semantic unmixing of fluorescence microscopy data with** Micro𝕊plit.
**a**, Multiplexed fluorescent imaging is conducted by labeling cellular components with different fluorescent markers and imaging one after another into separate image channels. **b**, We propose a method that allows multiple cellular structures to be imaged simultaneously in a single image channel. The superimposed structures are then split into separate channels using an adequately trained neural network. **c**, Imaging multiple structures in one go saves precious photon budget. It is well known that the total photon budget limits the capabilities of light microscopes (left). Using content-aware denoising methods, for example[3,4], can help repurpose some photon budget by acquiring more noisy raw images

(middle). Imaging multiple cellular components in a single fluorescent channel saves additional photon budget that can then, for example, be invested into imaging additional structures, imaging more gently or imaging faster (right). **d–f**, Exemplary qualitative results of two-channel splitting (**d**), three-channel splitting (**e**) and four-channel splitting (**f**). Each panel shows the image channel containing multiple structures to be split (input), insets of the noisy target channels ($Ci$, $i \in \{1, 2, 3, 4\}$) as they are used during training (target) and insets of the split channels as predictions by Micro𝕊plit (prediction). See Extended Data Figs. 9 and 10 and supplementary material for more qualitative results.

volumetric image data by creating a highly optimized network architecture for it (Fig. 1 and Table 1), and (3) we propose a procedure to assess the calibration of a trained network and the estimation of prediction errors (Fig. 2 and Section on 'Error estimation, data uncertainty and calibration'). Micro𝕊plit is designed for easy use by microscopists and life scientists, even those who are not machine-learning experts.

Using Micro𝕊plit allows users to (1) simultaneously image more structures than previously possible by combining up to four structures

in a single channel, or to (2) image the same number of structures but in fewer channels and, hence, at reduced photon exposure (Fig. 1c). The saved photon budget can then be allocated toward other objectives such as faster temporal sampling, better signal-to-noise ratio (SNR) in raw acquisitions, for example to image more gently and avoid phototoxicity, or any combination of these possibilities. Note that each semantic unmixing task requires training a dedicated Micro𝕊plit model. While training a universal foundation model is conceptually possible, we deliberately

**Table 1 | Quantitative performance on two-, three- and four-channel splitting tasks**

| Task | Dataset | Task detail | 2D/3D | Number of channels | PSNR | MicroMS-SSIM |
|------|---------|-------------|-------|--------------------|------|--------------|
| I | HT-H24 | TM-I | 3D | 2 | 38.8, 33.8 | 0.970, 0.951 |
| II | HT-P23A | TM-I | 3D | 2 | 25.1, 31.4 | 0.747, 0.932 |
| III | HT-P23B | TM-I | 3D | 2 | 26.4, 21.6 | 0.847, 0.599 |
| IV | Pavia-P24 | TM-III, high, 50:50 | 2D | 2 | 24.3, 29.9 | 0.682, 0.839 |
| V | Pavia-P24 | TM-III, high, 66:33 | 2D | 2 | 28.2, 25.6 | 0.780, 0.696 |
| VI | Pavia-P24 | TM-III, high, 84:16 | 2D | 2 | 25.2, 24.1 | 0.722, 0.623 |
| VII | Pavia-P24 | TM-III, mid, 50:50 | 2D | 2 | 23.1, 24.3 | 0.673, 0.755 |
| VIII | Pavia-P24 | TM-III, mid, 66:33 | 2D | 2 | 24.0, 22.3 | 0.729, 0.678 |
| IX | Pavia-P24 | TM-III, mid, 84:16 | 2D | 2 | 24.3, 22.4 | 0.735, 0.659 |
| X | Pavia-P24 | TM-III, low, 50:50 | 2D | 2 | 21.9, 23.0 | 0.612, 0.684 |
| XI | Pavia-P24 | TM-III, low, 66:33 | 2D | 2 | 22.9, 21.7 | 0.593, 0.564 |
| XII | Pavia-P24 | TM-III, low, 84:16 | 2D | 2 | 23.3, 22.8 | 0.682, 0.663 |
| XIII | HT-T24 | TM-III | 2D | 2 | 40.3, 32.8 | 0.978, 0.951 |
| XIV | HT-LIF24 | TM-III | 2D | 2 | 32.0, 32.9 | 0.965, 0.960 |
| XV | Chicago-Sch23 | TM-I, C0 versus C1 | 2D | 2 | 41.3, 42.9 | 0.984, 0.993 |
| XVI | Chicago-Sch23 | TM-I, C0 versus C2 | 2D | 2 | 38.4, 41.1 | 0.973, 0.987 |
| XVII | Chicago-Sch23 | TM-I, C0 versus C3 | 2D | 2 | 58.2, 41.7 | 0.998, 0.993 |
| XVIII | Chicago-Sch23 | TM-I, C1 versus C2 | 2D | 2 | 43.7, 44.9 | 0.995, 0.995 |
| XIX | Chicago-Sch23 | TM-I, C1 versus C3 | 2D | 2 | 65.4, 44.3 | 1.000, 0.996 |
| XX | Chicago-Sch23 | TM-I, C2 versus C3 | 2D | 2 | 66.6, 43.6 | 1.000, 0.996 |
| XXI | CBG-Z18 | TM-I | 3D | 3 | 28.1, 28.9, 29.1 | 0.920, 0.929, 0.913 |
| XXII | CBG-N18 | TM-I | 3D | 3 | 38.4, 42.4, 35.0 | 0.977, 0.981, 0.974 |
| XXIII | HHMI-D25$_{8bit}$ | TM-I | 3D | 3 | 22.5, 31.3, 24.3 | 0.840, 0.768, 0.793 |
| XXIV | HT-LIF24 | TM-III, 2 ms | 2D | 3 | 31.0, 32.2, 36.3 | 0.940, 0.973, 0.944 |
| XXV | HT-LIF24 | TM-III, 3 ms | 2D | 3 | 30.8, 32.2, 36.1 | 0.940, 0.973, 0.948 |
| XXVI | HT-LIF24 | TM-III, 5 ms | 2D | 3 | 32.9, 34.3, 37.6 | 0.960, 0.983, 0.963 |
| XXVII | HT-LIF24 | TM-III, 20 ms | 2D | 3 | 37.0, 39.7, 41.4 | 0.984, 0.994, 0.989 |
| XXVIII | HT-LIF24 | TM-III, 0.5 s | 2D | 3 | 39.5, 41.6, 43.1 | 0.991, 0.995, 0.995 |
| XXIX | HT-LIF24 | – | 2D | 4 | 32.1, 32.5, 34.2, 37.6 | 0.964, 0.959, 0.983, 0.957 |
| XXX | Chicago-Sch23 | – | 2D | 4 | 35.6, 39.4, 38.8, 36.2 | 0.956, 0.986, 0.980, 0.942 |

Throughout this work, we refer to specific splitting tasks by their task ID (first column). Some of the microscopy datasets we use (column two; see also Table 2), give rise to multiple splitting tasks, for example by selecting a subset of the fluorescent channels or using different noise levels or channel mixing weights (see Section on 'Used microscopy datasets'). Such task details are in abbreviated form given in the third column along with the training mode used for the task. For tasks with task ID XXIX and XXX, the training modes used were TM-III and TM-I, respectively. For volumetric tasks (labeled 3D in column 2D/3D), we fed a 3D image stack to MicroSplit, which in turn also predicts 3D outputs (posterior samples). We evaluated all tasks using held-out test sets, reporting MicroMS-SSIM[8] and CARE-PSNR (PSNR) metrics for each predicted unmixed channel. Accordingly, the PSNR and MicroMS-SSIM columns contain as many entries as there are channels (fifth column). In Supplementary Table 12 we list all standard errors for the results in the above table. Note that Task XXIII is the most difficult semantic unmixing task of visually different cellular structures we encountered so far, particularly with regard to the quality of predictions for the third channel. In the Section 'Limitations of MicroSplit' and Supplementary Note 1.1.1, we elaborate on how low SNR and several other properties of the raw data are contributing to the simplicity or difficulty of semantic unmixing and how potentially occurring problems can be mitigated.

employ narrow, task-specific models to avoid out-of-distribution issues and to ensure robust performance on the data at hand.

Our experiments show that MicroSplit is capable of separating two, three or even four jointly imaged structures into denoised unmixed channel predictions, even if the training data are itself noisy (Fig. 1d–f). A MicroSplit model learns to unmix superimposed structures by learning from examples (that is in a supervised way). At the same time, it co-learns to denoise the data without supervision (Fig. 2a). This means that it is sufficient to use a body of noisy training data and MicroSplit will still learn to predict denoised images for each unmixed structure (images labeled 'Target' in Figs. 1 and 2 show such noisy training data. Note that in many supervised training settings, the supervision data are referred to as 'ground truth'. In this work we instead refer to the supervision signals as 'Target' since MicroSplit can be trained exclusively on noisy data that is therefore not ground truth). These

properties make our method straightforward to apply in practical applications, as we show in detail in the Results.

An additional and distinctive feature of the variational splitting encoder-decoder (VSE) networks we are using is their ability to generate multiple plausible solutions for a given input image[7]. When given inputs are unambiguous and thus can be split with high certainty, the generated solutions will closely resemble each other. In contrast, as the uncertainty in inputs increases, the diversity among the sampled solutions will reflect these inherent ambiguities by becoming more different from one another. We show that the networks we trained are calibrated, allowing us to estimate the otherwise hard to assess true error of predictions, even in the absence of ground-truth images that are notoriously hard or even impossible to come by. Technically, this is enabled by evaluating the easy-to-compute inter-sample variability (see Section on 'Error estimation, data uncertainty and calibration'). This feature is

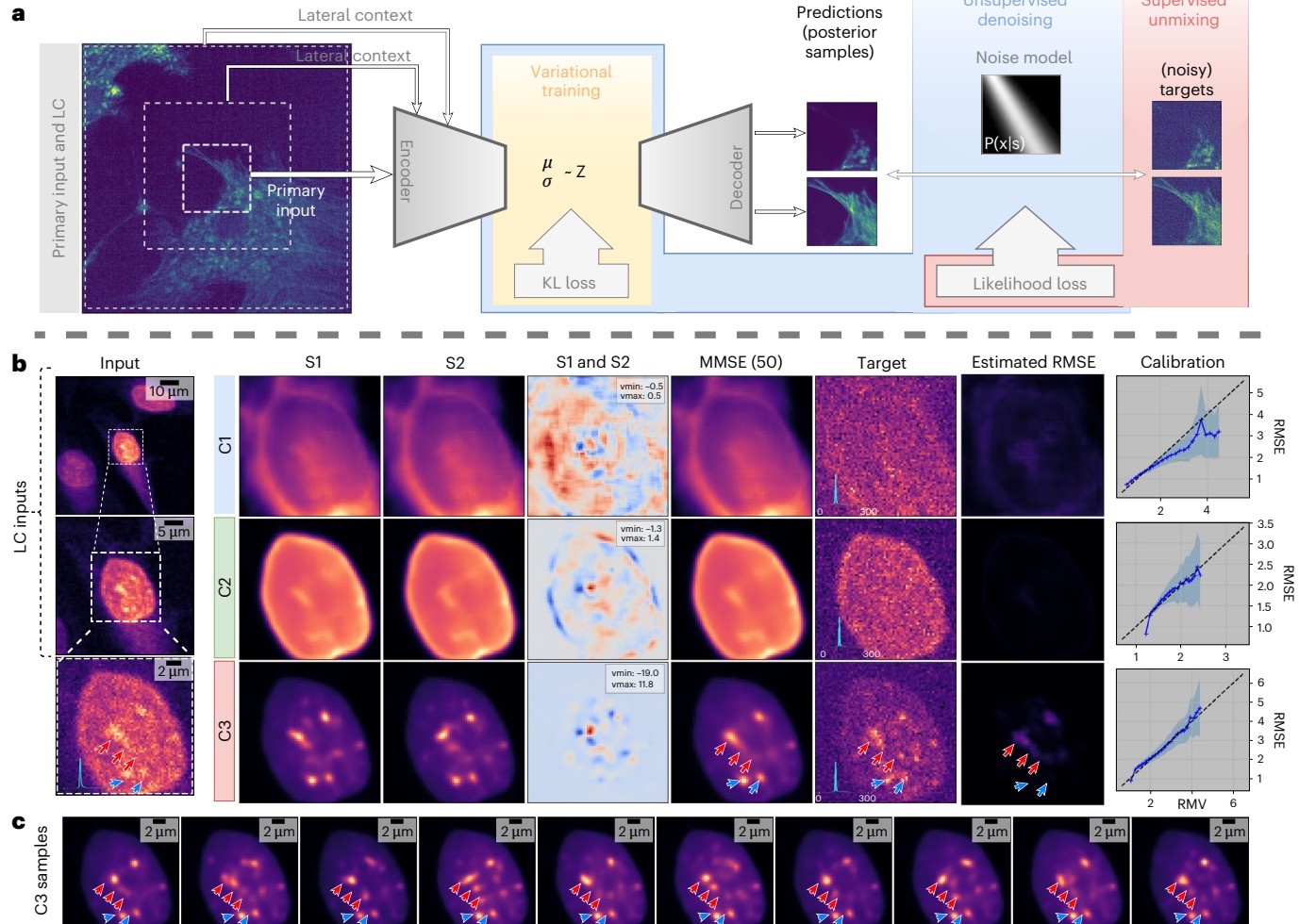

**Fig. 2 | Network architecture, posterior sampling and calibration.**
**a**, Micro𝕊plit jointly learns to denoise input images (unsupervised, blue shaded area) and split superimposed structures (supervised, red shaded area). For this, we use a hierarchical network architecture, variational training (yellow shaded area) and use LC[6] for better performance. Due to unsupervised denoising, using an adequate noise model[7,22], the supervised target images can be subject to noise, and the final prediction will still be noise-free. **b**, Left-most column shows an input patch containing three labeled structures in a single-channel at native resolution (bottom) and two LC input patches that add additional spatial context surrounding the primary input patch (toward the top). As the input is noisy, not all structural details in the data are visible. To account for this noise-induced data uncertainty, the variational architecture we use is capable of sampling plausible 'interpretations' from a learned posterior of possible solutions. For each of the three superimposed image structures (C1–C3) we show two such posterior

samples in columns two and three, and their difference as a heat map in column four. The fifth column shows an approximation of the posterior MMSE, computed by pixel-wise averaging 50 posterior samples. The last three columns show the noisy target images (as used during training), an estimate of the pixel-wise RMSE and the calibration plots for the three channels, respectively. The calibration plots show that the uncertainty estimated solely from predictions (RMV) correlate with the true prediction error (RMSE), which means that this easy-to-calculate quantity (RMV) can be used as a surrogate for an otherwise difficult-to-assess prediction error of Micro𝕊plit. S1, sample 1; S2, sample 2. **c**, Ten posterior samples for C3, the channel showing the highest estimated RMSE values in **b**. Note that puncta at low estimated RMSE value locations (cyan arrows) remain relatively unchanged and closely resemble the structure present in the noisy target at that location, while puncta at high estimated RMSE locations (red arrows) show considerable variations between samples in **c**.

crucial because it enables users to identify the parts of their data where Micro𝕊plit predictions may be unreliable due to ambiguities in the fed inputs. Such regions can either be disregarded or subjected to expert review, effectively addressing a persistent challenge in AI-driven bioimage analysis, the difficulty in evaluating the accuracy of predictions[1].

We show the performance of Micro𝕊plit on 24 two-dimensional (2D) and six three-dimensional (3D) semantic unmixing tasks (with six additional tasks in Supplementary Table 2), showing that we achieve accurate results in a wide range of noise levels and superimposed structures, consistently leading to high-quality denoised single-structure predictions.

In addition to the above-mentioned use cases, we propose a way to use Micro𝕊plit to remove unwanted structured image artifacts by unmixing them from the structures in which we are truly interested.

We showcase this capability by removing spurious fluorescent puncta from a fluorescent microscopy dataset (see Section on 'Removing unwanted imaging artifacts' and Table 2).

## Results

As introduced above, Micro𝕊plit enables microscopists to image multiple structures that would typically be imaged in separate fluorescence channels and instead (1) capture these structures in one channel, and then (2) unmix the superimposed structures contained in that channel by computational means. The training procedure we propose requires supervision signals (target images) for each structure to be unmixed. In the following section, we outline three practical ways to obtain suitable training data and will use them in various experimental settings to demonstrate their effectiveness.

**Table 2 | Dataset overview**

| Dataset | 2D/3D | Pixel size (nm) | Exposure (ms) | Pixel/voxel (million) |
|---|---|---|---|---|
| Pavia-P24 | 2D | 68 | 200 | 27.9 |
| HT-T24 | 2D | 110 | 400 | 2,831.3 |
| HT-LIF24 | 2D | 285 | 2, 3, 5, 20, 500 | 4,126.8 |
| Chicago-Sch23 | 2D | 270 | 100 | 7,516.2 |
| HT-H24 | 3D | 285 | 300 | 1,985.8 |
| HHMI-D25$_{8bit}$ | 3D | 45 | – | 3,850.4 |
| CBG-Z18 | 3D | 300 | – | 1,362.1 |
| CBG-N18 | 3D | 196 | – | 510.7 |
| HT-P23A | 3D | 37 | – | 637.5 |
| HT-P23B | 3D | 42 | – | 416.5 |

This table summarizes key quantitative characteristics of the datasets used in the presented work. The column 'Pixel/voxel' shows the total number of pixels/voxels in the entire dataset. Note that individual training tasks only use a (at times) small subset of the entire dataset they are derived from.

### Training modes and required training data

Traditionally, data are acquired using conventional multiplexing (Fig. 1a). Hence, we propose Training Mode I, where previously recorded multiplexed image channels are used as targets for supervised training to achieve semantic unmixing of cellular structures (Fig. 2a). We generate mixed input images by pixel-wise summation of multiplexed image channels. These input images closely resemble what can later be acquired in a single acquisition (Fig. 1b). Although this approach yields high-quality training data, some differences in noise properties and intensity variations between structures may arise, as discussed in the Section 'Training Mode I versus Training Mode II (how important are spatial correlations?)' Experiments using Training Mode I show consistently high semantic unmixing performance, as seen in Figs. 1 and 2, Table 1, and Supplementary Note 1.2.

In cases where multiplexed data for Training Mode I are not available, we propose Training Mode II. Here, we assume that images of each single structure exist, but we no longer assume that all structures have been imaged in each sample. As before, here we also generate superimposed inputs by summing images showing different fluorescent structures; however, unlike before, summed-up input images no longer originate from the same sample, and any spatial correlations that might exist between these cellular structures are lost. As a network trained with Training Mode II cannot leverage these correlations, we reasoned that Training Mode I should be at least as performant as Training Mode II. In the Section 'Training Mode I versus Training Mode II (how important are spatial correlations?)', we test this hypothesis and show that Training Mode II indeed comes with a slight performance drop in cases where the channels to be unmixed offer actionable spatial correlations that MicroSplit can leverage.

A variation of this approach, Training Mode IIb, was used to obtain the results shown in the Section on 'Removing unwanted imaging artifacts'. Instead of summing uncorrelated sets of images, we used image data of superimposed structures. If those structures mix in some areas, but are visible in isolation in others, we cropped regions of interest showing individual structures and summed them randomly into superimposed training inputs. This training mode is best understood in the context of the results we present in the Section on 'Removing unwanted imaging artifacts'.

Finally, in Training Mode III, we do not create input images by summing images of individual structures but rather acquire them also at the microscope. In this mode, as in Training Mode I, all structures of interest must be individually labeled to allow multiplexed imaging of the required training targets. Additionally, we acquire an additional image channel by exciting all used labels at once and collecting the entirety of the emitted light, hence, directly imaging also the superimposed input directly at the microscope. Naturally, Training Mode III then uses this channel instead of the summation of the target channels as input to MicroSplit. The advantage of this is that the input image is also subject to realistic image noise and that the relative intensity of the different structures is also realistic (see results for tasks with IDs IV–XIV and XXIII–XXIX).

These training modes provide flexibility in the preparation of training data for MicroSplit, ensuring robust performance under a variety of experimental conditions and noise levels. We provide an overview of training modes in Supplementary Table 1.

### MicroSplit yields high-quality unmixed structures

To explore the performance of MicroSplit in various biological samples, imaging modalities, and training modes (see Section on 'Training modes and required training data'), we collected a total of ten datasets and defined a total of 30 + 6 semantic unmixing tasks, as shown in Table 1 and Supplementary Table 2. Details about the datasets can be found in the Section on 'Used microscopy datasets'.

In Fig. 1d–f we show qualitative results for three of these tasks. A quantitative assessment to the available ground truth, called training targets throughout this paper, can be found in Table 1. In the same table, we list all tasks and the achieved quality of unmixed channels in terms of CARE-PSNR (PSNR)[1] and MicroMS-SSIM[8], a variation on structural similarity index measure (SSIM) optimized for quantitative evaluation of microscopy image data.

Throughout all tasks, we observed average PSNR and MicroMS-SSIM values of 32.53 and 0.886, respectively, which for more common tasks, such as image denoising, would typically be considered high enough to warrant downstream processing and analysis. The lowest score of all semantic unmixing experiments still shows PSNR/MicroMS-SSIM values of 21.6/0.564 (Task III-channel 2 and Task VI-channel 2, respectively), which, depending on the desired analysis to be performed, is arguably still reliable enough for downstream analysis.

However, neither PSNR nor MicroMS-SSIM are sufficient to know how trustworthy the predictions of MicroSplit are, since these metrics can only be calculated when ground truth targets are available. For this reason, MicroSplit employs a variational training paradigm of its splitting encoder-decoder neural network, as shown in Fig. 2. The network architecture we use is similar to a hierarchical variational autoencoder (HVAE)[9], sometimes also referred to as a Ladder-VAE[10]. The most prominent difference in our setup is that MicroSplit is not an autoencoder, since the predictions are not meant to be the same as the given input. Details about the precise network architectures and the training procedure of MicroSplit can be found in the Section on 'Used microscopy datasets', Extended Data Fig. 1 and in refs. 6,7. In the next section, we show how the variational nature of MicroSplit can be exploited to estimate prediction errors caused by ambiguous inputs being fed.

### Error estimation, data uncertainty and calibration

MicroSplit exploits the variational nature of its underlying architecture to enable uncertainty quantification. As variational networks are not simple point-predictors, but instead are capable of learning an approximate posterior of possible solutions[1], MicroSplit can efficiently sample such solutions. As for VAEs[11], more likely solutions will be sampled more frequently, suggesting that the analysis of the inter-sample variability can be a good surrogate for the uncertainty in the input data and, therefore, also for the trustworthiness of semantic unmixing results.

We tested this hypothesis and show our findings in Fig. 2b,c. More specifically, we show an input patch and the lateral context that was fed to MicroSplit, two posterior samples, the difference between them as a heat map, and the (approximate) minimum mean-squared error

(MMSE) prediction, obtained by pixel-wise averaging of 50 posterior samples. We also show the target patches used to train the shown three-channel splitting task (Task XXIII). Note also here how much lower the SNR is in the training data than in the predictions of a trained Micro$\mathbb{S}$plit network. Finally, we show the estimated pixel-wise root mean-squared error (RMSE), computed using the 50 posterior samples that we also used to generate the previously shown MMSE solution.

The overall idea might be best conveyed by looking at a concrete example. Cyan and red arrows in Fig. 2b,c point at image locations that might show puncta-like structures. Note that in the ten posterior samples shown in Fig. 2c, the locations pointed at with cyan arrows remain relatively unchanged and the predictions match the structure present in the target channel. However, those locations pointed at with red arrows change their appearance considerably from one posterior sample to the next. This is, in fact, also reflected in the estimated RMSE patches shown in Fig. 2b, where higher estimated RMSE values appear precisely at locations where the posterior samples show elevated variability and is hence indicating that the input data are ambiguous in these regions; however, whether variability in posterior samples actually manifests itself in locations where predictions of Micro$\mathbb{S}$plit are prone to error and, conversely, also only in these places, is not obvious. Posterior samples could consistently predict structures in places where they are not, or consistently predict the absence of structures where, in fact, such structures should be. To show that a trained Micro$\mathbb{S}$plit network does not consistently hallucinate the presence or absence of structures, we propose to check how well calibrated the network is[7,12–14].

A calibrated neural network is one whose predicted probabilities or uncertainties accurately reflect the true likelihood of outcomes. In the context of a regression task like the one executed by Micro$\mathbb{S}$plit, calibration means that the predicted error estimates of the model align with the actual distribution of errors in the data. We therefore estimated the true error by computing the pixel error between the MMSE prediction of Micro$\mathbb{S}$plit and the ground truth target images we derived from the available training data, and plotted this 'true error' against the RMSE errors we described above. As the calibration plots in Fig. 2b show, the true error and RMSE scale roughly linearly with respect to each other, meaning that the estimated RMSE, which can be computed from posterior samples only, is a good estimator of the true error. Hence, calibrated Micro$\mathbb{S}$plit networks offer a reliable and efficient way to estimate the uncertainty of the data and, therefore, also the magnitude of the error of its predictions. This property is a considerable advantage over nonvariational approaches and will help users to (1) filter images or image regions that lead to uncertain predictions, and (2) provide evidence to themselves and others that unmixed images do not suffer from hallucinations.

### Downstream processing of unmixed data

Sample preparation and microscopy data acquisition are typically only the first steps in the pursuit of scientific insight. Once images have been acquired, their content must be analyzed and quantified. In this work, we cannot cover the full breadth of image data analyses that life scientists routinely perform on microscopy data[15]; however, in many analysis pipelines, image segmentation is a key step since it determines the identity, location, shape, and relative location of biological entities of interest. In the following section, we have quantified the segmentation performance on Micro$\mathbb{S}$plit predictions compared to the same analysis carried out on traditionally multiplexed image data.

**Segmentation of unmixed image data is of high quality.** We show that a typical segmentation task, as it is conducted in biological research laboratories on a daily basis, leads to comparable quality results when conducted on regular multiplexed microscopy data or on unmixed images. To this end, we compared the segmentation results of three bioimage analysts who were instructed to interactively train a pixel-classifier until they reach best-possible results. We show the results in Fig. 3 and Supplementary Fig. 2, and explain the details of this experiment below.

First, each analyst segmented the single-channel microscopy data acquired by regular multiplexed fluorescent microscopy (target) and the two-channel semantic unmixing predictions of Micro$\mathbb{S}$plit (prediction) without being informed about the nature of the data. We then compared the consistency between the segmentations of each analyst on the two sets of data and the consistency of the segmentations between the analysts. We observed that the variability between segmentations on target and unmixed images lies within the range of the inter-observer variability between the three analysts. We qualitatively show the segmentation results on two experiments (tasks) in Fig. 3b,c and plot the measured intra-observer variabilities (A1, A2, A3) and the inter-observer variability.

Hence, in all experiments we conducted, the quality of segmentations remained unchanged when using unmixed image data compared to multiplexed images. But more notably, imaging two structures in a single channel can free up photon budget, which then becomes available for imaging at a higher frame rate, imaging at a higher SNR, or imaging more labeled structures in additional image channels.

**Removing unwanted imaging artifacts.** Naturally, Micro$\mathbb{S}$plit predictions are just images and any subsequent analysis can be performed with them. As a second downstream processing example, we present an innovative way to remove imaging artifacts. More specifically, we imaged the post-mitotic neuronal marker CTIP2 in sliced hPSC-derived forebrain organoids using a secondary antibody conjugated to a 555 dye. The resulting image data not only had CTIP2$^+$ nuclear staining (wanted signal), but also puncta that accumulated because of nonspecific signal (undesired artifacts).

Ideally, we could use Micro$\mathbb{S}$plit to unmix the labeled nuclei (desired image content) and the undesired puncta (that is imaging artifacts); however, for this we would need training targets that contain only nuclei and others that contain only puncta, a requirement that the experimental setup of our colleagues does not permit. In the Section on 'Training modes and required training data', we introduce Training Mode IIb, the manual selection of image regions that show only the artifacts (puncta) or only the desired structures (the signal, in this example, nuclei). The selected image regions are then combined, just as in any Training Mode II experiment, and used to train a Micro$\mathbb{S}$plit network. As crops without puncta or crops that show only puncta are of limited size with respect to full microscopy images (the smallest

**Fig. 3 | Segmentations on unmixed channels are in line with inter-observer variability.** We define three segmentation tasks, each on one unmixed channel predicted by Micro$\mathbb{S}$plit, and compare the segmentations created by three bioimage analysts with each other. **a**, For the three tasks at hand, we show overview images (left) and insets (right) of three two-channel splitting tasks, with the superimposed raw input image in the top row, the predicted channel to be segmented in the middle row and single-channel control acquisitions via regular multiplexing (target, Fig. 1a) in the bottom row. **b,c**, For task 1 (**b**) and task 2 (**c**), we show segmentation results obtained by three analysts (A1–A3) obtained on the Micro$\mathbb{S}$plit predictions (top row), and the single-channel control acquisitions (target, middle row). The rightmost column shows overlays of the results obtained by all three analysts, with consistently segmented pixels being shown in white. The bottom row first shows the overlay of the segmentation results of a single analyst on predicted (top) versus single-channel control inputs (middle), and in the last column a box plot of all pairwise DICE scores between predicted versus target channel inputs of the entire test set (5 images of size 1,600 × 1,600 for task 1 and size 4,096 × 4,096 for task 2). In the box plot, the last column is variability, which captures inter-observer variability between the segmentation of 5 target frames by the 3 analysts (15 data points). Pred, prediction; Tar, target.

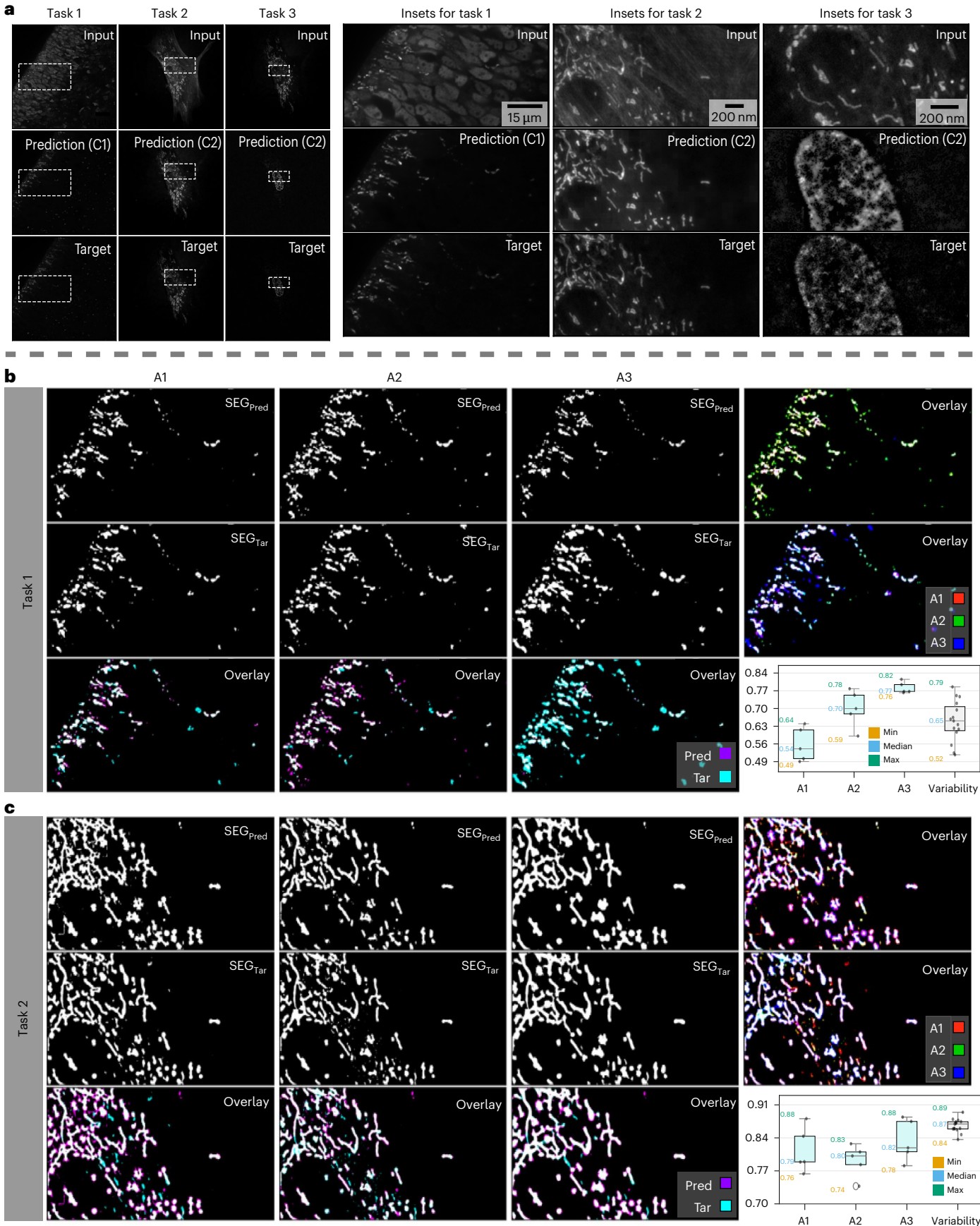

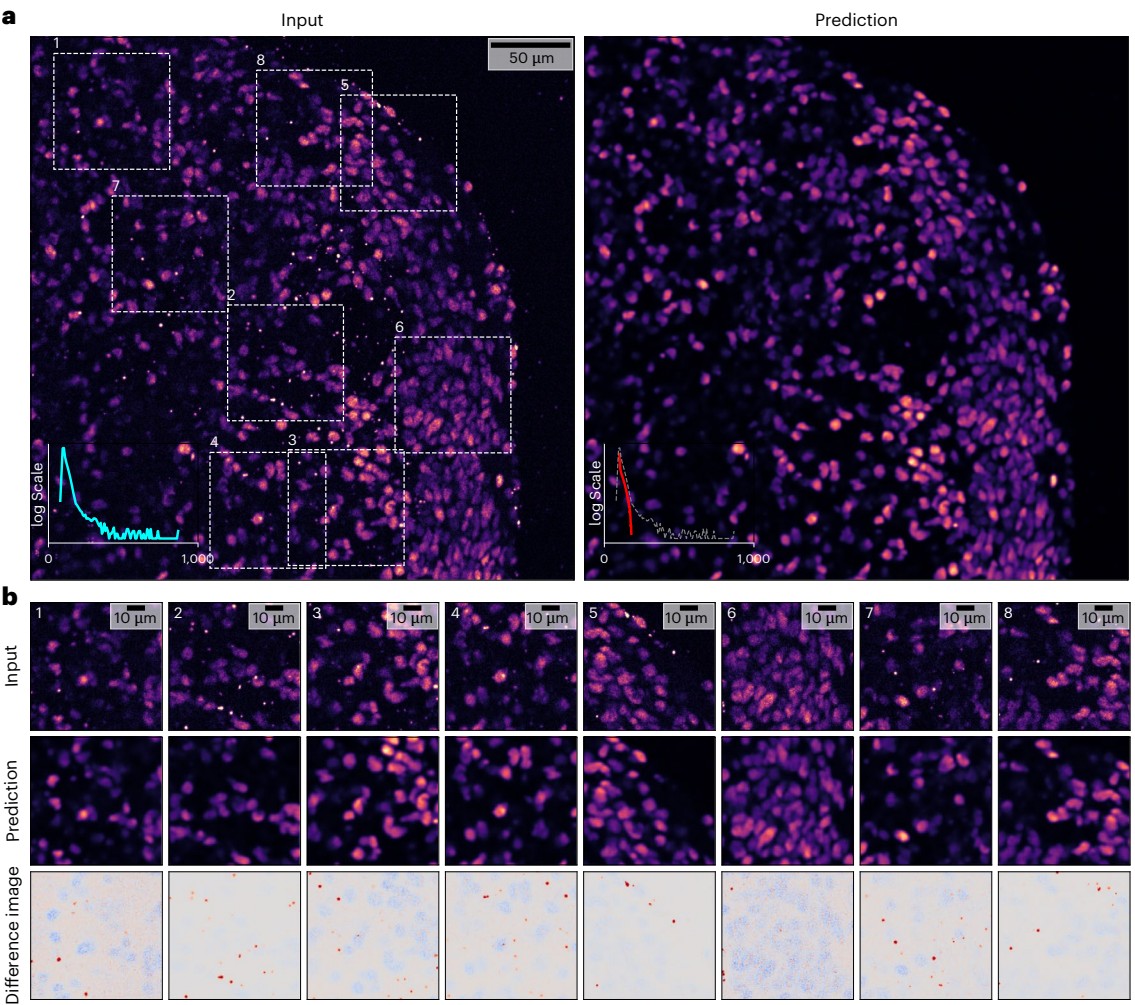

**Fig. 4 | Removing image artifacts with Micro§plit. a**, An example of a raw input image (left), showing labeled nuclei and unwanted puncta and the puncta-free prediction by Micro§plit. Both images show intensity histograms in cyan and red, respectively. The dashed gray plot in the predicted image corresponds to the cyan plot of the input. The *y* axes are shown on a logarithmic scale to better emphasize the puncta intensities Micro§plit successfully removed. **b**, For better visual comparison, we show input and prediction insets from **a** in the top two rows, respectively, and the difference images (input − prediction) in the bottom row (red, intensities removed; gray, intensities unchanged; blue, intensities increased).

crops we collected were 100 × 100 pixels), lateral contextualization (LC; Fig. 2a) could not be used and was therefore disabled during training. In total, we have manually cropped 82 content regions and 48 puncta regions from the data. This led to a total of 53.7 million pixels for training and took an analyst about 8 h of focused work.

After training had converged, we evaluated the model on previously held-out full-size microscopy images. A representative result is shown in Fig. 4a,b. The insets in Fig. 4b allow a more detailed comparison of the raw input data containing the undesired puncta and the predictions of the trained network. In the bottom row of the figure, we plot difference heat maps between input and prediction, which show how puncta are consistently absent in the predicted image while nuclei intensities remain mostly unaltered.

## Limitations of Micro§plit

In this study, we conduct an in-depth analysis of various factors that impact the performance of semantic unmixing. As semantic unmixing is a new method in the toolbox of microscopists, its full positive impact and its most useful application areas will only reach community consensus in the next few years. Here, we share our experience as accurately and unfiltered as possible and recommend practical and actionable steps to improve the achievable performance.

After working with semantic unmixing for a few years, we found the following data and network properties to be the key factors affecting the performance of Micro§plit: (1) multifold brightness differences (intensity skew) of the structures to be unmixed; (2) the SNR of the raw data; (3) the degree of structural (dis-)similarity between the structures to be unmixed; (4) helpful and reliable spatial correlations between the structures to be unmixed; and (5) the size of structures to be unmixed in relation to the input patch and receptive-field size of the used network architecture. In Supplementary Note 1 we further discuss limitations and mitigation strategies in greater detail.

To better understand the limitations of Micro§plit, we have varied the quality of image channels in two directions: SNR and relative brightness of two imaged channels (skew). More specifically, in the Pavia-P24 dataset, we have manually chosen three laser power settings per channel corresponding to a relatively high SNR, a medium SNR, and a low SNR setting. We then acquired nine sets of training data (for all combinations of illumination laser power settings). We quantify our findings in Supplementary Note 1.4, where we have trained nine Micro§plit networks, one for each set of training data mentioned above, and observed that a higher degree of skew (unequal channel intensities) makes it more difficult for Micro§plit to pick up the weaker of the two structures (Supplementary Table 3).

We also analyzed the effect of skewness with the HT-P23A (Task II) and Chicago-Sch23 (Task XX) datasets. In all cases, predictions seem fit for downstream processing, although we observe blurriness in predictions for the weaker channel in a skewed HT-P23A derived task. More details can be found in Supplementary Note 1.4, and in Supplementary Figs. 3 and 4.

To look closer at the effect of the overall SNR of the used training data, we acquired the HT-LIF24 dataset, which also partitions into multiple subsets that are imaged at increasingly shorter exposure times (and hence at lower SNR). Again, we have trained a Micro⑤plit network for each data subset. We ensured that the underlying content across these training sets is identical, which allowed us to carry out a quantitative evaluation of the performance drop as the SNR of the training data decreases. Quantitative results are shown in Table 1 and Supplementary Note 1.1 and qualitative examples are given in Extended Data Fig. 2 and Supplementary Fig. 1. Although the results clearly show the expected trend (semantic unmixing quality decreases when using lower SNR training data), even the shortest exposure time of 2 ms still leads to unmixed predictions that are fit for downstream processing and further analysis (in all cases, we measured a PSNR > 30.8 and Micro⑤MS-SSIM > 0.94, as shown in Table 1).

We additionally conducted experiments with synthetic Gaussian and Poisson noise on two sub-datasets from the HHMI-D25 dataset and observed that increased noise reduces the precision of fine structural details. Qualitative and quantitative results are provided in Supplementary Note 1.1.1 and Extended Data Figs. 3 and 4. These findings are consistent with previous observations made for denoi⑤plit[7]. Relative to high-SNR ground truth, Micro⑤plit tends to smooth fine structural details. Although other architectures may produce outputs that appear visually sharper, Micro⑤plit behaves in line with the well-understood behavior of models receiving ambiguous inputs: when the input does not contain enough information to uniquely determine the underlying structure, a model can either generate smooth, 'averaged' predictions that minimize expected error, or produce artificially sharp predictions that commit to one of several plausible interpretations. This trade-off is known as the perception-distortion trade-off[16,17], which states that beyond a certain point, improving perceptual sharpness necessarily increases reconstruction error and vice versa.

When working with noisy input images, predictions on full-sized micrographs, which are not fitting into GPU memory in one go, are prone to show some tiling artifacts[6,7]. As Micro⑤plit uses a variational network architecture and the data uncertainty in noisy inputs is higher, posterior samples (and even MMSE predictions) of neighboring tiles can have small intensity mismatches along their edges, causing the aforementioned tiling artifacts. Additionally, the used network architectures are deep, having receptive-field sizes well beyond the primary input patch being fed to the network. It is therefore advisable to use inner tiling instead of the more commonly used outer tiling, as previously described[6].

The potential downstream applications of Micro⑤plit include segmentation, tracking, detecting the presence or absence of certain structures, counting structures and estimating dimensional properties such as radius or length of structures of interest; however, we would not recommend using Micro⑤plit's prediction in downstream tasks that rely on precise pixel intensities.

Finally, as Micro⑤plit relies on distinguishable structural appearances (structural priors) for effective semantic unmixing, we performed a series of experiments to determine the sensitivity to structure similarity using the microtubule channel from the HT-LIF24 dataset. We created a series of increasingly more challenging two-channel splitting tasks by superimposing patches from the same microtubule channel, effectively mixing structurally identical data. To provide a single parameter that can render the semantic unmixing task increasingly simpler, we decided to scale one of the two superimposed copies, resulting in structurally similar yet controllably different appearances (unscaled and scaled by factors of 1.032, 1.063, 1.125, 1.25, 1.5 and 2). The results of these experiments, see Extended Data Figs. 5 and 6, show that Micro⑤plit can successfully unmix the structures starting at the scaling factor of 1.063 (see also Supplementary Note 1.3 for more details); however, network training performance improves more slowly (that is longer training is needed) as structural similarity increases, as evidenced by the delayed inflection point and convergence behavior in Extended Data Fig. 6. These results underscore the importance of structural dissimilarity for effective semantic unmixing, but they also highlight the capabilities of our method handling even very similar (only slightly scaled) distributions of structures.

## Discussion

In this work, we introduced Micro⑤plit, a new method that enables the semantic separation of multiple cellular structures superimposed in a single fluorescent image. Using VSE networks, we show that Micro⑤plit achieves high-quality semantic unmixing results for up to four superimposed cellular structures, while reliably denoising the given data and additionally providing a mechanism for estimating the error in its own predictions by evaluating the localized data uncertainty in the given input. This is particularly valuable in fluorescence microscopy, where the data are typically meant to be used for scientific downstream analyses, and the soundness of unmixed channels is essential.

Arguably, the greatest utility of Micro⑤plit lies in its ability to dramatically reduce photon exposure while still producing unmixed image channels of quality comparable to conventional multiplexed acquisitions. In the Section on 'Micro⑤plit enables a more effective use of the available photon budget', we quantitatively demonstrate, on a representative example, that Micro⑤plit enables about a tenfold overall reduction in emitted photos while still leading to comparable image quality. Such gains stem from two factors: (1) by filtering fewer photons during image acquisition (Extended Data Fig. 7), and (2) via Micro⑤plit's built-in denoising capability. In this way, Micro⑤plit effectively reduces the required photon budget, such that the additionally available photons can then be used to achieve faster imaging, higher signal-to-noise images or the imaging of additional structures that might previously not be imaged at the same time in the same sample due to photon budget limitations. This can be particularly beneficial in live-cell imaging, where minimizing phototoxicity and maximizing temporal resolution are critical for many real-world applications.

Moreover, Micro⑤plit's self-supervised denoising capability allows it to produce denoised predictions even when trained exclusively on noisy data. This feature is crucial for practical applications, as it reduces the need for high-quality, low-noise training data, which can be difficult or even impossible to acquire. The ability to estimate prediction errors through inter-sample variability further enhances the utility of Micro⑤plit, providing users with a measure of uncertainty that can guide the user to exclude unreliable images or image regions from further downstream analysis.

Micro⑤plit distinguishes itself from existing computational multiplexing techniques, such as spectral unmixing[18] or PICASSO[19], by requiring only a single superimposed image as input, rather than multiple images with different spectral overlaps. This not only simplifies the imaging process, but also allows for faster data acquisition, making Micro⑤plit particularly useful for live-cell imaging. Furthermore, our results demonstrate that in noisy conditions, Micro⑤plit performs better than PICASSO, highlighting Micro⑤plit's robustness and applicability to real-world microscopy data (see Section 'Micro⑤plit versus PICASSO' and Extended Data Fig. 8 for an in-depth comparison to PICASSO).

Last, Micro⑤plit can also be used to remove imaging artifacts, and we demonstrate this in the Section on 'Removing unwanted imaging artifacts'. This is possible when artifacts and structures can, at least occasionally, be seen in isolation (nonoverlapping in localized regions of the superimposed image). We believe that Micro⑤plit will be a useful

tool to remove structured noises (image artifacts), something that other content-aware denoising methods are generally not designed for[1,4,5].

Although Micro𝕊plit offers all the above-mentioned advantages, it naturally also has limitations users must be informed about. As elaborated on, in detail, in the Section 'Limitations of Micro𝕊plit', the performance of the model can degrade when the SNR of the input data becomes too low, when the relative intensities of the structures to be unmixed are highly skewed, or when the structures to be unmixed are too similar to each other.

While the experiments we presented give clear examples for how microscopists can benefit from Micro𝕊plit, they are only scratching the surface of what we believe microscopists and life scientists will be using Micro𝕊plit and its future derivations for. Not only will currently infeasible experimental setups become possible (imaging eight or even more cellular structures on a regular microscope setup to image up to four fluorescent channels is one such example), existing imaging protocols can be rendered more light-efficient, faster and cost-effective. We predict that also beyond fluorescence microscopy, image unmixing will find applications, for example, in various biomedical imaging modalities. Future work could additionally explore the integration of Micro𝕊plit with adaptive imaging strategies, where the imaging parameters (for example exposure time or laser power) are dynamically adjusted based on the uncertainty estimates provided by the model.

To make experimentation with Micro𝕊plit easy and to reduce the entry barrier as much as possible, we have developed a comprehensive open-source library CAREamics https://github.com/CAREamics/careamics and user-friendly example notebooks hosted at https://careamics.github.io for all experiments we conducted. To this end, we have made all training and evaluation data publicly available. Please find all data at https://github.com/CAREamics/MicroSplit-reproducibilitylinks-to-all-dataseets-used-in-the-manuscript. Hence, our work is fully transparent, enables others to reproduce our results, and reuse our methods and implementations to elevate the rate of their scientific discovery process[20,21].

In summary, Micro𝕊plit represents an advancement in semantic unmixing for fluorescence microscopy, enabling the simultaneous imaging of multiple structures in a single channel and addressing key limitations in imaging speed, resolution and photon budget. Its ability to provide high-quality denoised predictions of unmixed cellular structures, along with reliable uncertainty estimates, makes it a powerful and practical tool. We believe that Micro𝕊plit will find widespread use in a variety of biological imaging applications, enabling applications that are today simply not possible.

## Online content

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

## Methods

### Used microscopy datasets

Here, we describe the ten microscopy datasets we have used in this work, eight of which have been generated de novo by the authors; two others were taken from a previous publication[3]. All datasets will be publicly available, enabling full reproducibility of all experiments and results we have presented, and allowing others to directly compare their own methodological improvements to the approach we have presented. The datasets cover a broad spectrum of noise levels (SNR), number of imaged fluorescent channels, overall intensity and relative intensity of these channels, imaging modality (for example confocal, spinning disk confocal, structured illumination, live, fixed, and expanded samples, 2D and 3D). Using these datasets, we have trained MicroSplit models for a total of 36 tasks in total, covering two-, three- and four-channel semantic unmixing experiments (as shown throughout all figures and summarized in Table 1 and Supplementary Table 2). All datasets are openly available (https://github.com/CAREamics/MicroSplit-reproducibility#links-to-all-datasets-used-in-the-manuscript). Next, we describe the datasets. In Table 2, we provide a brief overview of the different datasets used in this work.

**The HT-P23A and HT-P23B datasets.** This dataset was acquired by the Pigino group at Human Technopole, Milan, Italy. The data generation procedure was based on previous work[23].

*Cryo-ExM of MDCK-II cells.* MDCK-II cells were seeded on 12-mm coverslips contained within six-well plates at a density of 300,000 cells per well, grown at 37 °C and 5% $CO_2$. When cells reached approximately 50% confluence, a subset of coverslips was incubated with 150 nM MitoTracker Deep Red FM (Thermo Fisher Scientific, M22426) for 30 min and checked for dye incorporation using an EVOS M5000 Imaging System (Thermo Fisher Scientific). Cells were then immediately processed for cryo-ExM1. In brief, cells on coverslips were rapidly plunged frozen in −180 °C liquid ethane using a manual plunger. Coverslips were transferred to an Eppendorf tube containing frozen, desiccated acetone supplemented with 0.1% paraformaldehyde (PFA) and 0.02% glutaraldehyde (GA) for overnight freeze substitution in dry ice. The next morning, the coverslips underwent successive ethanol baths in progressively decreasing percentages, as follows: ethanol 100% (5 min), ethanol 100% (5 min), ethanol 95% (5 min), ethanol 95% (5 min), ethanol 70% (3 min), ethanol 50% (3 min) and PBS. Substituted cells then underwent anchoring in 2% acrylamide and 1.4% formaldehyde in PBS for 3 h. Coverslips were then exposed to activated monomer solution containing sodium acrylate (19%), acrylamide (10%), N,N'-methylenbisacrylamide (0.1%) and ddH$_2$O, activated by APS and TEMED (0.05%). Expansion gels polymerized for 1 h at 37 °C, and were then denatured in denaturation solution containing SDS (200 mM), NaCl (200 mM) and Tris (50 mM) in ultrapure water at 95 °C for 1.5 h. Expanded gels were then washed in PBS 3×, 10 min for each wash. Final concentrations of solutions are reported.

*Expansion immunolabeling.* The following day, gels were stained with anti-α-tubulin (ABCD Antibodies, ABCD_AA345) and anti-β-tubulin antibodies (ABCD antibodies, AA_344), diluted 1:300 in PBS 2% BSA. Half of the MitoTracker labeled gels were given Mouse IgG2a raised antibody, and the rest of the gels were stained with guinea pig IgG raised antibody. Tubulin-labeled gels were incubated at 37 °C for 3 h while gently shaking. After primary labeling, gels were washed 3× in PBS 0.1% Tween, 10 min for each wash. Mouse IgG2a tubulin-labeled gels were then stained with goat anti-mouse IgG secondary antibody Alexa Fluor 488 (Thermo Fisher Scientific, A-11001), while guinea pig IgG tubulin-labeled antibodies were stained with goat anti-guinea pig IgG secondary antibody Alexa Fluor 647 (Thermo Fisher Scientific, A-21450). Secondary antibody labeled gels were then incubated at 37 °C for 3 h while gently shaking. The gels were washed 3× in PBS 0.1%

Tween, 10 min for each wash. Gels were further washed 2×, 30 min in ddH$_2$O, and then placed in ddH$_2$O overnight for expansion.

In HT-P23A, the fluorescence intensity of labeled microtubules strongly overpowered the mitochondrial signal in the condition in which both microtubules and mitochondria were co-fluorescent in the 647 channel. To ensure a more equal signal, we imaged HT-P23B, where we reduced the concentration of tubulin guinea pig IgG labeling to 1:450, and the concentration of anti-guinea pig IgG secondary antibody to 1:450 to ensure a more equal signal.

*Imaging expansion gels.* Expanded gels were cut into approximately 1.5-cm$^2$ pieces and placed on a 24-mm coverslip coated in poly-D-lysine (Sigma, P7886) and placed within a 35-mm imaging chamber. Imaging was performed on a Zeiss LSM980-NLO, in confocal mode. z-stacks with a step size of 0.15 μm were collected with a frame size of 2,024 × 2,024 pixels using a ×63 oil immersion objective (1.4 NA).

**The Pavia-P23 dataset.** This dataset was acquired at the Synthetic Physiology Laboratory at the University of Pavia, Italy.

*HaCaT cell line culture and medium.* The HaCaT cell line was kindly gifted by H.de Jonge (Department of Molecular Medicine, University of Pavia). Cells were maintained in DMEM/F12 no phenol red (Gibco, 21041-025) with 10% fetal bovine serum (FBS) (Gibco, 10270-106) and 1% penicillin−streptomycin (P−S) (HiMedia, A001). Cells were never allowed to go beyond 75−80% confluency during routine splitting.

*Generation of the HaCaT FUCCIplex-prototype clonal cell line.* The HaCaT cell line expressing the structural actin and tubulin fluorophores and the FUCCIplex sensor was generated as described previously[24]. In brief, cells were plated at the density of $6 × 10^4$ cells per well on a 12-well plate and infected with the FUCCIplex lentivirus (10 μl of concentrated FUCCIplex lentivirus particles (>$10^8$ TU ml$^{-1}$, VectorBuilder). Positive cells were then selected with 20 μl ml$^{-1}$ of hygromycin B (50 mg ml$^{-1}$ PBS) (Invitrogen, 10687010) and expanded. Positive cells expressing the FUCCIplex sensor were then infected with a genetically encoded Lifeact-ACTB-RFP probe (rLV Ubi-LifeAct-TagRFP, $1 × 10^7$ TU ml$^{-1}$, 10 μl) that specifically tags the (F)-actin filaments with a red fluorescent reporter. Last, the α-tubulin locus (TUBA1B, NM006082.3) was genome-edited via CRISPR/Cas9 using the Thermo Fisher Scientific True Tag system (A42992) to tag the tubulin N terminus of the protein with an EGFP fluorophore. Positive cells were clonally selected and expanded. The True Tag homology arm primer forward sequence is TCCTGTCGCCTTCGCCTCCTAATCCCTAGCCACTATGGGAGGTAAGCC-CTTGCATTCG and the reverse is CCTGAAAGCAGCCGGGAGCCGCACG-GCTTACTCACACCGCTTCCACTACCTGAACC. The TrueGuide synthetic gRNA (sgRNA) sequence is GCACGGCTTACTCACCATAG. For imaging experiments, HaCaT cells were plated in culture medium into porcine skin-coated (0.2%, w/v) ibidi μ-slide eight-well-chambered coverslips (Ibidi, 80807) at a density of 50,000 cells per well.

*Imaging methods for Pavia-P24 dataset.* Imaging was performed using a Nikon Ti2 Eclipse inverted microscope integrated with a Crest V3 X-Light spinning disk confocal unit and a Teledyne Photometrics Kinetix Scientific sCMOS camera. The dataset was acquired using a CFI SR HP Plan Apo Lambda S ×100C objective lens (silicon oil immersion, NA 1.35, WD 0.31−0.28 mm, MRD73950).

A Lumencore Celesta Light Engine (TSX5030FV, Lumencore) provided illumination for fluorescence imaging, delivering light at multiple wavelengths (405 nm, 446 nm, 477 nm, 520 nm, 546 nm, 638 nm and 740 nm).

Specifically, the dataset was acquired using 446-nm and 477-nm laser lines. In the 477-nm line optical configuration, a multiband dichroic and excitation filter set (MXR00543-CELESTA-DA/FI/TR/Cy5/Cy7-A, CELESTA) was used, along with a FITC emission filter

(MXR00541 FITC, CELESTA). Instead, the 446 nm optical configuration was equipped with dual-band filters for dichroic, excitation and emission (MXR00544, CELESTA CFP/YFP, CELESTA).

In addition, the microscope has an advanced environment setup to support live imaging experiments. The Oko-Cage incubation system (Okolab) equipped with environment controllers can maintain optimal conditions for live-cell imaging, including a temperature setpoint of 37 °C, passive humidity control and $CO_2$ levels of 5%.

Nikon NIS Elements software (v.5.42.06, Nikon) was used to manage the imaging system. Spinning disk confocal mode was selected for this dataset of two-channel $z$-stacks. In detail, channels 477 nm and 446 nm were imaged sequentially with identical exposure settings (200 ms). Splitting performance was tested by changing two factors: the SNR level (high, mid and low) and the relative laser power percentage of each channel, as shown in Supplementary Table 10. Multiple $z$-stacks of six planes with a step size of 1 μm were acquired to populate the dataset for each condition.

**The HT-H24 dataset.** The HT-H24 dataset was imaged by the Harschnitz group at Human Technopole and contains immunofluorescent staining of SOX2 (555) and MAP2 (488) in DIV25 dorsal forebrain organoids generated from WTC-11 (UCSFi001-A) induced pluripotent stem (iPS) cells (UCSFi001-A). Human iPS cell-derived forebrain organoids were generated following a triple-inhibition patterning protocol, without LIF supplementation, as published in a previous work[25]. At days in vitro (DIV) 25, forebrain organoids were fixed in 4% paraformaldehyde at 4 °C overnight, transferred in PBS with 30% sucrose until fully immersed, embedded in OCT compound (Scigen), frozen rapidly, and stored at −20 °C. The 15-μm-thick sections were sliced on a cryostat (Leica). For immunofluorescence staining, organoid slices were rinsed in PBS, incubated with Retrieval Solution (Dako) for 45 min at 70 °C, permeabilized with 0.5% Triton X-100 in PBS 1× for 20 min and blocked with 0.1% Triton X-100, 10% donkey serum in PBS 1× for 1 h at room temperature. Sections were incubated with either rat α SOX2 1:200 (Invitrogen, 14-9811-82) and chicken α MAP2 1:5,000 (Invitrogen, PA1-10005) or rat α CTIP2 1:500 (Abcam, ab18465) overnight, and counterstained with 1:1,000 donkey α chicken Alexa Fluor 488 (Jackson ImmunoResearch, 703-545-155) or 1:1,000 donkey α rat Alexa Fluor Plus 555 (Invitrogen, A48270). Nuclei were stained with DAPI (Thermo Fisher, 62248). Immunostained sections were scanned using Ti2 CREST spinning disk (Nikon), with an objective magnification of ×40.

**The HT-T24 dataset.** This dataset is a three-channel confocal microscopy imaging of fixed E37 ferret brain sections acquired by the Taverna group at Human Technopole. It is a three-channel 2D dataset. The first two channels contain the SOX2 (transcription factor used to label stem cell nuclei) and Golgi marker Grasp65, and the last channel contains the superimposed image containing the above-mentioned two structures. For more details, please look into the metadata of the dataset files. 'Organism' is ferret (*Mustela furo*) and the 'Sample' is cryosection of E37 ferret brain stained with SOX2 and GRASP65 antibodies.

*Experimental animals.* All experimental procedures were conducted in agreement with the German Animal Welfare Legislation after approval by the Landesdirektion Sachsen (license for ferret TVV2/2015). Animals used for this study were kept in standardized hygienic conditions at the Biomedical Services Facility (BMS) of the MPI-CBG with free access to food and water. All experiments were performed in the dorsolateral telencephalon of ferret embryos, at a medial position along the rostro–caudal axis at a stage corresponding to mid-neurogenesis.

*Protocol for immunofluorescence staining.* After incubation at 70 °C for 30 min in Dako Target Retrieval Solution, citrate, pH 6, cryosections were permeabilized with 0.3% Triton X-100 in PBS for 30 min at room temperature. Blocking was performed in a blocking solution (0.2% gelatin, 300 mM NaCl and 0.3% Triton X-100 in PBS) for 30 min. Primary antibodies were incubated in the blocking solution overnight at 4 °C. The following antibodies were used: SOX2 (goat polyclonal, AF2018, 1:200 dilution, R&D Systems) and GRASP65 (rabbit polyclonal, PA3-910, 1:200, Invitrogen). Subsequently, the sections were washed three times in PBS and incubated for 1 h at room temperature with the following secondary antibodies: donkey anti-goat IgG (H+L) highly cross-adsorbed secondary antibody, Alexa Fluor Plus 555 (A32816, 1:500 dilution, Invitrogen) and donkey anti-rabbit IgG (H+L) highly cross-adsorbed secondary antibody, Alexa Fluor Plus 647 (A32795, 1:500 dilution, Invitrogen). After three washes in PBS, stained sections were mounted with Mowiol.

*Acquisitions of HT-T24 dataset.* All images were acquired using a spinning disk confocal system, consisting of a CrestOptics V3 Light scanhead (configured with 50-μm pinholes) mounted on a Nikon Ti2-E inverted microscope equipped with a motorized stage and four Photometrics Prime 95B 25 mm cameras (pixel size 11 μm). The samples were acquired in confocal mode with a Plan Apochromat Lambda S ×100/1.35 silicon immersion objective using Celesta Lumencor solid-state lasers as the light source. Fluorescence was collected using the following elements: channel 1 (C1), excitation wavelength 638 nm laser lines at 20%, excitation filter and dichroic mirror MXR00543-CELESTA-DAPI/FITC/TRITC/Cy5/Cy7-Full Multiband Penta, Cy5 emission filter; Channel 2 (C2), excitation wavelength 547 nm laser lines at 20%, excitation filter and dichroic mirror MXR00543-CELESTA-DAPI/FITC/TRITC/Cy5/Cy7-Full Multiband Penta, TRITC emission filter; superimposed channel (Input), excitation wavelength 547 nm and 638 nm laser lines both at 20%, excitation filter and dichroic mirror MXR00543-CELESTA-DAPI/FITC/TRITC/Cy5/Cy7-Full Multiband Penta, no emission filter. All images were acquired with the same camera parameters: binning 1, 16-bit and 400 ms of exposure time. For every field of view, a $z$-stack with a 1-μm-step size was acquired. Once the parameters of acquisition had been defined, they were kept constant. The software used for all acquisitions was NIS Elements AR 5.42.02 (Nikon).

**The HT-LIF24 dataset.** This dataset was acquired at the Light Imaging Facility at Human Technopole, Milan, Italy.

*Sample preparation of HT-LIF24 dataset.* HeLa cells were maintained in DMEM (EuroClone) containing 10% FBS (Thermo Fisher Scientific), supplemented with 2 mM L-glutamine (EuroClone) and penicillin/streptomycin both 100 μg ml$^{-1}$ (EuroClone), at 37 °C in a humidified atmosphere and 5% $CO_2$. Cells were plated onto glass number 1.5 coverslips for immunofluorescence microscopy and then fixed with 4% PFA for 10 min. They were permeabilized with 0.1% Triton X-100 and 0.2% BSA in PBS for 10 min. Blocking was performed in a blocking solution (2% BSA in PBS) for 30 min. Primary antibodies were incubated in the blocking solution for 1 h. The following antibodies were used: anti-α-tubulin mouse IgG monoclonal (Sigma-Aldrich, T5168; 1:100 dilution); anti-laminin B1 rabbit IgG polyclonal (Abcam, ab16048; 1:200 dilution); and anti-centromere protein human IgG polyclonal (Antibodies Incorporated, 15-234; 1:400 dilution). Subsequently, after washing in PBS, cells were incubated for 40 min with the following secondary antibodies in blocking solution- Alexa Fluor 488 donkey anti-mouse IgG (Thermo Fisher Scientific, A-21202; 1:400 dilution); Cy3 donkey anti-rabbit IgG (Jackson Immunoresearch, 711-165-152; 1:400 dilution); and Cy5 goat anti-human IgG (Jackson Immunoresearch, 109-175-088; 1:50 dilution). Finally, cells were also counterstained with 4,6-diamidino-2-phenylindole (DAPI) (Sigma-Aldrich, D9542; 1:40,000 dilution) for 15 min before the mounting step. All the steps were performed at room temperature. The samples were then mounted in Mowiol-DABCO and acquired with a spinning disk system as described below. At least 200 nonoverlapping and randomly distributed fields of view were acquired and analyzed.

*Acquisitions of HT-LIF24 dataset.* All images were acquired using a spinning disk confocal system, consisting of a CrestOptics V3 Light scanhead (configured with 50-µm pinholes) mounted on a Nikon Ti2-E inverted microscope equipped with a motorized stage and a Photometrics Prime 95B 25-mm camera (pixel size 11 µm). The samples were acquired with a Plan Apochromat Lambda S ×40/1.25 silicon immersion objective using Celesta Lumencor solid-state lasers as the light source. Fluorescence was collected using 19 different lightpath configurations. In each configuration, a penta-band excitation filter (MXR00543-CELESTA-DAPI/FITC/TRITC/Cy5/Cy7-Full Multiband Penta), a penta-band dichroic filter (MXR00543-CELESTA-DAPI/FITC/TRITC/Cy5/Cy7-Full Multiband Penta) and a penta-band emission filter (Semrock FF01-441/511/593/684/817-25) were used. Ground truth images were acquired using a specific additional band-pass emission filter inserted before the penta-band emission filter (in particular the DAPI, FITC, TRITC and Cy5 emission filters were added for GT-A, GT-B, GT-C and GT-D configurations, respectively), whereas for all the other channels only the penta-band emission filter was used. The excitation wavelengths of the 19 channels were set as follows: GT-A, 405 nm (40%); GT-B, 477 nm (35%); GT-C, 547 nm (5%); GT-D, 638 nm (50%); A, 405 nm (40%); B, 477 nm (35%); C, 547 nm (5%); D, 638 nm (50%); AB, 405 nm (40%), 477 nm (35%); AC, 405 nm (40%), 547 nm (5%); AD, 405 nm (40%), 638 nm (50%); BC, 477 nm (35%), 547 nm (5%); BD, 477 nm (35%), 638 nm (50%); CD, 547 nm (5%), 638 nm (50%); ABC, 405 nm (40%), 477 nm (35%), 547 nm (5%); ABD, 405 nm (40%), 477 nm (35%), 638 nm (50%); ACD, 405 nm (40%), 547 nm (5%), 638 nm (50%); BCD, 477 nm (35%), 547 nm (5%), 638 nm (50%); and ABCD, 405 nm (40%), 477 nm (35%), 547 nm (5%), 638 nm (50%). All images were acquired at binning 1. For each field of view, a series of images was captured using the following exposure times: 2 ms, 3 ms, 5 ms, 20 ms and 500 ms. The software used for all acquisitions was NIS Element AR 5.42.02 (Nikon).

Simply put, the data contain four different structures: (1) the whole nuclei (DAPI staining); (2) microtubules, one component of the cytoskeleton, which is a filamentous system in the cytoplasm (α-tubulin staining); (3) nuclear envelope, the membrane that separates the nucleus from the cytoplasm (lamin B1 staining); and (4) the kinetochore/centromere-specific area along the chromosomes-DNA, which is used to connect the chromosomes themselves to the microtubules during mitosis (CREST staining).

**The Chicago-Sch23 dataset.** This dataset is four-color structured illumination super-resolution microscopy imaging of live human BJ fibroblast cells acquired at the Scherer Lab at the University of Chicago, Chicago, USA[26–29]. It has four channels of different structures: (1) actins (CellMask Orange); (2) mitochondria (MitoTracker Green); (3) microtubules (Tubulin Tracker Deep Red); and (4) nuclei (Hoechst).

*Sample preparation.* Human BJ fibroblast cells were cultured in high-glucose DMEM (Life Technologies, 10569) supplemented with 10% FBS (Life Technologies, 26140) and penicillin–streptomycin. Cells were maintained in a humidified incubator at 37 °C with 5% carbon dioxide. Before imaging, live cells were washed with PBS (Life Technologies, 15140) and trypsinized using 2.5 ml of 0.05% trypsin (Life Technologies, 25300) at 70–80% confluence. The cells were then transferred to 35-mm glass-bottom dishes (MatTek, P35G-1.5-14-C) for microscopy.

The staining solution was prepared by diluting Tubulin Tracker Deep Red (Thermo Fisher Scientific, T34077) to 1×, CellMask Orange Actin Tracking Stain (Thermo Fisher Scientific, A57247) to 2×, MitoTracker Green FM (Thermo Fisher Scientific, M7514) to 100 nM, and Hoechst 34580 (Thermo Fisher Scientific, H21486) to 5 µg ml⁻¹ in growth medium. For imaging, cells were incubated with 1 ml of the staining solution for 30 min at 37 °C, rinsed five times with FluoroBrite DMEM (Thermo Fisher Scientific, A1896701) and subsequently imaged and analyzed in FluoroBrite DMEM.

*Multicolor structured illumination microscopy super-resolution imaging.* A custom-built structured illumination microscope (SIM) was used for multicolor imaging. Excitation wavelengths of 642 nm (Spectra-Physics, Excelsior One 642), 532 nm (Spectra-Physics, Millennia V), 488 nm (Spectra-Physics, Excelsior One 488) and 405 nm (Spectra-Physics, Excelsior One 405) were employed to excite Tubulin Tracker, CellMask Orange, MitoTracker and Hoechst, respectively. The four laser beams were combined using three dichroic mirrors (Semrock, Di03-R405-t1; Semrock, Di03-R488-t1; Thorlabs, DMSP550R) and subsequently expanded by 4×. The combined beams were first diffracted into monochromatic beams by a blazed grating (Thorlabs, GR13-0605) and then recombined at the plane of a digital micromirror device (DMD; Texas Instruments, DLP9000X VIS WQXGA). The DMD-generated patterns were projected onto the sample plane through an objective lens (Nikon, SR Plan Apo, ×60, 1.27 WI). A multiband dichroic mirror (Semrock, Di01-R405/488/532/635-25x36) and an emission filter (Semrock, FF01-446/510/581/703-25) were used to separate excitation and emission light. Additionally, a 2× beam expander was employed to further magnify the emission signal, resulting in a total magnification of ×120. Fluorescence images were captured using an sCMOS detector (Photometrics, Kinetix).

For SIM super-resolution imaging, striped binary patterns with a second-order spatial frequency of 2.86 µm⁻¹ at the sample plane were displayed on the DMD. Patterns at three different angles with six phase shifts each were sequentially projected, with an exposure time of 100 ms per pattern. Super-resolution SIM reconstruction was performed using FairSIM 2, an ImageJ-based open-source software. Each raw image stack of 2,048 × 2,048 × 18 was reconstructed into 4,096 × 4,096 super-resolution images, where each pixel corresponds to 27 nm.

**The HHMI-D25 dataset.** *Animal experiments.* Heterozygous PhAMexcised female mice carrying the Mito-Dendra2 transgene were generated through in-house breeding. First, PhAMexcised heterozygous males and females (strain #018397, The Jackson Laboratory) were crossed to derive homozygous males, which were then bred with wild-type C57BL/6J females. Mice were housed in sound-attenuated, temperature- and humidity-controlled rooms under a 12-h light–dark cycle, with food and water provided ad libitum. All procedures were conducted in accordance with National Institutes of Health guidelines and were approved by the Institutional Animal Care and Use Committee at the Janelia Research Campus (protocol number 22-0229.04). Livers were collected via cardiac perfusion: first with 1× PBS to remove blood, followed by 30 ml of 4% paraformaldehyde (PFA) at a flow rate of 2.5 ml min⁻¹ to minimize endothelial damage. Tissues were post-fixed in 4% PFA for 24 h, rinsed three times in 1× PBS and stored in 1% PFA until further processing.

*Immunostaining and image acquisition.* For imaging, samples were embedded in 4.6% low-melting-point agarose and sectioned into 120-µm slices using a Leica VT 1200S vibratome. Sections were blocked in 10% FBS with 0.5% Triton X-100 for 1 h and incubated with primary antibodies at 4 °C for 48 h. Mouse anti-PMP70 (MilliporeSigma, SAB4200181, 1:75 dilution) and rabbit anti-LAMP1 (Abcam, AB208943, 1:50 dilution) were used to label peroxisomes and lysosomes, respectively. Secondary antibodies Alexa Fluor 647 goat anti-mouse (Thermo Fisher, A21235, 1:500 dilution) and Alexa Fluor 750 goat anti-rabbit (Thermo Fisher, A21039, 1:500 dilution) were applied overnight at 4 °C. Additional markers included Alexa Fluor Plus 555 Phalloidin (Thermo Fisher, A30106, 1:100 dilution) and HCS LipidTox Red (Thermo Fisher, H34476, 1:100 dilution) to label actin and lipid droplets, respectively. Nuclei were stained with DAPI (Thermo Fisher, D3571, 1:1,000 dilution of 1 mg ml⁻¹ stock) during a 25-min PBS wash the following day. Sections were cleared using EasyIndex (LifeCanvas Technologies, EI-500-1.52) by incubating samples first in 50% EasyIndex for 1 h, followed by 100% EasyIndex for 3–5 h, and mounted using Secure-Seal spacers (Thermo

Fisher, 0523073). Imaging was performed on a Leica Stellaris 8 confocal microscope using a ×63/1.40 NA oil immersion objective. Images were acquired at 2,048 × 2,048 pixels using bidirectional scanning, ×2 optical zoom, and a pinhole size of 0.5 Airy units. A total depth of 10 µm was captured across 50 z-sections. Fluorophores were imaged using two acquisition lines: the first included mitochondria, actin and lysosomes, while the second included nuclei, lipid droplets and peroxisomes.

## Evaluation metrics

In this work, we have predominantly used the metrics CARE-PSNR[3] and MicroMS-SSIM[8], which are slight alterations of PSNR (peak SNR) and SSIM, respectively[1]. We did so, as CARE-PSNR and MicroMS-SSIM are designed to better assess the quality of fluorescence microscopy images than their classical alternatives[3,8]. Please refer to Supplementary Note 3 for more details.

## Model architecture and training

As shown in Fig. 1a, a superimposed image patch is fed to MicroSplit as input. For a $k$ channel semantic unmixing task (in this work, $k \in [2, 3, 4]$), MicroSplit outputs $k$ predicted images, each containing one of the structures that are superimposed in the given input patch.

In MicroSplit, we have combined the benefits of µSplit[6] and denoiSplit[7], cast those ideas in a common learning framework, enabled direct training and prediction on volumetric data, and have extensively tested and evaluated its performance on a wide range of datasets and semantic unmixing tasks. We have also made openly available all training and prediction code and all data we used. We do this to foster rapid adoption of MicroSplit by the scientific community and to allow others to improve our approach and compare our results with relative ease. In this section, we will discuss in detail the aspects of the above-mentioned works that were integrated into MicroSplit. Subsequently, we also describe our loss function and important training hyperparameters.

From µSplit, we inherit the ability to efficiently incorporate spatial context using additional inputs, called LC inputs[6]. Here we feed, next to the primary input, for which the prediction will be made, additional LC inputs that help the network to better understand the image context from which the primary input is taken (Fig. 2a and Extended Data Fig. 1). These successive LC inputs are larger and larger patches centered on the primary input patch, but downscaled to the same pixel dimensions as the primary input itself. Hence, LC inputs capture the spatial context around the primary input patch but do so at lower resolutions to ensure efficient learning and predictions in a reasonably sized overall network.

The network architectures we proposed come in flavors that trade computational complexity and GPU consumption with the best-possible prediction quality. Its most GPU-efficient variant, Lean-LC, can train on a single GPU using less than 5 GB of GPU memory. If more resources are available, it is advisable to opt for setups such as Deep-LC, which show better predictive performance at increased computational cost.

In Extended Data Fig. 1, we have briefly described the three µSplit variants. The white regions correspond to features originating from the primary input patch. As in U-Net architectures, spatial resolution halves at each successive hierarchy level through pooling operations, causing these white areas to progressively diminish in size. In µSplit (and hence also in MicroSplit), the pooled embeddings undergo zero-padding before being concatenated with feature maps from lateral contextualization (LC) inputs. These LC features are processed through dedicated 'Input branch' sub-networks consisting of convolutional layers with nonlinear activations, dropout, and normalization components. 'Input branch' does not have pooling operations, and so their feature maps, at each hierarchy level, maintain spatial dimensions identical to the primary input (represented by gray regions). This preservation enables merged embeddings from both pathways to retain the original spatial dimensions throughout the network

hierarchy (gray squares). This is the core idea of LC, which MicroSplit has inherited from µSplit. In the caption of Extended Data Fig. 1, we provide more details regarding the differences in the three variants of µSplit. Ref. 6 provides further details.

From denoiSplit, we inherit the ability to jointly perform unsupervised denoising, using suitable noise models. This also enables MicroSplit to sample diverse predictions from a learned approximate posterior that captures a notion of the data uncertainty, as demonstrated by our trained networks being calibrated (see Section on 'Error estimation, data uncertainty and calibration'). While also µSplit is a variational approach that is, in theory, capable of generating multiple predictions from its posterior, we found that denoiSplit, arguably due to its different Kullback–Leibler (KL) loss formulation, produces a higher diversity that is better in line with the uncertainty in the data. In Extended Data Fig. 1, we present the architecture of MicroSplit, which has LC inputs and noise models, all integrated into a single setup.

As mentioned above, we also enabled MicroSplit to operate directly on volumetric image data, a possibility that was absent in both µSplit and denoiSplit.

**Loss function used to train MicroSplit.** The loss function of MicroSplit is the weighted average between the µSplit loss and denoiSplit loss:

$$\text{loss}_{\text{MicroSplit}} = w \times \text{loss}_{\text{denoiSplit}} + (1 - w) \times \text{loss}_{\text{µSplit}} \tag{1}$$

Unless explicitly specified, $w = 0.9$ is used in all experiments we conducted. This simple design also gives us the ability to switch to pure µSplit or denoiSplit mode by simply setting $w$ to 0 or 1, respectively.

To incorporate LC inputs into the denoiSplit setup, we observed the need to modify the KL loss formulation used in $\text{loss}_{\text{denoiSplit}}$. In denoiSplit, pixel-wise KL divergence is computed at every hierarchy level. Let $\text{KL}_i$ denote the pixel-wise KL divergence tensor at the $i$th hierarchy level. KL loss component for this hierarchy level, $\text{kl}_i$ is defined as

$$\text{kl}_i = \alpha \times \sum_{j,h,w} \text{KL}_i[j, h, w]. \tag{2}$$

With LC inputs, the spatial dimensions of the latent space tensors and therefore $\text{KL}_i$ do not decrease and the summing operation in this formulation leads to a higher value, owing to the larger number of summands (which are all non-negative). This gives unnecessarily high weight to the KL loss with respect to the likelihood loss component. We observed that this degrades the performance. To handle this, we center-cropped $\text{KL}_i$ to the shape they would assume if there were no LC inputs. Let $\text{KL}_i^{\text{cropped}}$ denote the appropriately center-cropped version of $\text{KL}_i$. Our modified KL loss for denoiSplit becomes

$$\text{kl}_i = \alpha \times \sum_{j,h,w} \text{KL}_i^{\text{cropped}}[j, h, w]. \tag{3}$$

We encountered a very similar issue when working with volumetric data. In that case, pixel-wise KL divergence is a four-dimensional tensor $C \times Z \times H_i \times W_i$. As we work with larger and larger $Z$, the summation in equation (2) would increase since the summation would be on all 4 dimensions. This again leads to giving more weight to the KL loss component against the likelihood loss, thus rendering the performance inferior. Note that this affect will become more severe when we increase the number of $z$ frames in the input. In other words, adding more information in the input was not beneficial. To handle this, we separately took care of the extra $z$ dimension by taking the average along this dimension. The resultant 3D tensor is then passed to equation (3) to compute the KL loss component. Note that there is an additional dimension of batch size which we have not mentioned in the above explanation. That is because KL loss is computed separately for every element in the batch.

**Hyperparameters used during training.** We use PyTorch package for creating our training and evaluation pipelines. We use a batch_size of 32, max_epoch of 400 and learning_rate of 0.001. We use Adamax optimizer and ReduceLROnPlateau as the learning rate scheduler with lr_scheduler_patience set to 150. During training, we use 16-bit precision. We use two LC inputs in our Deep-LC configuration (multi-scale_lowres_count = 3). Please refer to our code (https://github.com/CAREamics/MicroSplit-reproducibility) for more details. All experiments were conducted using code hosted at https://github.com/juglab/MicroSplit; however, we developed https://github.com/CAREamics/MicroSplit-reproducibility with the objective of providing user-friendly code with easier adaptability to custom datasets.

## Additional experiments

**MicroSplit versus PICASSO.** In this section, we compare the performance of MicroSplit with PICASSO[19] (also Extended Data Fig. 8). For semantic unmixing $k$ channels MicroSplit needs as input a single superimposed image from the microscope whereas PICASSO needs $k$ images from the microscope, which correspond to $k$ spectrally overlapping fluorophores. Due to this mismatch in the data requirement, a direct comparison is not feasible. So, to compare MicroSplit with PICASSO, we generate synthetic inputs using our HT-LIF24 dataset. However, we argue that our way of generation does not degrade the performance of PICASSO but instead, it should be easier for PICASSO to predict on this data as compared to the real data.

As fluorophores can be ordered according to the wavelength of the maximum intensity in their emission spectra, we first define such an order of our structure types. Next, for generating every channel of the input for PICASSO, we define three weights and take the weighted average of the three structures using these weights. The weights are set according to the order of the structures set above. For example, for the first channel, the weight given to the second structure will be higher than the weight given to the third structure. We generate two sets of weights, one being harder than the other. In the hard case, the dominant structure type is given 1.5 times more weight than the next dominant type and three times more weight than the least dominant type. In the easy version, the dominant structure type is given 2.5 times more weight than the next dominant type and five times more than the least dominant type. Once the input channels are generated, we add Gaussian noise of $\sigma = 500$ to each channel independently. We also train MicroSplit on this data with the same Gaussian noise applied on top of the data. In Extended Data Fig. 8, we show the results on one random frame. In Supplementary Tables 7 and 8, we show the quantitative results.

**Training Mode I versus Training Mode II (how important are spatial correlations?).** In this experiment, we inspect the performance drop between data acquisition types I and II. In cells, the location of different structures is often quite co-related with one another. For example, actin is mostly concentrated on the cell periphery whereas the nucleus is typically found at the center of the cell. In acquisition modes I and II, input is created by summing the crops from individual channels. In acquisition type I, as the channels are independently acquired, summing the crops of these different channels will generate an input patch where the naturally occurring colocation property cannot be preserved. In acquisition type II, since both channels are concurrently acquired, the inputs are created from summing the crops of different channels with each crop taken from the same location in the micrographs and therefore these inputs preserve the naturally occurring colocation property. In this experiment we quantify the benefit of using this colocation information.

We work with the HT-T24 dataset, which falls under Training Mode III. We train three models. In the first model, we create the input using Training Mode I (we create the input by picking target patches from the same location and therefore maintain the spatial co-relation). In the second model, we use Training Mode II, meaning that we pick crops

from random locations from the different channels and use them to create the input. This model naturally does not have access to naturally occurring spatial co-relation information in its training data. The third model is trained using Training Mode III. We evaluate all three models on the held-out test set where the inputs have spatial co-relation preserved and are not synthetic (they are imaged from the microscope). In Supplementary Table 4, we find that the first model outperforms the second by 1.3 CARE-PSNR and 0.009 MicroMS-SSIM. Naturally, the model trained with Training Mode III is best and outperforms Training Mode I by 0.6 CARE-PSNR and 0.004 MicroMS-SSIM.

**Training Mode I versus Training Mode III (summed versus acquired inputs).** Here, we quantify how much the performance degrades if, during training, the input is created by simply summing the two channels as compared to input coming directly from the microscope. We find that while there is indeed a performance drop as can be seen in Supplementary Table 4, the drop is not detrimental. This experiment shows the utility of our approach in the case when synthetic input is used for its training but for evaluation, inputs coming directly from the microscope are used.

**MicroSplit enables a more effective use of the available photon budget.** By filtering fewer photons: traditional multiplexed imaging relies on emission filters that selectively pass photons from one fluorophore at a time to minimize spectral overlap. As a consequence, a substantial fraction of emitted photons is discarded, and relaxing the filters leads to bleedthrough artifacts. MicroSplit changes this trade-off because multiple structures can be imaged in the same acquisition. This allows microscopists to use substantially broader emission filters, collecting photons from several fluorophores simultaneously, without introducing the ambiguities that would otherwise arise in a multichannel setting.

To roughly quantify this advantage, we analyzed three fluorophores from the HT-LIF24 dataset (DAPI, FITC and TRITC), using their emission spectra (downloaded from fpbase.org) normalized as probability mass functions. We compared the fraction of photons transmitted by conventional multicolor filter configurations to the photons captured when using a broad, highly permissive filter suitable for MicroSplit. Across three representative scenarios, with emission filter thresholds chosen to (1) maximize photon collection; (2) reduce bleedthrough by 25%; and (3) reduce bleedthrough by 50%, with conventional multiplexed imaging being respectively 22%, 34% and 55% fewer photons efficient than the results obtained using MicroSplit (Extended Data Fig. 7).

In practical terms, this means that even in bleedthrough-optimized multiplexed imaging, each channel discards a large fraction of emitted photons, whereas MicroSplit can reclaim much of this loss by aggregating photons from multiple fluorophores in a single measurement.

By enabling gentler imaging due to denoising: our method, next to performing semantic unmixing, also performs unsupervised denoising. As denoising improves the SNR of images it is applied to, microscopists can acquire the raw data more gentle, accepting a lower initial SNR[3]. To illustrate this on a concrete example, we assessed the similarity of a biological structure imaged at various exposure times between 2 ms and 20 ms with very high SNR data of the same regions of interest acquired at 500-ms exposure time. More concretely, we have conducted these experiments on the three-channel data (Nucleus, Microtubules and Kinetocore) of the HT-LIF24 dataset. As shown in Supplementary Table 11 (bottom), even the denoised 2-ms exposure micrographs lead to a considerably higher quality with regard to the 500-ms images than even the 5-ms raw data, for channel 1 even compared to the 20-ms raw acquisitions, suggesting at least a threefold to tenfold reduction of the required photon budget and therefore enabling users of MicroSplit to image considerably more gentle to reach the same quality required for downstream processing.

## Statistics and reproducibility

For both qualitative and quantitative evaluations, we generate 50 predictions per input using the trained MicroSplit model, running it 50 times. Predictions employ tiled inference with overlap: each test input frame is divided into overlapping $64 \times 64$ patches; the 50 predictions per patch are averaged, and the central $32 \times 32$ region is extracted. These regions from overlapping patches are then stitched together to form a full-frame prediction. The same approach applies to 3D models. Please refer to the supplementary material for more details. Thus, metrics in Table 1 (with standard errors in Supplementary Table 12) and qualitative results in Figs. 1d–f, 3a and 4 all use these 50-prediction averages.

## Reporting summary

Further information on research design is available in the Nature Portfolio Reporting Summary linked to this article.

## Data availability

The following enumerated list includes the URLs where each dataset can be downloaded individually. (1) HT-H24 https://download.fht.org/jug/msplit/ht_h24/data/ht_h24.zip. (2) HT-P32A https://download.fht.org/jug/msplit/ht_p23a/data/ht_p23a.zip. (3) HT-P23B https://download.fht.org/jug/msplit/ht_p23b/data/ht_p23b.zip. (4) Pavia-P24 https://download.fht.org/jug/msplit/pavia_p24/data/pavia_p24.zip. (5) HT-T24 https://download.fht.org/jug/msplit/ht_t24/data/ht_t24.zip. (6) HT-LIF24 (2 ms) https://download.fht.org/jug/msplit/ht_lif24/data/ht_lif24_2ms.zip. (7) HT-LIF24 (3 ms) https://download.fht.org/jug/msplit/ht_lif24/data/ht_lif24_3ms.zip. (8) HT-LIF24 (5 ms) https://download.fht.org/jug/msplit/ht_lif24/data/ht_lif24_5ms.zip. (9) HT-LIF24 (20 ms) https://download.fht.org/jug/msplit/ht_lif24/data/ht_lif24_20ms.zip. (10) HT-LIF24 (500 ms) https://download.fht.org/jug/msplit/ht_lif24/data/ht_lif24_500ms.zip. (11) Chicago-Sch23 https://download.fht.org/jug/msplit/chicago_sch23/data/chicago_sch23.zip. (12) HHMI-D25-8bit https://download.fht.org/jug/msplit/hhmi_d25/hhmid25_8bit.zip. (13) HHMI-D25-16-bit https://download.fht.org/jug/msplit/hhmi_d25/hhmi_d25_16bit.zip. (14) HHMI-D25-16-bit-0.25 https://download.fht.org/jug/msplit/hhmi_d25/hhmi_d25_16bit_binned.zip.

## Code availability

We have integrated MicroSplit's implementation into the Python package CAREamics, which is available at https://careamics.github.io/. User-friendly notebooks for training and evaluating MicroSplit using CAREamics can be found on our GitHub repository at https://github.com/CAREamics/MicroSplit-reproducibility. The original code used to generate the results is also accessible at https://github.com/juglab/MicroSplit.

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

## Acknowledgements

We thank T. Lambert, Lecturer on Systems Biology and Associate Director of Imaging Technology in the Core for Imaging Technology and Education for useful discussions and invaluable feedback. This work was supported by the European Union through the Horizon Europe program (IMAGINE project, grant agreement 101094250-IMAGINE and AI4LIFE project, grant agreement 101057970-AI4LIFE) as well as core funding of Fondazione Human Technopole. We thank the IT and HPC team at Human Technopole and the BMBF-funded de.NBI Cloud within the German Network for Bioinformatics Infrastructure (de.NBI) (031A532B, 031A533A, 031A533B, 031A534A, 031A535A, 031A537A, 031A537B, 031A537C, 031A537D and 031A538A) for access to their compute infrastructure. We are grateful to N. Kalebic (Human Technopole) for providing the ferret sample and to J. Helppi and his team (Biomedical Services of the MPI-CBG in Dresden). We thank S. Barozzi from the Imaging Technological Development Unit of the Istituto Fondazione di Oncologia Molecolare (IFOM, Milan, Italy) for the preparation of the samples used for the generation of the HT-LIF24 dataset. Finally, we thank Janelia and the Howard Hughes Medical Institute for their support.

## Author contributions

F.J. conceived and supervised the project, established collaborations, provided technical guidance throughout the study, supervised figure preparation, and rewrote the paper based on the original draft by A.A. A.A. developed the method, performed all semantic unmixing, uncertainty quantification and calibration, and puncta-removal experiments, generated all figures, drafted the paper and supplementary information, and coordinated data generation with contributing laboratories. F. Carrara, I.Z., V.G., M.C. and J.D. took code from A.A. code and adapted it into the CAREamics Python package, developed user-friendly training and evaluation notebooks, and ensured reproducibility; F. Carrara and I.Z. led this effort. E. Cammarota, J.M.B., J.D. and D.E.D.N. trained segmentation models and performed the analyses shown in Fig. 3; J.D. and D.E.D.N. designed the segmentation assessment strategy. D.E.D.N. additionally proposed the puncta-removal task. M.P., D.G. F. Casagrande, E. Colombo, S.G., E.R., N.K., M.D.S., R.A., D.F., G.P., E.T., O.H., N.M., N.S. and F.P. generated the datasets used in this study; F.P. and M.D.S. additionally contributed to early conceptual discussions.

## Competing interests

The authors declare no competing interests.

## Additional information

**Extended data** is available for this paper at https://doi.org/10.1038/s41592-026-03082-1.

**Correspondence and requests for materials** should be addressed to Florian Jug.

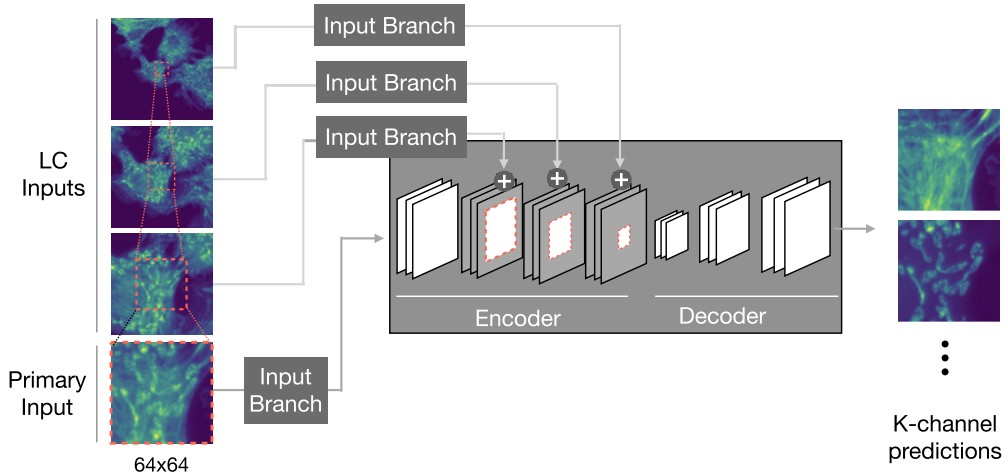

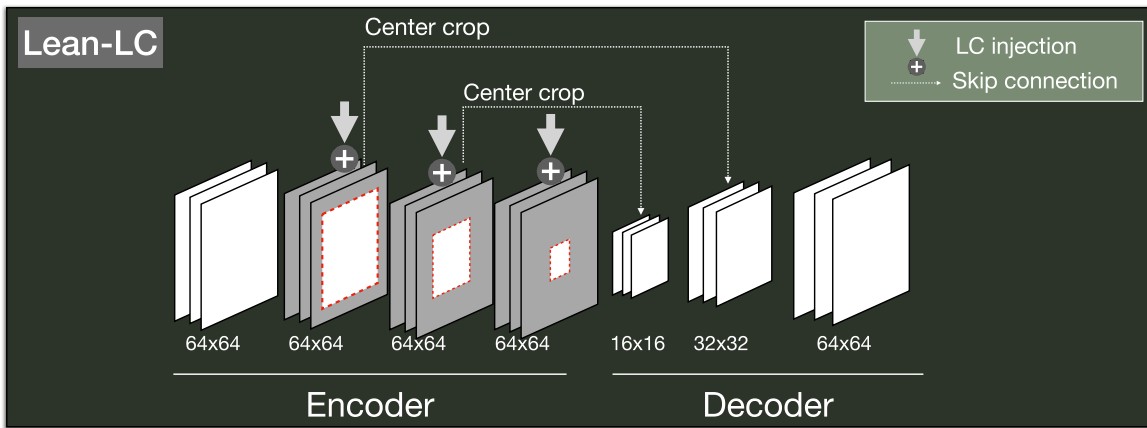

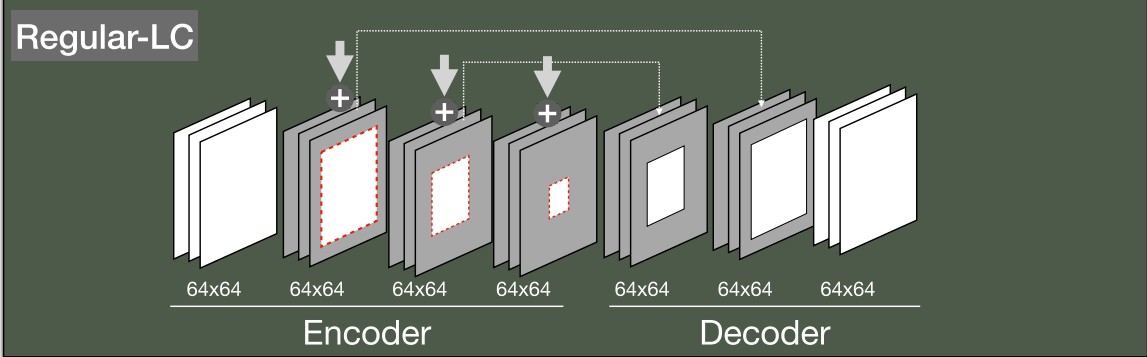

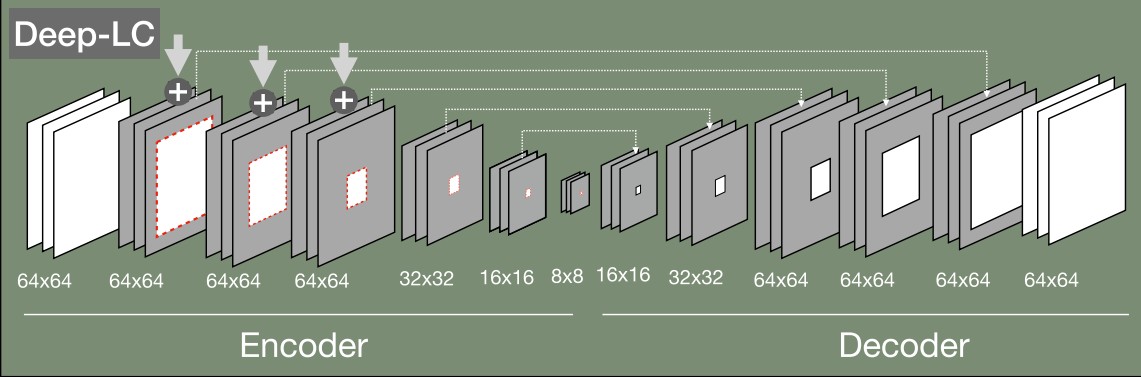

**Extended Data Fig. 1 | See next page for caption.**

**Extended Data Fig. 1 | Network architectures employed by MicroSplit.** This figure provides a detailed breakdown of the components shown in Figure 2a. Specifically, it illustrates the encoder-decoder structure of our architecture, which is adapted from our prior work, μSplit [6]. Since this diagram focuses on the transformation of input data into predictions, it does not include the loss terms (such as the KL divergence loss and the noise-model-based likelihood loss). As in [6], we improve the predictive capacity of MicroSplit by feeding not only an image patch (primary input), but also additional image context. To this end, we introduced lateral contextualization (LC). These LC inputs capture a larger portion of the input data but at increasingly lower pixel-resolution. This lead to three meta-architectures, Lean-LC, Regular-LC, and Deep-LC, each with increasingly higher computational and GPU memory demands but also leading to increasingly higher unmixing performance. The gray region shown in the embeddings in all three meta-architectures represents the extra spatial size in the embeddings coming solely due to LC inputs. In Lean-LC, embeddings of different hierarchy levels in the encoder benefit from LC inputs, but the decoder does not. In Regular-LC, a more GPU-consuming variation, both the encoder's and the decoder's embeddings use LC (that is show gray regions in the figure). Finally, since in Regular-LC the spatial dimensions of the embedding layers do not decrease, it enabled us to stack some additional hierarchy levels on top of the previously used ones. We labeled the resulting architecture as Deep-LC, the most performant but also most resource heavy variation. In order to get best results, we have used Deep-LC whenever possible, but it is important to note that with just two simple hyper-parameter switches, one can instantly make use of Regular-LC or Lean-LC and train on cheaper and older consumer GPUs.

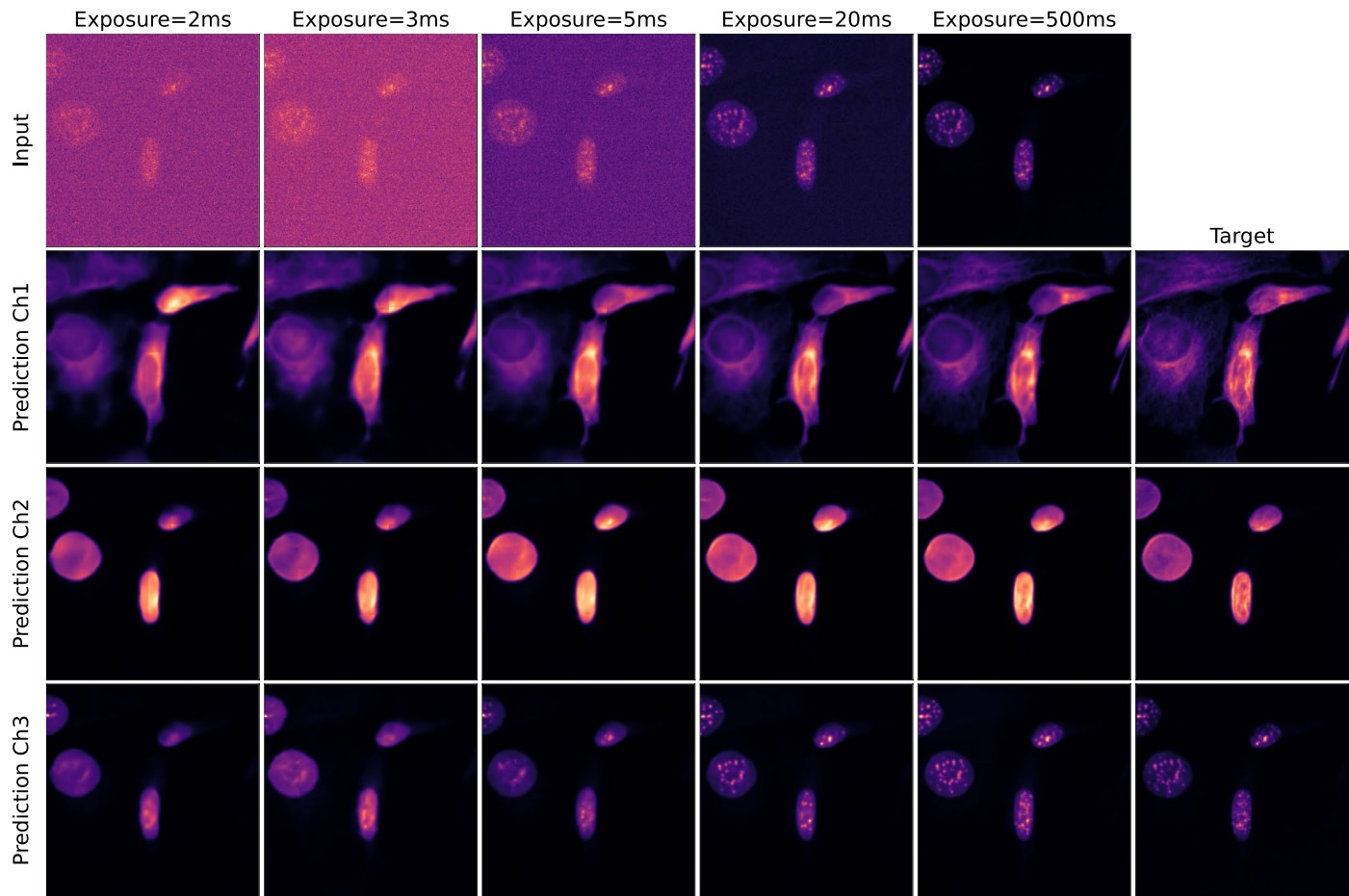

**Extended Data Fig. 2 | Effect of SNR on Model Performance (HT-LIF24 Dataset).** We evaluate how SNR influences model performance using the HT-LIF24 dataset. Different models are trained on data subsets acquired with varying exposure durations-leading to different SNRs-and their predictions are compared over a common region of interest. High-frequency details in the predictions (especially the third channel) are visibly reduced when the input SNR is lower.

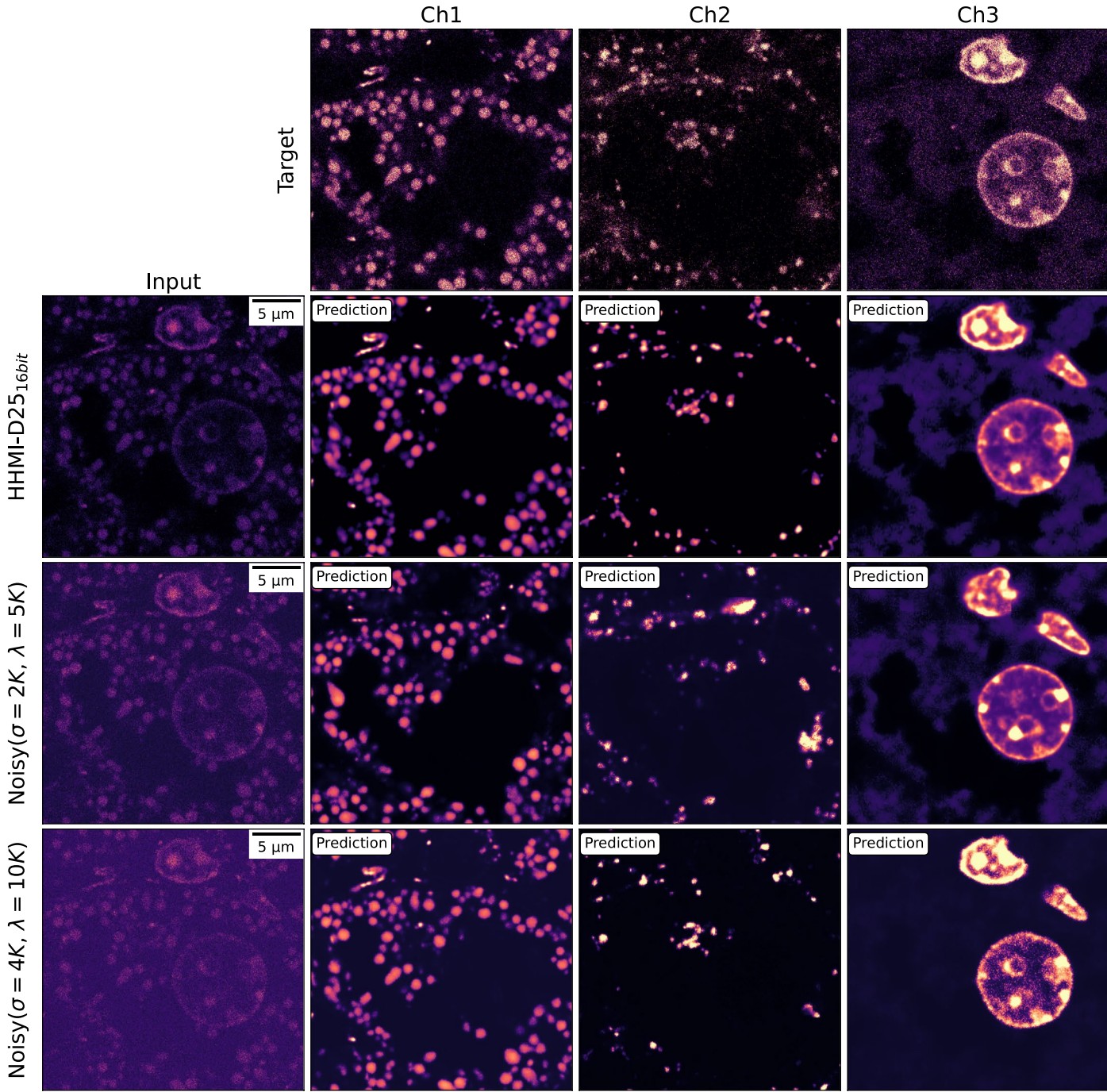

**Extended Data Fig. 3 | Effect of SNR on Model Performance (HHMI-D25$_{16bit}$ Dataset).** We assess the impact of SNR on the HHMI-D2516bit dataset. For this part of the HHMI-D25 data, the predictions (row 2, Task XXXIII) are visually more close to the target images compared to results on HHMI-D258bit dataset (Figure S61, row 2, Task XXIII). We then introduce two levels of additional Gaussian and Poisson noise to the HHMI-D2516bit data to reduce SNR and retrain MicroSplit on those noisier versions of the data (row 3 and 4, Tasks XXXIV and XXXV, respectively). As it was likely to be expected, the reduced SNR leads to a noticeable decline in the semantic unmixing performance.

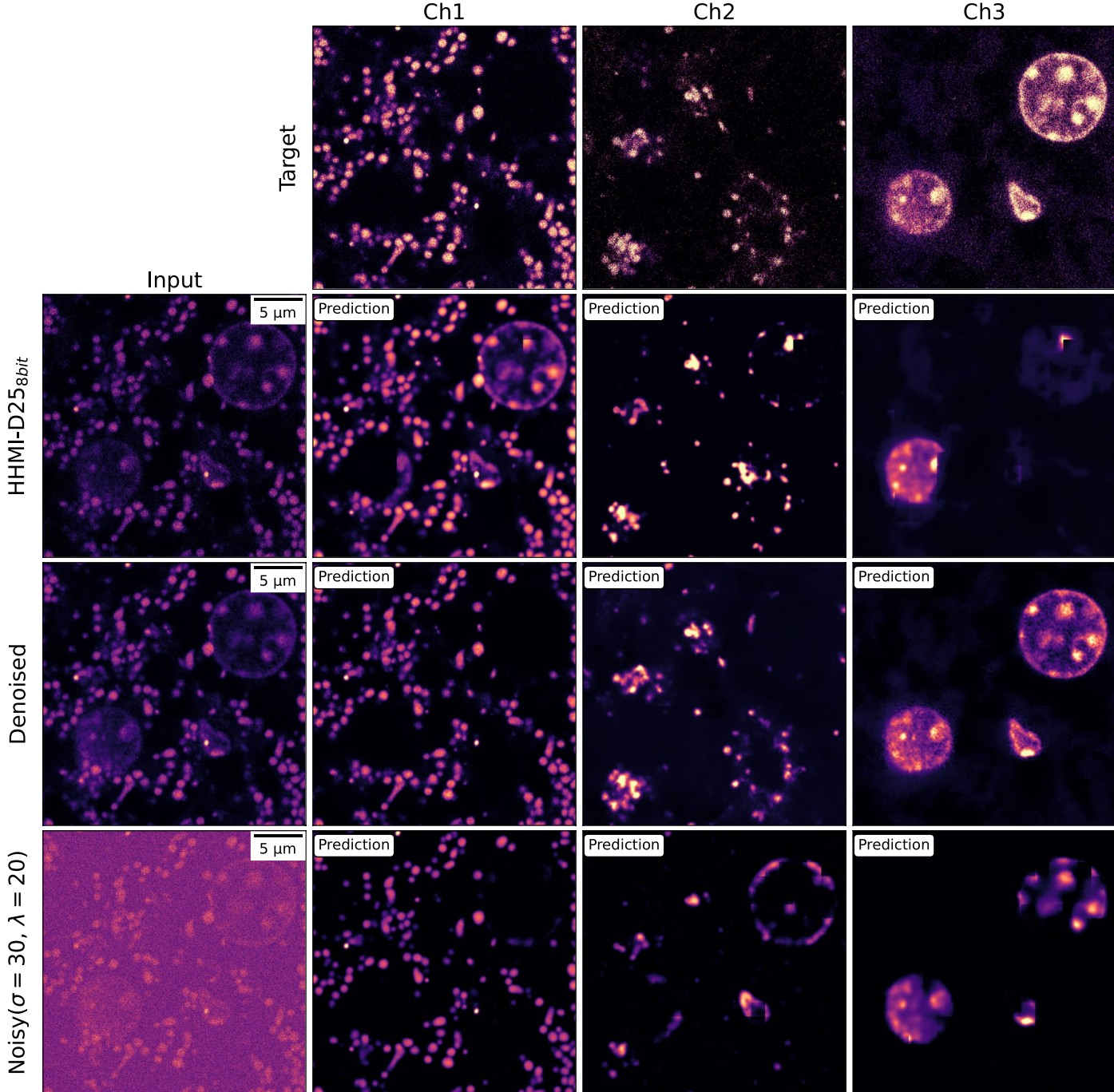

**Extended Data Fig. 4 | Effect of SNR on Model Performance (HHMI-D258bit Dataset).** We assess the impact of SNR on a subset of the HHMI-D25 dataset. Comparing the predictions (row 2, results of Task XXIII) with the ground truth (row 1), we observe that the prediction quality, particularly for the third channel, is not good, with entire parts of the structures being put into the other channels. We then train MicroSplit using a Noise2Void [4] denoised version of the same data, leading to much improved semantic unmixing performance (row 3, Task XXXI). Finally, we re-introduced synthetic Gaussian and Poisson noise to the denoised HHMI-D25 data used in Task XXXI and retrained MicroSplit on this lower-SNR data (row 4, Task XXXII). As it was likely to be expected, this does again drop the semantic unmixing performance.

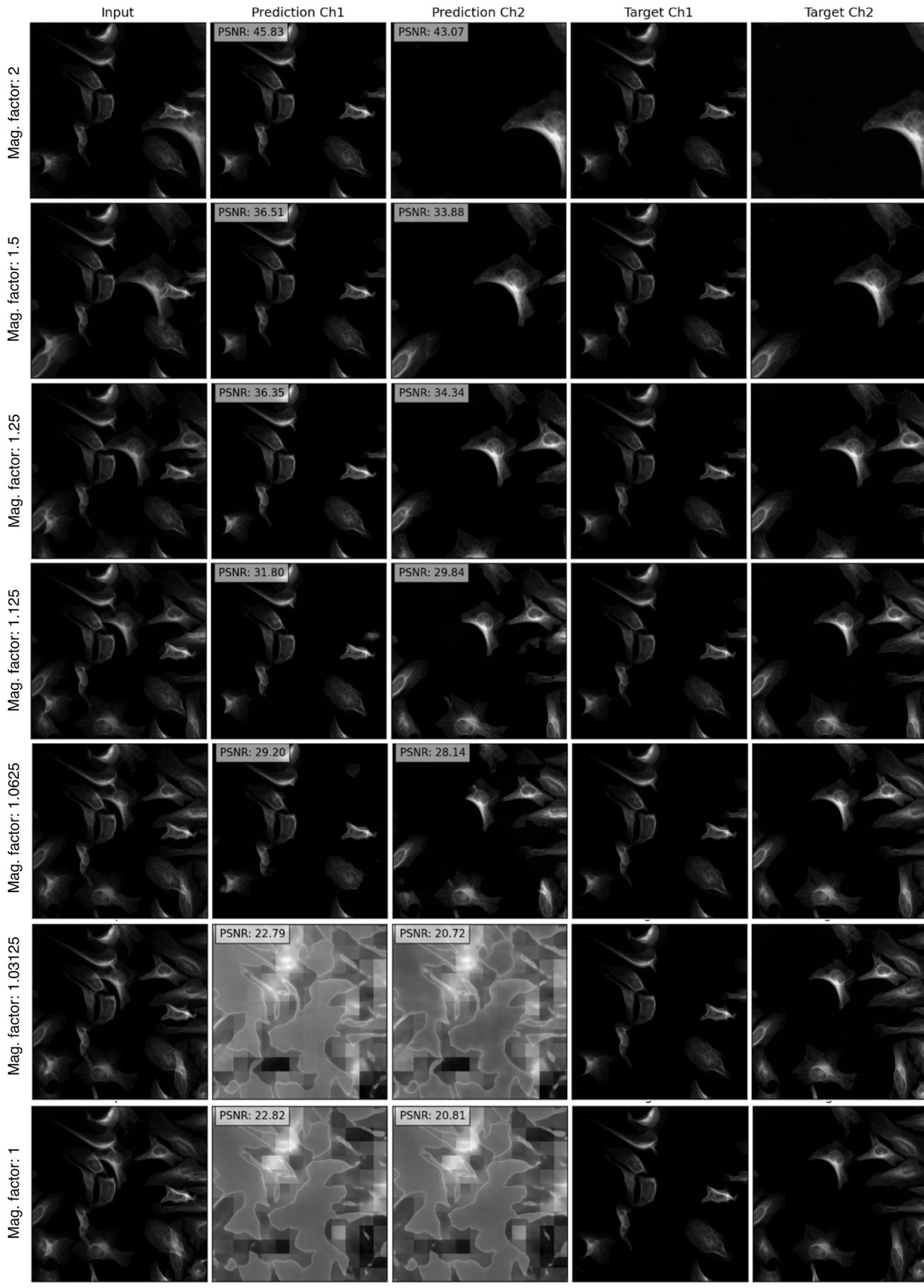

**Extended Data Fig. 5 | Semantic unmixing of similar structures with MicroSplit.** We show how the same structure in two target channels can still be unmixed, even if the only structural difference is a controllable scaling factor. See Supplementary Section B.3 for a detailed description of the conducted experiments.

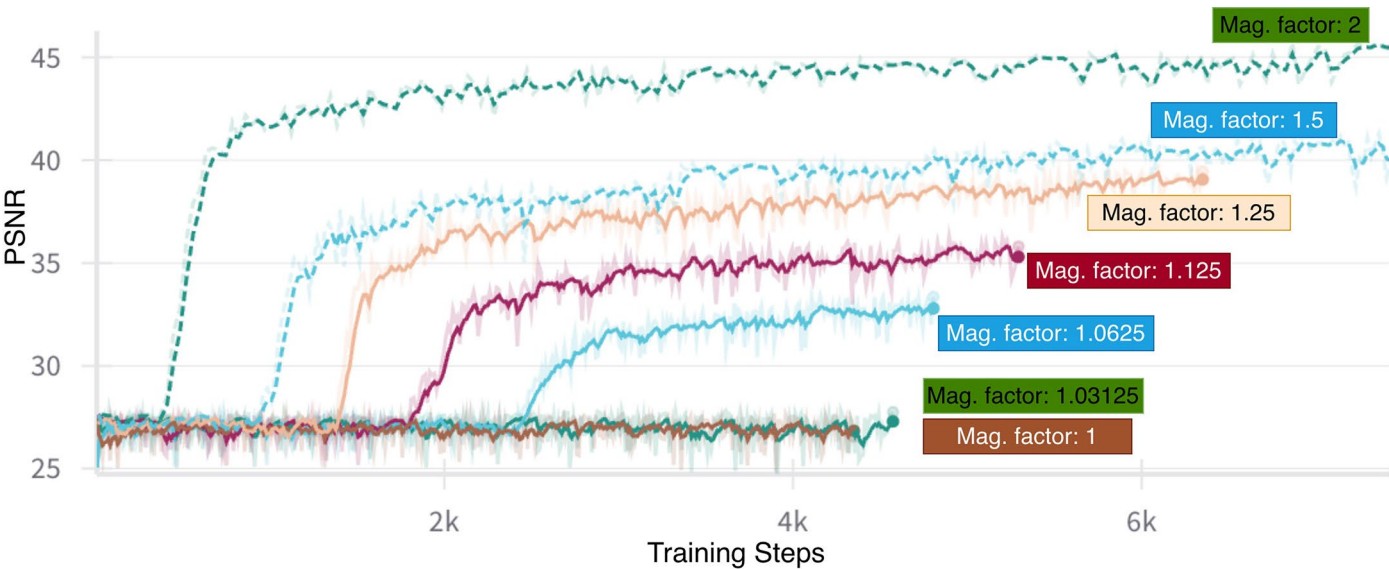

**Extended Data Fig. 6 | Semantic unmixing of similar structures with MicroSplit.** Here we plot Validation PSNR vs. training steps for all experiments shown in Figure S2. Observe that, as the magnification factor gets closer to 1, it takes longer for the network to initiate the learning, and the quality of the splitting converges to an overall lower level. See Supplementary Section B.3 for a detailed description of the conducted experiments.

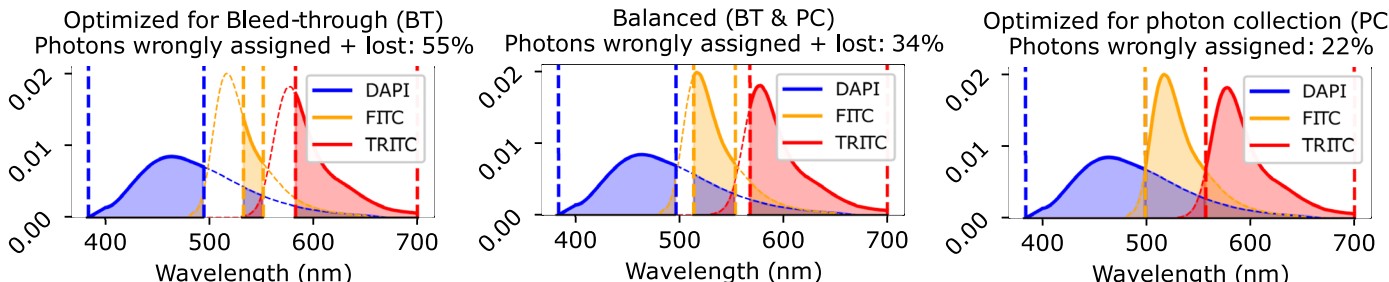

**Extended Data Fig. 7 | Photon efficient imaging with MicroSplit: Reduced photon filtering.** Emission spectra of the three fluorophores DAPI, FITC, and TRITC are shown together with example emission filter bands used in conventional multi-color imaging. From left to right, we illustrate three settings that increasingly trade-off bleedthrough (BT) against photon efficiency: a configuration that strongly suppresses BT, a balanced configuration that collects more photons at the cost of some BT, and a highly permissive configuration that captures nearly all emitted photons but suffers notable spectral overlap. MicroSplit, by contrast, allows imaging multiple structures within a single broad emission band and subsequently reassigning the collected intensities to their respective output channels leading to the indicated photon-efficiency increases.

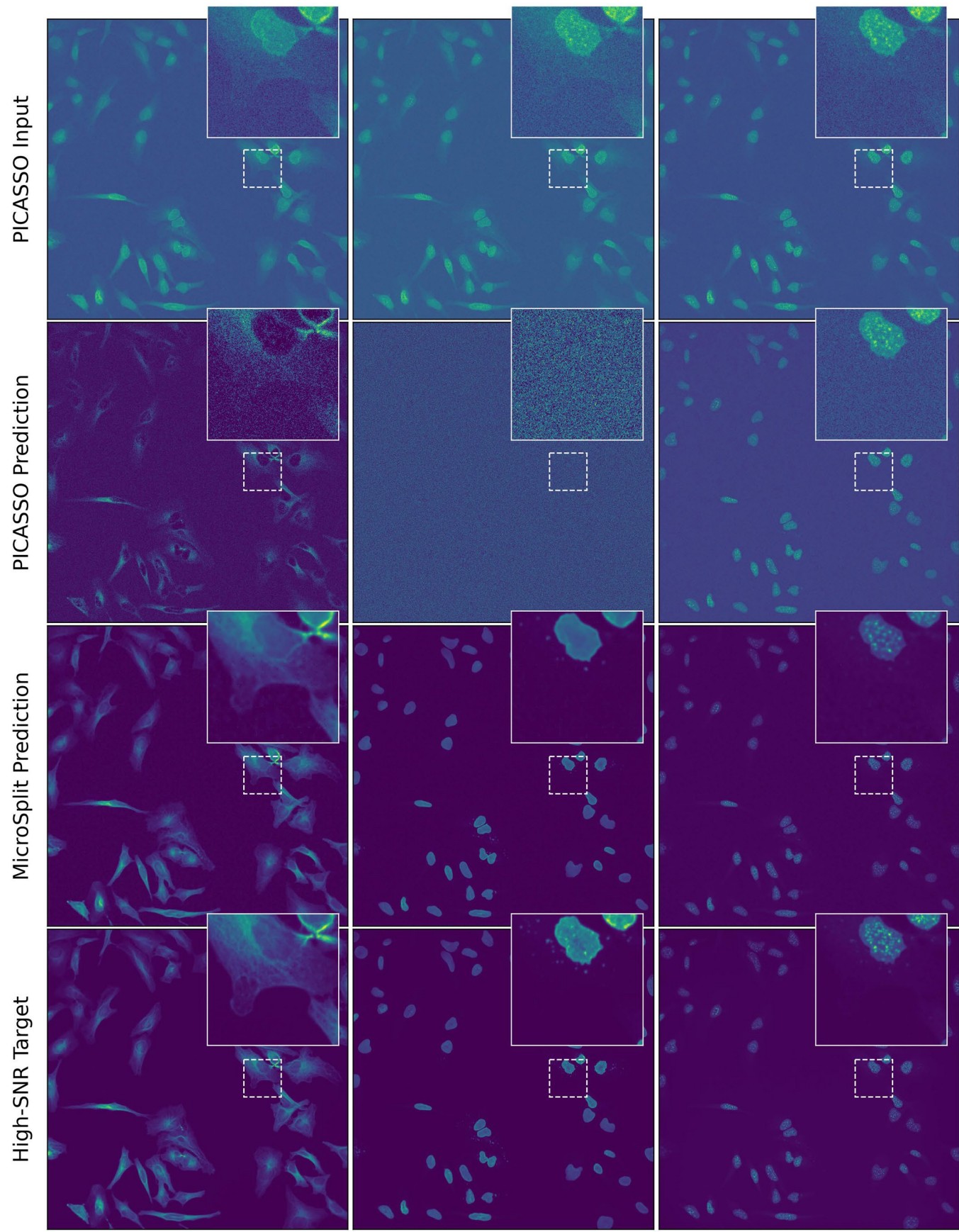

**Extended Data Fig. 8 | See next page for caption.**

**Extended Data Fig. 8 | Comparison with PICASSO.** Comparison with PICASSO on a 3-channel unmixing task. Please note that MicroSplit distinguishes itself from PICASSO [20] by only requiring a single superimposed image as input, rather than multiple images with different spectral overlaps. Here, we used the high-SNR data from the HT-LIF24 dataset, prepared inputs that are suitable for PICASSO, and added additional (synthetic) noise to demonstrate how different both systems deal with noisy data. In row 1, we show example images from the three input channels used for PICASSO. In rows 2 and 3, we show prediction by PICASSO and MicroSplit, respectively. In the last row, we show the high-SNR ground truth for visual evaluation. We observe that, in the presence of noise, Picasso starts to be challenged to unmix the data, mostly if it starts having similar features (see columns 2 and 3, where PICASSO removes data around the nucleus region seen in column 1.

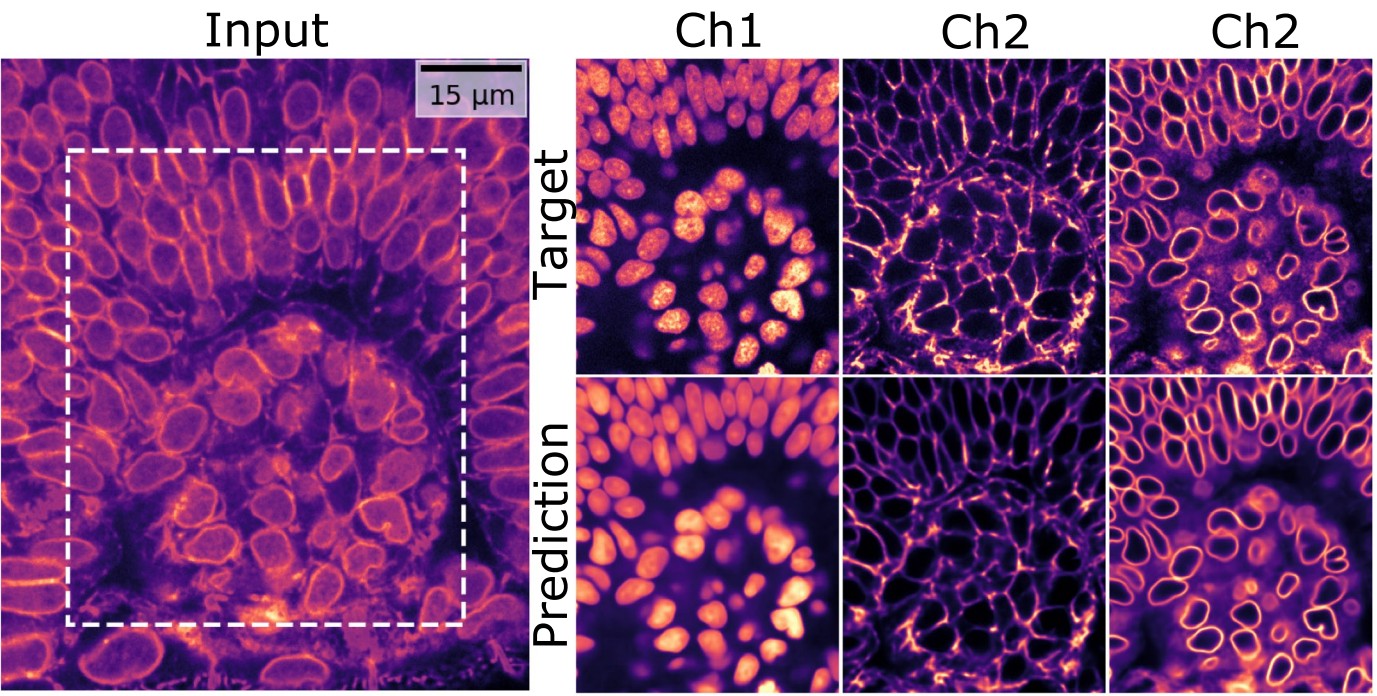

**Extended Data Fig. 9 | Qualitative Evaluation.** Qualitative Evaluation for Task XXI from CBG-Z18 dataset. Note that we show the target and the prediction corresponding to the input crop which is denoted in Input panel by a white dotted rectangle.

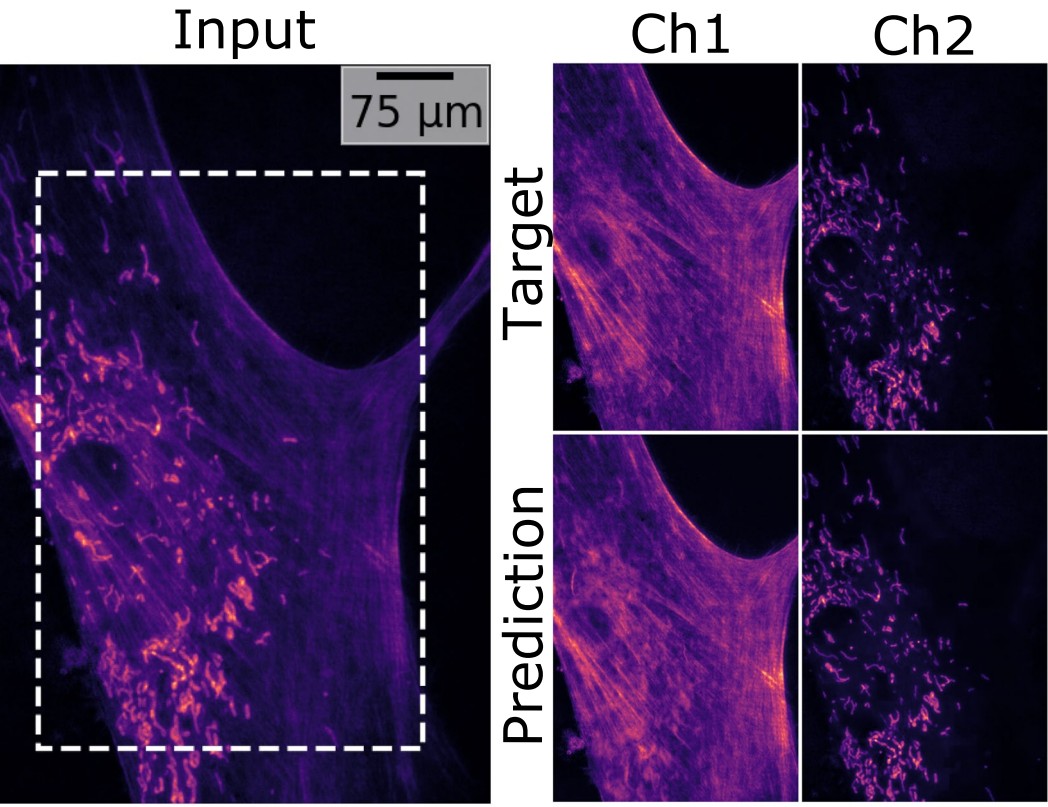

**Extended Data Fig. 10 | Qualitative Evaluation.** Qualitative Evaluation for Task XV from Chicago-Sch23 dataset. Note that we show the target and the prediction corresponding to the input crop which is denoted in Input panel by a white dotted rectangle.

# Reporting Summary

## Statistics

For all statistical analyses, confirm that the following items are present in the figure legend, table legend, main text, or Methods section.

| n/a | Confirmed | |
|---|---|---|
| ☐ | ☒ | The exact sample size (*n*) for each experimental group/condition, given as a discrete number and unit of measurement |
| ☐ | ☒ | A statement on whether measurements were taken from distinct samples or whether the same sample was measured repeatedly |
| ☐ | ☒ | The statistical test(s) used AND whether they are one- or two-sided<br>*Only common tests should be described solely by name; describe more complex techniques in the Methods section.* |
| ☒ | ☐ | A description of all covariates tested |
| ☐ | ☒ | A description of any assumptions or corrections, such as tests of normality and adjustment for multiple comparisons |
| ☐ | ☒ | A full description of the statistical parameters including central tendency (e.g. means) or other basic estimates (e.g. regression coefficient) AND variation (e.g. standard deviation) or associated estimates of uncertainty (e.g. confidence intervals) |
| ☒ | ☐ | For null hypothesis testing, the test statistic (e.g. $F$, $t$, $r$) with confidence intervals, effect sizes, degrees of freedom and $P$ value noted<br>*Give P values as exact values whenever suitable.* |
| ☒ | ☐ | For Bayesian analysis, information on the choice of priors and Markov chain Monte Carlo settings |
| ☒ | ☐ | For hierarchical and complex designs, identification of the appropriate level for tests and full reporting of outcomes |
| ☒ | ☐ | Estimates of effect sizes (e.g. Cohen's *d*, Pearson's *r*), indicating how they were calculated |

*Our web collection on statistics for biologists contains articles on many of the points above.*

## Software and code

Policy information about availability of computer code

| Data collection | Most of the data was acquired for this work, a few training tasks are derived from the CARE (NatMeth 2018) work. We have listed all datasets and linked to them here: https://github.com/CAREamics/MicroSplit-reproducibility?tab=readme-ov-file#links-to-all-datasets-used-in-the-manuscript |
|---|---|
| Data analysis | Results reproducibility notebooks: https://github.com/CAREamics/MicroSplit-reproducibility<br>CAREamics library (fully open): https://careamics.github.io/0.1/ |

For manuscripts utilizing custom algorithms or software that are central to the research but not yet described in published literature, software must be made available to editors and reviewers. We strongly encourage code deposition in a community repository (e.g. GitHub). See the Nature Portfolio guidelines for submitting code & software for further information.

## Data

Policy information about <u>availability of data</u>

All manuscripts must include a <u>data availability statement</u>. This statement should provide the following information, where applicable:

- Accession codes, unique identifiers, or web links for publicly available datasets
- A description of any restrictions on data availability
- For clinical datasets or third party data, please ensure that the statement adheres to our <u>policy</u>

All data used in the manuscript is public and can be found here: https://github.com/CAREamics/MicroSplit-reproducibility?tab=readme-ov-file#links-to-all-datasets-used-in-the-manuscript

## Research involving human participants, their data, or biological material

Policy information about studies with <u>human participants or human data</u>. See also policy information about <u>sex, gender (identity/presentation), and sexual orientation</u> and <u>race, ethnicity and racism</u>.

| Reporting on sex and gender | No human data was used in this work. |
|---|---|
| Reporting on race, ethnicity, or other socially relevant groupings | No human data was used in this work. |
| Population characteristics | No human data was used in this work. |
| Recruitment | No human data was used in this work. |
| Ethics oversight | No human data was used in this work. |

Note that full information on the approval of the study protocol must also be provided in the manuscript.

# Field-specific reporting

Please select the one below that is the best fit for your research. If you are not sure, read the appropriate sections before making your selection.

☒ Life sciences          ☐ Behavioural & social sciences          ☐ Ecological, evolutionary & environmental sciences

For a reference copy of the document with all sections, see <u>nature.com/documents/nr-reporting-summary-flat.pdf</u>

# Life sciences study design

All studies must disclose on these points even when the disclosure is negative.

| Sample size | Does not apply. We used diverse datasets to convince readers that our method generally applicable. |
|---|---|
| Data exclusions | No data was excluded. |
| Replication | All results have been reproduced multiple times and can be reproduced using our open datasets and open source code. |
| Randomization | I think this does not apply. We used in our data loaders random sampling of ROIs being feed to the network during network training, but I doubt this is the randomization this form is asking about. |
| Blinding | Does not apply. |

# Reporting for specific materials, systems and methods

We require information from authors about some types of materials, experimental systems and methods used in many studies. Here, indicate whether each material, system or method listed is relevant to your study. If you are not sure if a list item applies to your research, read the appropriate section before selecting a response.

## Materials & experimental systems

| n/a | Involved in the study |
|-----|----------------------|
| ☐ | ☒ Antibodies |
| ☐ | ☒ Eukaryotic cell lines |
| ☒ | ☐ Palaeontology and archaeology |
| ☐ | ☒ Animals and other organisms |
| ☒ | ☐ Clinical data |
| ☒ | ☐ Dual use research of concern |
| ☒ | ☐ Plants |

## Methods

| n/a | Involved in the study |
|-----|----------------------|
| ☒ | ☐ ChIP-seq |
| ☒ | ☐ Flow cytometry |
| ☒ | ☐ MRI-based neuroimaging |

# Antibodies

| | |
|---|---|
| Antibodies used | 1. Puncta (HT-H23): (does not name it, but says "a secondary antibody conjugated to a 555 dye")<br>2. Pigino lab: anti alpha-tubulin (ABCD antibodies AA_345), anti-beta tubulin (AA_344) Mouse IgG2a raised antibody, Guinea Pig IgG raised antibody, Goat Anti-Mouse IgG Secondary Antibody, donkey anti-Rabbit IgG(H+L) Highly Cross-Adsorbed Secondary Antibody.<br>3. HT-T24: S*OX2 (goat polyclonal, AF2018, 1:200, R&D Systems) and GRASP65 (rabbit polyclonal, PA3-910, 1:200, Invitrogen);* donkey anti-Goat IgG(H+L) Highly Cross-Adsorbed Secondary Antibody, Alexa Fluor Plus 555 (A32816, 1:500, Invitrogen) and donkey anti-Rabbit IgG(H+L) Highly Cross-Adsorbed Secondary Antibody, Alexa Fluor Plus 647 (A32795, 1:500, Invitrogen).<br>4. Feliciano lab: Mouse anti-PMP70 (MilliporeSigma, SAB4200181, 1:75), rabbit anti-LAMP1 (Abcam, AB208943, 1:50), Alexa Fluor 647 goat anti-mouse antibody (Thermo Fisher, A21235, 1:500), Alexa Fluor 750 goat anti-rabbit antibody (Thermo Fisher, A21039, 1:500)<br>5. HT-LIF24: anti-α-tubulin mouse IgG monoclonal primary antibody (T5168, Sigma-Aldrich; 1:100); anti-laminin B1 rabbit IgG polyclonal primary antibody (ab16048, Abcam; 1:200); anticentromere protein human IgG polyclonal primary antibody (15-234, Antibodies Incorporated; 1:400); Alexa Fluor 488 donkey anti-mouse IgG secondary antibody (A-21202, Thermo Fisher Scientific; 1:400); Cy3 donkey anti-rabbit IgG secondary antibody (711-165-152, Jackson Immunoresearch; 1:400); Cy5 goat anti-human IgG secondary antibody (109-175-088, Jackson Immunoresearch; 1:50).<br>6. HT_H23 dataset. Primary antibody = Rat α CTIP2 1:500 (1:500, Abcam, ab18465). Secondary antibody = Donkey α Rat Alexa Fluor™ Plus 555 (1:1000, Invitrogen, A48270)<br>7. HT_H24 dataset: Primary antibodies = Rat α SOX2 (1:200, Invitrogen, 14-9811-82), Chicken α MAP2 (1:5000, Invitrogen, PA1-10005). Secondary antibodies = Donkey α Chicken Alexa Fluor 488 (1:1000, Jackson ImmunoResearch, 703-545-155), Donkey α Rat Alexa Fluor™ Plus 555 (1:1000, Invitrogen, A48270). |
| Validation | No validation. For this work, this was not relevant. |

# Eukaryotic cell lines

Policy information about cell lines and Sex and Gender in Research

| | |
|---|---|
| Cell line source(s) | HT_P23A and HT_P23B datasets from Pigino lab: MDCK-II ccll line<br><br>HT-LIF24 dataset: HeLa cell line (tested negative for Mycoplasma), no authentication needed.<br><br>HT_H23 dataset: WA-09 human pluripotent stem cells (hPSCs)<br><br>HT_H24 dataset: WTC-11 (UCSFi001-A) induced pluripotent stem cells (iPSCs)<br><br>Pavia-P23 dataset: HaCaT cell line<br><br>Chicago-Sch23 dataset: Human BJ fibroblast cell line |
| Authentication | *Describe the authentication procedures for each cell line used OR declare that none of the cell lines used were authenticated.* |
| Mycoplasma contamination | HT_P23A, HT_P23B from Pigino lab: MDCK-II: mycoplasma free certification, regular testing.<br><br>HT-LIF24: HeLa cell line, tested negative for Mycoplasma.<br><br>Pavia-P23: tested for mycoplasma and bacterial contamination twice a year. |
| Commonly misidentified lines<br>(See ICLAC register) | *Name any commonly misidentified cell lines used in the study and provide a rationale for their use.* |

# Animals and other research organisms

Policy information about studies involving animals; ARRIVE guidelines recommended for reporting animal research, and Sex and Gender in Research

| | |
|---|---|
| Laboratory animals | HT-T24 (Ferret (Mustela furo)) |

HHMI-D25, Feliciano Lab: Heterozygous PhamExcised females carrying the Mito_Dendra2 transgene obtained by crossing PhamExcised males (strain #018397 derived homozygous males) from Jackson Laboratories and wild type C57BL/6J females

Wild animals

*Provide details on animals observed in or captured in the field; report species and age where possible. Describe how animals were caught and transported and what happened to captive animals after the study (if killed, explain why and describe method; if released, say where and when) OR state that the study did not involve wild animals.*

Reporting on sex

HT-T24 (A pregnant female was sacrificed to take the embryos whose sex was unknown, because they were at an early stage to determine it.)

HHMI-D25: In this study, we developed Microsplit, a computational multiplexing technique
based on deep learning that allows multiple cellular structures to be imaged in a single fluorescent channel and then unmix them by computational means. Although gender was not the focus of this study and not a critical factor, HHMI-D25 data sets were acquired from female mice due to their availability at the time of experiment.

Field-collected samples

*For laboratory work with field-collected samples, describe all relevant parameters such as housing, maintenance, temperature, photoperiod and end-of-experiment protocol OR state that the study did not involve samples collected from the field.*

Ethics oversight

HT-T24: Animals used for this study were kept in standardized hygienic conditions at the Biomedical Services Facility (BMS) of the MPI-CBG with free access to food and water. All experimental procedures were conducted in agreement with the German Animal Welfare Legislation after approval by the Landesdirektion Sachsen (license for ferret TVV2/2015

HHMI-D25, Feliciano Lab: All animal care and procedures were conducted according to NIH guidelines and were approved by the Institutional Animal Care and Use Committee (Protocol #22-0229.04) at Janelia Research Campus, Howard Hughes Medical Institute.

Note that full information on the approval of the study protocol must also be provided in the manuscript.

# Plants

Seed stocks

*Report on the source of all seed stocks or other plant material used. If applicable, state the seed stock centre and catalogue number. If plant specimens were collected from the field, describe the collection location, date and sampling procedures.*

Novel plant genotypes

*Describe the methods by which all novel plant genotypes were produced. This includes those generated by transgenic approaches, gene editing, chemical/radiation-based mutagenesis and hybridization. For transgenic lines, describe the transformation method, the number of independent lines analyzed and the generation upon which experiments were performed. For gene-edited lines, describe the editor used, the endogenous sequence targeted for editing, the targeting guide RNA sequence (if applicable) and how the editor was applied.*

Authentication

*Describe any authentication procedures for each seed stock used or novel genotype generated. Describe any experiments used to assess the effect of a mutation and, where applicable, how potential secondary effects (e.g. second site T-DNA insertions, mosiacism, off-target gene editing) were examined.*

