## [Peer Review File · Nature Methods]

MicroSplit: Semantic Unmixing of Fluorescent Microscopy Data

Corresponding Author: Dr Florian Jug

Version 0:

Decision Letter:

10th Apr 2025

Dear Florian,

I hope all is well!

Your Article, "MicroSplit: Semantic Unmixing of Fluorescent Microscopy Data", has now been seen by three reviewers. As you will see from their comments below, although the reviewers find your work of considerable potential interest, they have raised a number of concerns. We are interested in the possibility of publishing your paper in Nature Methods, but would like to consider your response to these concerns before we reach a final decision on publication. We therefore invite you to revise your manuscript to address these concerns.

We think the comments are generally constructive and that addressing them will improve the paper. We ask that you focus on addressing the technical concerns of the reviewers and would like to push you to make a stronger experimental case that your approach enables longer and/or gentler imaging. We think you should discuss as limitations (1) the achievable degree of multiplexing and (2) the fact that the separated images do not preserve intensity relationships. Please improve code documentation/user guidance so that potential users can readily try the approach on their own data.

We are committed to providing a fair and constructive peer-review process. Do not hesitate to contact us if there are specific requests from the reviewers that you believe are technically impossible or unlikely to yield a meaningful outcome. I am always happy to discuss your revision plans.

Link Redacted

We hope to receive your revised paper within three months. If you cannot send it within this time, please let us know. In this event, we will still be happy to reconsider your paper at a later date so long as nothing similar has been accepted for publication at Nature Methods or published elsewhere.

OPEN SCIENCE REQUIREMENTS

REPORTING SUMMARY AND EDITORIAL POLICY CHECKLISTS

IMAGE INTEGRITY

EXTENDED DATA FIGURES

DATA AVAILABILITY

All novel DNA and RNA sequencing data, protein sequences, genetic polymorphisms, linked genotype and phenotype data, gene expression data, macromolecular structures, and proteomics data must be deposited in a publicly accessible database, and accession codes and associated hyperlinks must be provided in the "Data Availability" section.

Please include a "Data availability" subsection in the Online Methods. This section should inform readers about the availability

of the data used to support the conclusions of your study, including accession codes to public repositories, references to source data that may be published alongside the paper, unique identifiers such as URLs to data repository entries, or data set DOIs, and any other statement about data availability. At a minimum, you should include the following statement: "The data that support the findings of this study are available from the corresponding author upon request", describing which data is available upon request and mentioning any restrictions on availability. If DOIs are provided, please include these in the Reference list (authors, title, publisher (repository name), identifier, year). For more guidance on how to write this section please see: <http://www.nature.com/authors/policies/data/data-availability-statements-data-citations.pdf>

CODE AVAILABILITY

Please include a "Code Availability" subsection in the Online Methods which details how your custom code is made available. Only in rare cases (where code is not central to the main conclusions of the paper) is the statement "available upon request" allowed (and reasons should be specified).

ORCID

Sincerely,
Rita

Rita Strack, Ph.D.
Senior Editor
Nature Methods

Reviewers' Comments:

Reviewer #1 (Remarks to the Author):

In the manuscript "MicroSplit: Semantic Unmixing of Fluorescent Microscopy Data" from Ashesh et al, the authors describe a computational approach aimed at overcoming the limitations of fluorescence multiplexing by predicting multiple fluorescently labeled structures from a reduced number of imaging channels. The proposed method seeks to address issues related to overlapping fluorophore excitation spectra, reducing the number of required exposures in multiplex acquisitions, and overcoming the limits of maximum sampling frequency in live cell imaging scenarios. Their approach builds upon two of their previous publications and introduces a volumetric image processing framework that enables the calibration of the network using the data itself while estimating prediction errors.

The claimed novelty:

1. Combining elements from two prior computational approaches (μ Split and denoiSplit) into a single framework, leveraging machine learning to address fluorescence multiplexing challenges.
2. Processing volumetric image data using a highly optimized network architecture designed specifically for 3D fluorescence imaging.
3. Introducing a self-calibration procedure that assesses the quality of the trained network and estimates prediction errors, enhancing reliability in fluorescence reconstruction.

Claim 1 is only referenced in the manuscript, not supported by any practical demonstration, code is made available for readers to review.

Claim 2 is addressed by mathematical demonstrations in supporting material but not associated with any explicit timing of analysis of 3d fluorescence imaging dataset.

Claim 3 is presented with multiple datasets and plots across the manuscript, with some supporting material.

The manuscript, in its current version, is framed to present an alternative to standard unmixing techniques (i.e. spectral), arguing that MicroSplit achieves efficient reconstruction of fluorescent structures through learned feature extraction rather than intensity-based spectral decomposition. The authors suggest that this enables more photon-efficient multiplex imaging, reducing phototoxicity and improving imaging speed.

The proposed method is designed to address these fluorescence imaging challenges, however, similar problems have been explored in similar studies and/or commercially, using alternative approaches:

1. Machine learning-based fluorescence prediction in both 2D and 3D, such as the work from Guo et al (DOI: 10.7554/eLife.55502) , that of Ounkomol et al (DOI: 10.1038/s41592-018-0111-2) , Wang et al (DOI: 10.1038/s41592-018-0239-0) , Christiansen et al (DOI: 10.1016/j.cell.2018.03.040) to name some, though with recent machine learning improvements the count of applications of machine learning articles focusing on fluorescence prediction is substantially larger
2. Commercial systems incorporating spectral unmixing methods are already widely used in fluorescence imaging and can be found in commonly available instruments from Zeiss, Leica, Nikon and Olympus.

A key difference between MicroSplit and these prior methods is that the latter approaches often leverage real intensity distributions to calibrate predictions, whereas MicroSplit primarily focuses on structural reconstruction without maintaining intensity fidelity.

If the claims stated by the authors are confirmed, the work may represent an advance in the application of machine learning to fluorescence imaging. However, fundamental concerns remain regarding the practical advantages of this method.

The study lacks strong experimental validation to support its claims, particularly regarding photon efficiency and phototoxicity reduction. The multiplexing scenario presented is relatively simple (3-4 channels, 3-4 labels), limiting its applicability to more complex imaging tasks. Additionally, the dataset size is small (24 datasets), raising concerns about overfitting and generalizability compared to larger-scale studies. The terminology surrounding “unmixing” is inconsistent, as MicroSplit does not perform true spectral unmixing but rather predicts structures without preserving intensity relationships. The presentation is highly technical, requiring extensive cross-referencing of prior work and supplementary materials, making it difficult to follow. There are also concerns with performance validation, as reported DICE scores do not align well with visual assessments, and potential blurring artifacts may compromise downstream image analysis. While the method may have potential in fluorescence imaging, its motivation and practical advantages over existing techniques remain unclear.

I ask the authors to address the following concerns.

Major comments

1. The primary motivation of this work is to overcome challenges in fluorescence multiplexing, reducing the number of required exposures in multiplex acquisitions, and overcoming the limits of maximum sampling frequency in live cell imaging scenarios.
 - a. In the definition used in this manuscript, a three-channel fluorescence microscopy dataset is considered multiplexing. Three-channel/three-labels fluorescence is a rather simplistic multiplexing scenario. The authors should clarify why such simple scenario is considered here, as opposed to some more complex scenario (10-15 labels).
 - b. The work here presented appears to require triple labeling with different fluorophores. The authors should clarify how this triple-labeling approach is advantageous compared to the approaches for machine learning-based fluorescence prediction referenced above (e.g. DOI: 10.7554/eLife.55502)
 - c. The authors should demonstrate how this approach overcomes the limits of maximum sampling frequency in live cell imaging scenarios while reducing photo-bleaching. Detecting fluorescence emission while exciting with 3 lasers as reported by the authors (“we acquire an additional image channel by exciting all used labels at once and collecting the entirety of the emitted light”) is known to cause photo-physical effects that increase non-linearly the photobleaching speed (DOI: 10.1038/nmeth.3891, DOI: 10.1038/nmeth.3405).
2. One of the claimed advantages of MicroSplit is that it enables more photon-efficient multiplex imaging, reducing phototoxicity and improving imaging speed.
 - a. Neither photon-efficiency nor reduced phototoxicity are demonstrated in this work. The authors should prove experimentally these claims.
 - b. The settings depicted in Figure 1A suggest an inefficient microscopy configuration, with surprisingly narrow emission bands, where a significant portion of the emission signal is lost. Using broader emission bands would collect a large number of fluorescent photons, increasing the efficiency of the system likely 2-fold and resolving, in a more practical approach, the motivation of this work. The authors should clarify how the choice of more efficient microscopy settings would affect the improvements of this approach.
 - c. Most samples used in this work are fixed samples labeled with Alexas dyes. The authors should clarify how phototoxicity has any effect on this type of samples.

d. Regarding live cell imaging: the authors should demonstrate and experimentally prove the reduced phototoxicity in live samples, ideally in 3D considering the claim 2 of optimized analysis of 3d fluorescence imaging dataset.

3. The authors freely intermix the terms unmixing, semantic unmixing, computational multiplexing. This may be misleading to readers as the term “unmixing” is traditionally associated with spectral unmixing, where signals are mathematically decomposed into component intensities. MicroSplit does not perform spectral unmixing but instead predicts structures without preserving spectral intensity relationships. I would suggest the authors clarify the terms or intent of the approach. Perhaps a more appropriate term would be “semantic splitting” or “computational multiplexing.”

4. This work appears to use a total of 24 datasets.

a. This number appears to be substantially small, in comparison to other works. For example, Christiansen et al uses ~200,000 images. The authors should comment on the risk of overfitting to specific training conditions, given the limited number of datasets.

b. How do the multiple optical elements affect the network? (e.g. different lenses, different beamsplitters/filters)? How robust is the approach to different optics/instruments? The authors should validate the generalizability of MicroSplit.

c. How do experimental settings affect MicroSplit? Different pixel sizes/zoom level, different type of detectors, different types of noise (which appear to be ignored in this work) contribute differently to fluorescence microscopy images. The authors should comment or prove effectiveness of the approach under varying experimental settings.

5. Training modes and results:

a. The authors state in reference to training mode I “We generate mixed input images by pixel-wise summation of multiplexed image channels. These input images closely resemble what can later be acquired in a single acquisition” this statement may be incorrect depending on the type of fluorophores, excitation wavelength and emission filters used, owing to the different excitation and emission spectra of fluorophores.

b. In one of the notes the authors state the “ground truth” used is the “target images” or images acquired by the instrument. These images are intrinsically still spectrally overlapping, hence suggesting the reference image used by the authors is already “spectrally mixed”. The authors should clarify if this was the intent, as these images are not “unmixed”.

c. Training mode III seems to have a more realistic acquisition using 3 illuminations and collecting one single channel. These more realistic samples yield the lowest performance of the method presented (Table 1). The authors should acquire more realistic data (e.g. using same fluorophore for 3 targets) and in larger quantity to prove more convincingly the effectiveness of the approach.

d. Most of the datasets used in this work utilize Alexa dyes, which are bright and generally difficult to photobleach when imaged with efficient experimental settings. The authors should clarify this choice as opposed to a broad variety of commonly used genetically encoded fluorophores. Based on supplementary material it seems (though it is not clearly described) that Pavia-P23 dataset uses eGFP. Assuming Pavia-P24 is using the same samples described in Pavia-P23, the results reported in Table 1 for this sample substantially score lower than the Alexa samples. The authors should clarify the performance of the system against less-bright fluorophores or state more clearly the limitation of this approach to bright samples.

6. Segmentation results (Figure 3):

a. Figures 3B and 3C report high DICE scores, but visual overlays show significant discrepancies between predictions and ground truth. Particularly, 3B shows substantial amount of under-prediction or over prediction for A3 and A2 respectively.

b. The reported DICE values do not seem to match the visible alignment between images. It is unclear whether the DICE scores are computed over thresholded regions (positive binary) or across the entire image. The authors should clarify how these values were computed and justify their accuracy.

7. Improving photon budget: the authors claim “imaging two structures in a single channel can free up photon budget, which then becomes available for imaging at a higher frame rate, imaging at a higher signal-to-noise ratio, or imaging more labeled structures in additional image channels”.

a. What assumption is used for this statement? The imaged channel has a broader spectral band? If labels are different, due to the stochastic emission of fluorescence, shaped as the spectrum of the fluorophore, collecting one channel or collecting multiple channels (if done efficiently) will not change the total number of photons emitted by the fluorophores or detected by the detectors. The authors should clarify and ideally demonstrate this statement.

8. Removing unwanted imaging artifacts: authors state that “any subsequent analysis can be performed with the images predicted by MicroSplit”. The results shown in figure 2 suggest a creation of additional signal in the images as well as a blurring artifact in the prediction. This is echoed by the results in figure 3 and 4 where intensity distributions are seemingly altered in a gaussian blur, altering any intensity-based analysis the predicted images. The data presented by the authors contradicts this statement from the authors. The authors should correct the statement to align with their results.

9. In the limitations section the authors suggest that “a higher degree of skew (unequal channel intensities) makes it difficult for

MicroSplit to pick up the weaker of the two signals. For users to balance the intensity emissions, the laser power for each fluorophore (in most of the experimental settings described by the authors in supplementary) need to be adjusted. By looking at each emission channel independently. This contradicts the motivation described at the beginning of the manuscript, of collecting a single channel. The authors should clarify this statement.

10. In limitation section the authors state “although we observe blurriness in predictions for the weaker channel in a skewed HT-P23A derived task” this is true also for the results in figure 1, 2, 3, 4, S4, S10, S11, S12, S13, S14, S15, S16, S18, S19, S20. The authors should update their statement accordingly or explain the visible blurriness that appears in majority of the predicted images.

11. In limitations, in reference to changing scale of images (Figures S2, S3) authors state that “MicroSplit can successfully unmix the structures even when the smallest nonzero scaling factor is applied”. The results in S2 Magnification factor 1 Prediction Ch1, Ch2 show very evident artifacts. Magnification factor 1.125, Prediction Ch2, cells on the right and bottom side show evident clipping artifacts, similarly in Mag factor 1.25. The authors should correct this statement or clarify the differences.

12. The authors propose using a self-referential calibration approach (Figure 2) where the same sample is used to evaluate its own error.

a. The authors should better discuss this approach with respect to tautological validation.

b. Would this estimation of calibration only reflect how confident the model is in itself, rather than how accurate it actually is? The results of this calibration appear in contrast with the empirical results shown in most of the figures, reporting a frequent alteration of intensities in the predicted channels respect to the target channels. This change in intensities would considerably alter results of any intensity-based analysis, affecting its reliability.

13. Quantification methods used by the authors seem to not match visual assessment. Example, the maps shown in Figure 3 “overlays” suggest a much lower visual performance then the DICE score reported. This may be due to the averaging over the entire image (mostly dark), instead of the regions with information content. Example in task 1, Analyst 3 overlay shows a substantial under-prediction, while the DICE score shows over .75. The authors could show more maps of estimated error and correlate those better with the single-value results (e.g. MicroSSIM or DICE) to better prove the method and increase the robustness of the presentation.

14. The differences between the combination of μ Split with denoiSplit and the current method are not well described. The authors should better describe MicroSplit with respect to μ Split and denoiSplit to position it as a new method rather than an incremental improvement.

15. Presentation and readability of the manuscript

a. Text clarity and accessibility should be considerably improved. The manuscript seems to be written for a highly technical journal, more in line with ML style, requiring readers to refer to multiple external sources to fully grasp the methodology and prior work, necessitating significant background reading, which may limit accessibility for a broader audience. The authors may consider refining the presentation to improve clarity and readability, particularly if aiming for a journal like Nature Methods, which caters to a diverse readership spanning different scientific disciplines.

b. Regarding data presentation and organization: the way results are reported deviates from common conventions for this type of journal, making interpretation more challenging. For example, Table 1 presents multiple tasks and performance metrics in a highly condensed format, requiring cross-referencing with the additional supplemental material to fully understand the comparisons and tasks performed. Error measurements for this same table are reported in a separate table (Supplementary Table ST9), making it difficult to quickly locate and interpret key findings. A more structured and intuitive layout (or plot) could enhance the readability and accessibility of the data.

c. Dataset information: the description of the datasets used in the study could be more clearly structured. While the manuscript states that datasets are shared, which is great, a comprehensive summary table detailing dataset characteristics, such as image size, pixel resolution, microscopy settings, bit depth, and time points, would improve transparency. Some of this information is included for certain datasets, but for others, it is either difficult to find or not explicitly stated. Providing a standardized dataset summary would make it easier for readers to assess the scope and applicability of the study.

Reviewer #1 (Remarks on code availability):

The MicroSplit code is available and includes installation instructions, but it lacks explicit examples on how to apply the method to real data. While the provided documentation describes how to set up the environment, there are no practical examples demonstrating how to run the model on fluorescence microscopy images. This structure caters more toward a computer science-oriented audience, making it less accessible to researchers without a good programming background. For those familiar with machine learning workflows, the code is understandable, but a step-by-step guide with example datasets would greatly improve and facilitate usability and reproducibility. Enhancing the documentation with practical use cases would make the resource more accessible to a broader community. A simple UI would also facilitate adoption.

Reviewer #2 (Remarks to the Author):

Summary of the Key Results:

The authors present MicroSplit, a deep-learning-based method for computational multiplexing in fluorescence microscopy. This technique enables multiple cellular structures to be imaged within a single fluorescent channel and subsequently unmixed computationally, reducing photon exposure while maintaining imaging quality. MicroSplit employs a variational splitting encoder-decoder (VSE) network, leveraging posterior sampling to estimate prediction uncertainty. The manuscript demonstrates that MicroSplit can effectively separate up to four superimposed structures, denoise images, and provide uncertainty quantification at a quality level suitable for downstream tasks.

Originality and Significance:

While the concept of computational multiplexing is not new, with prior work such as PICASSO addressing similar problems, MicroSplit distinguishes itself by requiring only a single superimposed image as input. This eliminates the need for spectral overlap measurements, which presents a significant advantage. The supplemental material compares MicroSplit to PICASSO effectively, though moving this comparison into the main text would strengthen the manuscript. Overall, the method has the potential for broad adoption and significant impact in fluorescence imaging.

Data & Methodology:

The channel segmentation methodology is well-reasoned, but the Error Estimation, Uncertainty, and Calibration section requires substantial revision and clarification. The authors state: "We therefore estimated the true error by computing the pixel error between the MMSE prediction of MicroSplit and the ground truth target images we derived from the available training data, and plotted this 'true error' against the RMSE errors we described above."

However, the calibration curve axes are labeled RMV vs. RMSE, yet RMV is not explicitly defined in the manuscript, making interpretation difficult. Additionally, the claim that RMSE can be calculated solely from posterior samples suggests that the authors are computing RMSE between the posterior samples and the posterior MMSE image. Since RMSE can be expressed as the square root of the sum of variance and squared bias, and given that the MMSE image is an unbiased estimator, pixel-wise RMSE should correspond to the standard deviation of the posterior. RMV represents a similar metric and would be expected to exhibit a nearly linear correlation with RMSE.

If the goal is to demonstrate that low posterior variance correlates with low error and can serve as a proxy for uncertainty, I would recommend the following modifications:

1. Increase the number of posterior samples beyond 50 to allow for a more robust estimation of pixel-wise variance.
2. Compute the pixel-wise 'true error' using the posterior MMSE image and plot it against the posterior variance, providing a quantitative correlation metric.
3. In Figure 2, replace the Sample 1 - Sample 2 heat map with a posterior variance heat map, and replace the RMSE column with the true error heat map, enabling readers to visually assess the relationship between variance and error.
4. Explicitly provide equations for all calculations related to calibration and uncertainty estimation.

Validity of Approach, Quality of Data, and Presentation:

The methodology is generally well-structured, with the exception of the Error Estimation, Uncertainty, and Calibration section. Several figures require revisions for clarity:

- * Figure 1.A: The axis labels are difficult to read, and a legend is necessary. Additionally, the figure caption refers to emission filters, whereas the main text describes them as excitation filters—this discrepancy should be resolved.
- * Figure 2.A: The second lateral context patch should be explicitly labeled for clarity.
- * Figure 2.B: The term RMV is undefined in both the calibration plots and the main text, making interpretation challenging. See further comments above
- * Supplemental Figure 5: The x-axis should have consistent spacing, either in linear or logarithmic scale, or alternatively, the figure could be reformatted as a bar graph for improved readability.

Regarding data quality, the authors have taken care to collect and describe three distinct datasets. However, the manuscript's clarity could be improved by more precisely distinguishing between traditional sequential multiplexing (multichannel acquisition) and the proposed simultaneous multiplexing (multiple structures within a single channel). Introducing explicit terminology for these concepts would reduce ambiguity.

References:

The manuscript cites an adequate body of prior work, and I did not identify any major missing references. However, the claim that 32.53 PSNR and 0.886 MicroMS-SSIM are sufficient for downstream analysis would benefit from additional supporting literature.

Final Recommendation:

The manuscript presents a promising computational multiplexing method with significant potential impact. However, revisions are necessary, particularly in the Error Estimation, Uncertainty, and Calibration section, as well as in figure clarity and

definitions. I recommend major revisions to improve methodological transparency, ensure rigorous uncertainty quantification, and enhance clarity in data presentation. With these improvements, MicroSplit has the potential to contribute meaningfully to the field of fluorescence microscopy.

- Eric Markley

Reviewer #3 (Remarks to the Author):

MicroSplit combines the advantage of uSplit and denoiSplit, also with extension to volumetric images, to tackle the semantic unmixing problem for fluorescence microscopy. Overall, this work is both theoretically novel and practically useful, supported not only by experiments on extensive and diverse datasets / applications, but also by comprehensive discussion regarding the performance, caveats, practical considerations, etc. I would definitely endorse for publication, but with some minor suggestions.

A few strengths I really want to highlight:

- Super flexible training settings (different modes in Section 2.1).

I think this is really useful in practice, since different applications / use cases may have different data availability and the proposed framework can still be applicable in a lot of different situations. Importantly, comprehensive discussion and caveats are provided for better understanding of different training modes. This will serve as a useful guide for users to design their assay accordingly for their needs.

- The variational model as the backend supports informative error estimation.

I really appreciate that the variational model backend, which fundamentally supports error estimations. The calibration step confirms the effectiveness. The error estimation and calibration make this method, not just something looking cool, but equipped with a handle for users to take full control of the trustworthy, which is important for deep learning based image predictions.

- Evaluation beyond PSNR and SSIM.

By showing MicroMS-SSIM, the evaluation already provides a comprehensive view from an image perspective. But, it is still not enough to represent its impact on downstream analysis or its impact for answering the underlying biological questions. I really appreciate the additional experiments in Section 2.4, which provides application-appropriate validation to make sure the prediction is biologically valid for the specific application.

Besides the highlight above, I can confirm, to the best of my knowledge, the described methods are sound, the quality of presentation is good, easy to understand, and the use of statistics / evaluation metrics are appropriate. There are only some minor suggestions:

(1) Section A in Supp. can be significantly improved.

In general, the method proposed in this work combines previous work in uSplit and denoiSplit into a single setup. But, it is not quite clear how exactly this "final product" looks. In theory, I could check the previous publications for details. But, this may not be straightforward, since this "final product" is not a simple copy-and-paste from previous works. Therefore, I would highly recommend to provide more details in the Supp. so that, in theory, people should be able to reproduce the exact method based on the description in this paper. Here are some examples of what are not quite clear to me in the technical details.

A. In Supp. page 1, "In Figure S1, we present the architecture of MicroSplit, which has, LC inputs and Noise models, all integrated into a single setup." But, when you look at Figure S1, it is not clear at all, where the Noise model is and how it is integrated exactly.

B. In Figure S1, what is the "Input Branch"? Any computation within the "Input Branch"? I assume it is just some kind of connection in the neural network, correct?

C. Also in Figure S1, the white squares in the network figure are getting smaller through layers, is it the result of pooling or unpadded convolution? If it is pooling, then the size is not halved.

D. Again in Figure S1, the first part of all three types of models are three white squares. Are these three consecutive convolution layers? Or something else?

E. In Figure S1, there are center crops in Lean-LC. Is there anything similar in Regular-LC and Deep-LC?

I know a lot of these details can be discovered from the released code (great! Open-source is important!), but it is still good to explain the technical details in the paper as a top-tier publication.

Note: the missing details mentioned above did not affect my understanding of the key idea of the method.

(2) Is there a discussion on why "up to four structures"? I am not sure if I miss this somewhere in the paper. But, in general, what is the biggest trouble if we make an attempt on five or six structures? If we use Deep-LC or even a Deeper-LC, will five or

six structures become possible?

(3) Will the model be released on Bioimage Model Zoo? I don't think this will affect the evaluation of this paper. This is more of my own wish from a user's perspective.

(4) There are some other really minor things in terms of presentation. You could even ignore this part.

- In Table 1, it would be better to use 1, 2, 3, ... or T1, T2, T3, ... as indices for tasks, rather than I, II, III.
- Would it be possible to provide a vision overview of all different training modes? I can appreciate the high flexibility of the propose method. A summary table or some kind of visual abstract / overview would be really helpful.
- Personally, I would really appreciate a few sentences (maybe in the Supp.) describing how the variational models allow error estimation using plain English. I believe these few sentences would be super helpful for the users without deep knowledge of machine learning to understand where this error comes from.

Reviewer #3 (Remarks on code availability):

I didn't run the code, but I read the code on Github, very clearly written scripts and software. Currently, the code is organized in a way focusing on reproducing the results in this paper. This is fair. But, from a user's perspective, I would like to see a clear instruction on what to do if I want to run training mode 1, training mode 2, etc., with my own data, and how should I organize my own data (2D or 3D). I have no doubt the documentation will get better.

Version 1:

Decision Letter:

Our ref: NMETH-A59732A

17th Nov 2025

Dear Florian,

Thank you again for your response to the reviewers' comments on your revised manuscript "MicroSplit: Semantic Unmixing of Fluorescent Microscopy Data" (NMETH-A59732A). I apologize for the delay, but I am very happy to now tell you that we can offer in principle to publish it in Nature Methods, pending minor revisions to satisfy the referees' final requests and to comply with our editorial and formatting guidelines.

TRANSPARENT PEER REVIEW

ORCID

IMPORTANT: Non-corresponding authors do NOT have to link their ORCIDs but are encouraged to do so. Please note that it will not be possible to add/modify ORCIDs at proof. Thus, please let your co-authors know that if they wish to have their ORCID added to the paper they must follow the procedure described in the following link prior to acceptance: <https://www.springernature.com/gp/researchers/orcid/orcid-for-nature-research>

Author names using non-Roman characters

Nature Portfolio journals can support presentation of author names using non-Roman characters in the HTML version of the article. If you wish to, please include author names in parentheses after the Roman-character spelling; [see example online here](https://www.nature.com/articles/s44222-024-00258-2). Currently supported scripts are: Arabic, Chinese, Cyrillic, Devanagari, Greek, Hebrew, Hangul, Japanese and Persian. You will be asked to verify the rendering is correct at proof stage.

Thank you again for your interest in Nature Methods. Please do not hesitate to contact me if you have any questions. We will

be in touch again soon.

With best regards,
Allison

Allison Doerr, Ph.D.
Chief Editor
Nature Methods

Reviewer #1 (Remarks to the Author):

The authors provided extensive responses, often detailed and technical, and I commend them for their work. They clearly invested effort into explaining their rationale, correcting terminology, and adding supplementary material. The expanded Limitations section is an improvement on the clarity of the technique, though it lacks some fundamental points made by authors in rebuttal, which would greatly benefit readers.

Some of the responses shift the burden of proof back onto the reviewers (e.g., suggesting that some objections are “misconceptions” and that readers/reviewers can find information in the metadata of images). Typically, such reviewer concerns would motivate concrete improvements to the manuscript itself, yet in this case many of these points remain unresolved in the main text. This style weakens the persuasiveness of the rebuttal.

For example, reviewer comments on figures remain somewhat unaddressed, despite multiple reviewers pointing to unclear representations (e.g. Figure 1a, which typically aligns with the experimental settings, the figure could be updated or it should be made clear in the caption that these are cartoon images and not actual experimental settings).

A well written manuscript would provide, in a self-contained setting, all essential information for a clear, not-misguided interpretation of the work. Manuscript clarity improvements rest with the authors, which can refer to the large literature in Nature Methods publications for practical examples on how to present methods and results for the broad readership of this journal.

The manuscript could, with proper improvements and proper address of reviewers’ concerns, be suitable for publication.

Major concerns:

1. A central limitation of MicroSplit is that the network must be trained from scratch for each dataset or experiment, rather than functioning as a broadly generalizable model. This is stated unambiguously in the rebuttal: “MicroSplit is meant to be trained from scratch for every semantic splitting task, effectively eliminating the risk of overfitting” and “MicroSplit will be trained on data from one optical setup and the trained model later applied to more data from the same setup.” These clarifications make it evident that the approach is task-specific and dataset-specific.

However, the main manuscript does not present this point with sufficient clarity. In the current version, this limitation is only implicit in the training sections (e.g., where the authors describe “each splitting task” or report results on a per-dataset basis). By contrast, the abstract states: “We demonstrate the robustness of MicroSplit across various datasets and noise levels and show its utility to image more, to image faster, and to improve downstream analysis.” This phrasing could easily be interpreted as implying that a single generalized network was trained and applied across multiple datasets, which is not the case. Readers may therefore be misled regarding the scope and generalizability of the approach.

To address this issue, the authors should clearly state in the main text, and explicitly in both the Abstract and the Methods/Approach/Limitations sections, that MicroSplit must be trained separately for each dataset/task. This clarification is important for text clarity and will prevent readers from misunderstanding the intended scope/application of the method.

2. With regard to photobleaching and photon efficiency. The authors rebuttal helps understanding the reasoning behind this claim, provide a valid theoretical explanation.

It is clear that, a broad band emission detection will yield a higher photon throughput compared to a narrow band (or sum of narrow bands). This fact however, based on the statements of authors, is independent of MicroSplit. A photon-efficient experimental setting would minimize the emission detection gap between channels, already improving the photon efficiency. Based on imaging settings reported, a sufficient empirical demonstration is currently reported as HT-LIF24 (ST12) the authors should quantify the improvement (e.g. x-fold based on the split acquisition) in support of the method.

3. The HT-LIF24 data reports (ST12) and Fig s59 are quite helpful. Impressive how the Exposure = 2ms in Prediction Ch3 and corresponding target image (S59) have a MicroMS-SSIM of 0.869 (Table ST12) despite losing almost all high-frequency details, as reported by the authors and as evident by visual inspection. This is a known challenge of SSIM that accounts primarily for global structural statistics and may explain the intensity mismatches reported in the work. For the scope of the MicroSplit downstream applications (“segmentation, tracking, detecting the presence or absence of certain structures, counting structures, and estimating dimensional properties such as radius or length of structures of interest”) this may be ok, provided that high-frequency details do not compose the structures being counted. The authors updated the limitations section to state “However, while not entirely impossible, caution is advised when applying MicroSplit to downstream tasks that rely on precise pixel intensities.”, which is a useful statement, though still a little broad.

I would suggest clarifying this section to specify that high-frequency features (for example those in S59 Prediction Ch3 2ms, 3ms and 5ms) will provide high amount of error in tracking, counting, estimating dimensions and similar downstream

applications, narrowing down the application of MicroSplit to more reasonable scenarios.

Minor comments:

4. Spell check "input branch does not has pooling"
5. Figure S2, S3 "Unmixing similar.. " should be "Semantic unmixing of similar.. " to keep consistency with updated terminology in the rest of the manuscript
6. HT-LIF24 is both referenced to as HTLIF24 and HT-LIF24, pick one for consistency

Reviewer #1 (Remarks on code availability):

The code appears appropriate, the descriptions (README and in code) provided for installation are sufficient to reproduce the work with examples on how to run the code.

Reviewer #3 (Remarks to the Author):

The authors have addressed all my concerns. Great work!

Reviewer #3 (Remarks on code availability):

the code is significantly improved.

Version 2:

Decision Letter:

26th Mar 2026

Dear Florian,

I am pleased to inform you that your Article, "MicroSplit: Semantic Unmixing of Fluorescent Microscopy Data", has now been accepted for publication in Nature Methods. The received and accepted dates will be 10 February 2025 and 26 March 2026. This note is intended to let you know what to expect from us over the next month or so, and to let you know where to address any further questions.

Over the next few weeks, your paper will be copyedited to ensure that it conforms to Nature Methods style. Once your paper is typeset, you will receive an email with a link to choose the appropriate publishing options for your paper and our Author Services team will be in touch regarding any additional information that may be required. It is extremely important that you let us know now whether you will be difficult to contact over the next month. If this is the case, we ask that you send us the contact information (email, phone and fax) of someone who will be able to check the proofs and deal with any last-minute problems.

Authors may need to take specific actions to achieve compliance with funder and institutional open access mandates.

If your research is supported by a funder that requires immediate open access (e.g. according to <https://www.springernature.com/gp/open-science/plan-s-compliance>> Plan S principles or the <https://www.springernature.com/gp/open-science/us-federal-agency-compliance>> NIH public access policy) then you should select the gold OA route, and we will direct you to the compliant route where possible. Because authors warrant under our subscription licensing terms that they haven't committed to licensing any version of their article under a licence inconsistent with the terms of our agreement – including the applicable embargo period – publication under the subscription model isn't suitable for authors whose funders require no embargo.

You may wish to make your media relations office aware of your accepted publication, in case they consider it appropriate to organize some internal or external publicity. Once your paper has been scheduled you will receive an email confirming the

publication details. This is normally 3-4 working days in advance of publication. If you need additional notice of the date and time of publication, please let the production team know when you receive the proof of your article to ensure there is sufficient time to coordinate. Further information on our embargo policies can be found here:
<https://www.nature.com/authors/policies/embargo.html>

If you are active on Twitter/X or Bluesky, please e-mail me your and your coauthors' handles so that we may tag you when the paper is published.

Best regards,
Allison

Allison Doerr, Ph.D.
Chief Editor
Nature Methods

** Visit the Springer Nature Editorial and Publishing website at http://editorial-jobs.springernature.com?utm_source=ejp_NMeth_email&utm_medium=ejp_NMeth_email&utm_campaign=ejp_NMeth for more information about our career opportunities. If you have any questions please click [here](mailto:editorial.publishing.jobs@springernature.com). **

Open Access This Peer Review File is licensed under a Creative Commons Attribution 4.0 International License, which permits use, sharing, adaptation, distribution and reproduction in any medium or format, as long as you give appropriate credit to the original author(s) and the source, provide a link to the Creative Commons license, and indicate if changes were made. In cases where reviewers are anonymous, credit should be given to 'Anonymous Referee' and the source. The images or other third party material in this Peer Review File are included in the article's Creative Commons license, unless indicated otherwise in a credit line to the material. If material is not included in the article's Creative Commons license and your intended use is not permitted by statutory regulation or exceeds the permitted use, you will need to obtain permission

directly from the copyright holder.

Answer to the Reviewers

We want to start this lengthy document by **thanking the reviewers** for their time and diligent work. We understand you are reviewing our work instead of doing your own, and we truly appreciate the depth at which you dove into our new method and the results it produces.

Below, we respond directly to all your feedback. To make it easier to read this document and also be reminded to your own opinion about our work, we decided to display the unedited reviewer text in a dark gray, while giving our responses a slightly blue color.

Before addressing the reviewers' specific comments, we would like to highlight a few updates we have made to the manuscript on our own initiative, which were not prompted by the reviewers' suggestions.

We have revised the results of our 3D tasks. While the older results were not incorrect, however, a few hyper-parameters used in their training were specifically tuned for the reported task. Since we do not want to optimally pick the hyper-parameters for each task, but instead are committed to be using **a single set of hyper-parameter across all tasks** of one type, *i.e.* all 3D tasks, and all 2D tasks¹, we have rerun those experiments and updated the reported quantitative evaluations. More specifically, we have updated:

- Table 1, Task I: The older result was achieved when using 9 consecutive frames from the z-stack as input. However, all other 3D semantic unmixing tasks used 5 consecutive frames from the z-stack as input. In Supplementary Table ST3, for completeness, we already did investigate how the performance improves with more z-stack elements as input.
- Table 1, Task XXI and XXII: The older result for these tasks were achieved when using LC inputs. To enable our setup to work also on GPUs with lower memory, we had decided to disable LC for 3D tasks. Still, for these two tasks (XXI and XXII), we had LC turned on by mistake.

We have also **added a new dataset called HHMI-D25**, which has three sub-datasets, from which we derive a new and quite hard task XXIII (see Supplementary Figure S56 and S60). We use this task as an example for how users can debug and fix difficult semantic unmixing tasks (see Section 2.5 and Supplementary Section B.1.1, and the respective Tasks XXXI to XXXVI which can maybe best be compared quantitatively in Supplementary Table ST1).

Additionally, we have also **expanded on an analysis and discussion of the role of SNR in the input data and this influences the performance of semantic splitting**. To this end we used the HT-LIF24 and HHMI-D25 datasets (see Supplementary Section B.1).

Below we comment directly to the reviewer comments we have received.

¹ Except tasks from the SIM dataset Chicago-Sch23, for which the noise properties are quite different from all other datasets.

Reviewer #1 (Remarks to the Author):

In the manuscript “MicroSplit: Semantic Unmixing of Fluorescent Microscopy Data” from Ashesh et al, the authors describe a computational approach aimed at overcoming the limitations of fluorescence multiplexing by predicting multiple fluorescently labeled structures from a reduced number of imaging channels. The proposed method seeks to address issues related to overlapping fluorophore excitation spectra, reducing the number of required exposures in multiplex acquisitions, and overcoming the limits of maximum sampling frequency in live cell imaging scenarios. Their approach builds upon two of their previous publications and introduces a volumetric image processing framework that enables the calibration of the network using the data itself while estimating prediction errors.

We would like to clarify that the volumetric image processing framework does not enable calibration of the network as inferred by the reviewer. Volumetric image processing helps improve the performance and reduces prediction time. We adopted the calibration procedure we have first described in denoiSplit [7], which is a technical precursor to this work.

The claimed novelty:

1. Combining elements from two prior computational approaches (μ Split and denoiSplit) into a single framework, leveraging machine learning to address fluorescence multiplexing challenges.
2. Processing volumetric image data using a highly optimized network architecture designed specifically for 3D fluorescence imaging.
3. Introducing a self-calibration procedure that assesses the quality of the trained network and estimates prediction errors, enhancing reliability in fluorescence reconstruction.

Claim 1 is only referenced in the manuscript, not supported by any practical demonstration, code is made available for readers to review.

We thank the reviewer for bringing to our attention that this was not well articulated in the submitted manuscript. We have updated the supplementary section A. Our line of argument is that both our previous works denoiSplit and μ Split, aimed for a Machine Learning audience, had complementary advantages. μ Split was GPU efficient and provided multiple meta-architecture variations, allowing the user to pick among the three μ Split variants, with each variant striking a different balance between performance and GPU utilization. denoiSplit did not have this feature, but it enabled denoising, posterior sampling, uncertainty quantification and calibration. In the creation of MicroSplit, we adopted the architecture design from μ Split, which makes it GPU efficient and allowed multiple architecture variants, and from denoiSplit adopted KL loss formulation, noise model integration, uncertainty quantification and calibration. Hence, we ended up with a GPU efficiency method that also conducts unsupervised denoising and can do all this also volumetrically (in 3 spatial dimensions) due to a new implementation we added in this work. With results presented in this manuscript, one can verify that MicroSplit performs unsupervised denoising and supports posterior sampling, uncertainty quantification and calibration. In our tutorial notebooks, we are describing that Deep-LC is the default configuration

and will, once fully implemented, describe how the user can switch between the three μ Split variants.

Claim 2 is addressed by mathematical demonstrations in supporting material but not associated with any explicit timing of analysis of 3d fluorescence imaging dataset.

We want to state that processing volumetric data is not about the inference speed. It is about performance. Performance improves when working on volumetric data. See Supplementary Table ST3.

Claim 3 is presented with multiple datasets and plots across the manuscript, with some supporting material.

The manuscript, in its current version, is framed to present an alternative to standard unmixing techniques (i.e. spectral), arguing that MicroSplit achieves efficient reconstruction of fluorescent structures through learned feature extraction rather than intensity-based spectral decomposition. The authors suggest that this enables more photon-efficient multiplex imaging, reducing phototoxicity and improving imaging speed.

The proposed method is designed to address these fluorescence imaging challenges, however, similar problems have been explored in similar studies and/or commercially, using alternative approaches:

1. Machine learning-based fluorescence prediction in both 2D and 3D, such as the work from Guo et al (DOI: 10.7554/eLife.55502) , that of Ounkomol et al (DOI: 10.1038/s41592-018-0111-2) , Wang et al (DOI: 10.1038/s41592-018-0239-0) , Christiansen et al (DOI: 10.1016/j.cell.2018.03.040) to name some, though with recent machine learning improvements the count of applications of machine learning articles focusing on fluorescence prediction is substantially larger

While all of the four mentioned works indeed use machine learning, the tasks they tackle are different in nature to the one we are addressing. One of these works performs a super-resolution task, while the other three perform virtual-staining. In virtual staining, the prediction objective is superficially somewhat similar to our objective: predict (multiple) fluorescence structures (channels) from a given input image. This input, however, is a label-free image, e.g. a brightfield image. With MicroSplit, instead, we propose to split superimposed fluorescent channels (note the pixel-wise sum of our predicted channels is in theory the given input image (modulo noise) — a fact that is by no means true for virtual staining approaches). Since the data itself is different, it is not easily possible to compare our method with virtual staining methods. Additionally, it would not be a fair comparison. We address virtual staining methods in more detail in our response to the question 1b.

2. Commercial systems incorporating spectral unmixing methods are already widely used in fluorescence imaging and can be found in commonly available instruments from Zeiss, Leica, Nikon and Olympus.

While spectral unmixing methods indeed perform image decomposition, our methodology is different from spectral unmixing and is, can, and should also be applied when data is imaged on camera-based systems. Spectral unmixing requires multiple image channels as input, with each image capturing a different frequency band of the overall emission spectrum: that is, it requires dedicated and expensive white lasers and spectral filters/detectors. Instead, our approach works with any standard fluorescence microscope and can, therefore, be immediately used by any lab across the globe.

A key difference between MicroSplit and these prior methods is that the latter approaches often leverage real intensity distributions to calibrate predictions, whereas MicroSplit primarily focuses on structural reconstruction without maintaining intensity fidelity.

Virtual staining approaches predict fluorescence from an entirely different imaging modality and predicted fluorescent intensities are arguably only as trustworthy as the quantity of the biological structures in question can be predicted from the input modality itself (which is often barely possible, leading to untrustworthy and imprecise predictions). MicroSplit, on the other hand, is taking the given input and splitting it into its individual “summands”. Spectral unmixing is a different computational task that naturally has its place in microscopy and in biological imaging. We believe that the same is true for MicroSplit, which can easily be used on off-the-shelf camera-based systems (spectral imaging can not) and in cases where virtual staining will lead to uncertain predictions (see also response to 1b, below).

If the claims stated by the authors are confirmed, the work may represent an advance in the application of machine learning to fluorescence imaging. However, fundamental concerns remain regarding the practical advantages of this method.

The study lacks strong experimental validation to support its claims, particularly regarding photon efficiency and phototoxicity reduction. The multiplexing scenario presented is relatively simple (3-4 channels, 3-4 labels), limiting its applicability to more complex imaging tasks. Additionally, the dataset size is small (24 datasets), raising concerns about overfitting and generalizability compared to larger-scale studies. The terminology surrounding “unmixing” is inconsistent, as MicroSplit does not perform true spectral unmixing but rather predicts structures without preserving intensity relationships.

We have fixed the nomenclature (see response to point 3). Here we want to make an important distinction between “spectral unmixing”, an important technique and method in the imaging field, from the “semantic unmixing” we are proposing. They are complementary techniques that can help microscopists and biologists in different use-cases. We will iterate on this also in other responses below.

The presentation is highly technical, requiring extensive cross-referencing of prior work and supplementary materials, making it difficult to follow. There are also concerns with performance validation, as reported DICE scores do not align well with visual assessments, and potential blurring artifacts may compromise downstream image analysis. While the method may have potential in fluorescence imaging, its motivation and practical advantages over existing techniques remain unclear.

I ask the authors to address the following concerns.

Major comments

1. The primary motivation of this work is to overcome challenges in fluorescence multiplexing, reducing the number of required exposures in multiplex acquisitions, and overcoming the limits of maximum sampling frequency in live cell imaging scenarios.

a. In the definition used in this manuscript, a three-channel fluorescence microscopy dataset is considered multiplexing. Three-channel/three-labels fluorescence is a rather simplistic multiplexing scenario. The authors should clarify why such simple scenario is considered here, as opposed to some more complex scenario (10-15 labels).

Even imaging just two fluorescent channels typically requires multiplexing, and surpassing four channels becomes challenging without multiple rounds of introducing and removing fluorescent markers. This limitation is especially pronounced in live-cell imaging. As detailed in Supplementary Section B, the difficulty depends on several factors. In Section B.3 ("Similarity of Structures to Be Unmixed"), we quantitatively demonstrate that as the structural similarity between targets increases, semantic unmixing becomes progressively harder for the network. This was further validated in μ Split using a synthetic dataset of simple sine-wave-generated structures.

Notably, Table 1(c) includes a successful four-channel semantic unmixing result, suggesting that in favorable cases, biologists could group up to four fluorescent structures into a single acquisition channel. By extending this approach across multiple multiplexed channels, it may be possible to image well over 10 distinct structures—even on a standard spinning-disk microscope in live-cell experiments. We are not aware of other methods in existence that could enable this.

b. The work here presented appears to require triple labeling with different fluorophores. The authors should clarify how this triple-labeling approach is advantageous compared to the approaches for machine learning-based fluorescence prediction referenced above (e.g. DOI: 10.7554/eLife.55502)

Virtual staining is a different task than the semantic unmixing. Here, we would like to raise few additional points in support of our approach. While we agree that virtual staining has indeed proven useful in several cases, we argue from a conceptual point of view that it has the following two issues not faced by our approach: (a) Incomplete information: Some of the structural details, particularly the high-frequency details in the fluorescence target structures often are absent in the brightfield input image. So, a virtual staining network needs to predict those missing details. In our setup, all the details present in the fluorescence target images are present in the input image. Hence, the network does not need to invent any structural details. It just needs to *decompose* the input into the predictions. (b) Structural Noise: Unwanted sample details present in the brightfield input image would act as noise, thereby complicating the prediction task. In our setup, input is, by construction the superposition of only those structures

which we wish to disentangle. We therefore argue that our approach can predict high frequency details and dim structures better than what a virtual staining task would do.

As emphasized earlier, virtual staining is fundamentally distinct from the unmixing task we present. To further support our approach, we highlight two key advantages over virtual staining:

1. *Complete Structural Information:* In virtual staining, high-frequency details of fluorescent structures are often missing in the brightfield input, forcing the network to predict absent features. In contrast, our input image inherently contains all structural details of the target fluorescence signals—the network only needs to decompose them, not “invent” them.
2. *Minimized Structural Noise:* Brightfield images include extraneous sample details that act as noise, complicating virtual staining. Our input, by design, consists only of the superposition of structures to be disentangled, eliminating this source of interference.

We therefore argue that our approach can predict high frequency details and even dim structures better than a virtual staining approach would do from a brightfield acquisition (since it is simply a much harder task).

c. The authors should demonstrate how this approach overcomes the limits of maximum sampling frequency in live cell imaging scenarios while reducing photo-bleaching. Detecting fluorescence emission while exciting with 3 lasers as reported by the authors (“we acquire an additional image channel by exciting all used labels at once and collecting the entirety of the emitted light”) is known to cause photo-physical effects that increase non-linearly the photobleaching speed (DOI: 10.1038/nmeth.3891, DOI: 10.1038/nmeth.3405).

Temporal multiplexing is conducted in virtually every imaging facility on a daily basis. In all such cases, each channel is acquired after one other sequentially and imaging k channels is about k times slower than imaging a single-channel. Hence, improving the temporal sampling frequency is trivially achieved by MicroSplit since it allows to acquire a single channel containing multiple structures that can later be unmixed.

Regarding photobleaching, we agree with the reviewer that using multiple excitation lasers will cause faster bleaching. However, since photobleaching depends upon the total laser power and exposure duration, the unsupervised denoising capability of MicroSplit helps to save some photon budget. In this work, we acquired the HTLIF24 datasets with varying levels of exposure duration, and acquired images with exposure duration being as low as 2ms. We did this to demonstrate that our method, even with noisy short exposure input images, provides useful denoised results (see Fig. S12, for results on 2ms acquisition duration).

Additionally, we also propose to adopt imaging protocols where the same fluorescent marker is used to label multiple structures. In this case, a single excitation laser will be needed and the photon exposure would be multi-fold reduced.

2. One of the claimed advantages of MicroSplit is that it enables more photon-efficient multiplex imaging, reducing phototoxicity and improving imaging speed.

a. Neither photon-efficiency nor reduced phototoxicity are demonstrated in this work. The authors should prove experimentally these claims.

We would like to draw the reviewer's attention to the fact that denoising isn't just about cleaner images-it's also a tool to rethink the required photon-budget. Similar to previous deep-learning based denoising setups, the denoising capability of MicroSplit opens up the possibility of using a smaller photon budget to get images having the SNR of images acquired with higher photon budget. To this end, we added the table ST11 in the supplement to demonstrate this. In the table we treat the 3 channel data (Nucleus, Microtubules and Kinetocore) of 500ms acquisition duration sub-dataset in HTLIF24 dataset as ground truth and estimate their similarity with sub-datasets of HTLIF24 having lower acquisition duration. The utility of this analysis is to enable comparing these metric values with those provided for Tasks XXIII-XXVI presented in Table 1. For example, for channel 1 (C1), our prediction on 2ms superimposed input (Task XXIII of Table 1) is better than the 20ms groundtruth (last row of the tableST11). In simple terms we compare two situations: (a) acquire individual channels with 20ms acquisition duration, and (b) acquire a single channel containing the superposition of three structures with 2ms as the acquisition duration and use our Microsplit network to unmix. By comparing the quantitative metrics, we find that the second approach is better.

b. The settings depicted in Figure 1A suggest an inefficient microscopy configuration, with surprisingly narrow emission bands, where a significant portion of the emission signal is lost. Using broader emission bands would collect a large number of fluorescent photons, increasing the efficiency of the system likely 2-fold and resolving, in a more practical approach, the motivation of this work. The authors should clarify how the choice of more efficient microscopy settings would affect the improvements of this approach.

We ask the reviewer to appreciate that Figure 1A is a cartoon optimized for visibility. The concrete imaging setup for all datasets was optimized by the microscopists who created the respective data and was by no means influenced by us. The only reason to image with less than ideal microscope configurations was when we wanted to test the limitations of our method.

c. Most samples used in this work are fixed samples labeled with Alexas dyes. The authors should clarify how phototoxicity has any effect on this type of samples.

Phototoxicity is not a concern on fixed samples. It is on live specimen though, where MicroSplit can help to reduce the photon exposure in the ways explained above and in the manuscript.

d. Regarding live cell imaging: the authors should demonstrate and experimentally prove the reduced phototoxicity in live samples, ideally in 3D considering the claim 2 of optimized analysis of 3d fluorescence imaging dataset.

We believe that the reduction in light exposure by either imaging multiple structures in one channel or by relying on MicroSplit's unsupervised denoising capability are obvious.

If the editors share the reviewer's opinion, we will of course conduct this experiment and accept the cost and delay this decision will cause.

3. The authors freely intermix the terms unmixing, semantic unmixing, computational multiplexing. This may be misleading to readers as the term “unmixing” is traditionally associated with spectral unmixing, where signals are mathematically decomposed into component intensities. MicroSplit does not perform spectral unmixing but instead predicts structures without preserving spectral intensity relationships. I would suggest the authors clarify the terms or intent of the approach. Perhaps a more appropriate term would be “semantic splitting” or “computational multiplexing.”

Now, we consistently use the term “semantic unmixing” and use computational multiplexing only when introducing the very concept of semantic unmixing. We hope this addresses the issue of the reviewer sufficiently well.

4. This work appears to use a total of 24 datasets.

9 datasets, 29 tasks which are derived from them. For the supplement, we have now added one more dataset and a new task to shed more light into one of the limitations of MicroSplit. See supplementary section B.5.

a. This number appears to be substantially small, in comparison to other works. For example, Christiansen et al uses ~200,000 images. The authors should comment on the risk of overfitting to specific training conditions, given the limited number of datasets.

MicroSplit is meant to be trained from scratch for every semantic splitting task, effectively eliminating the risk of overfitting. (If MicroSplit would overfit, the validation loss during training would clearly show this.)

To show how little data is sufficient to train a well working model, we conducted a series of experiments using the HTLIF24 dataset (20ms). We progressively reduced the amount of training data and consistently tested the trained network on the entire validation data. The plot below illustrates the resulting PSNR (Y-axis) as a function of the fraction of training data used (X-axis). The results show that performance plateaus well before utilizing the full training dataset of 90 images of size 1608x1608. However, we note that the amount of data required can vary across tasks, depending on the complexity and variability of the underlying structures.

Additionally, we believe that 24 total tasks to be acquired and shared with the world for this manuscript is a good number we are in fact rather proud of!

b. How do the multiple optical elements affect the network? (e.g. different lenses, different beamsplitters/filters)? How robust is the approach to different optics/instruments? The authors should validate the generalizability of MicroSplit.

As mentioned in our response to 4.a, we have assessed the performance of MicroSplit across multiple microscopy setups without problems. We are actually not entirely sure what the reviewer is suggesting us to do/test. MicroSplit will be trained on data from one optical setup and the trained model later applied to more data from the same setup. In what way will a lens of beamsplitter or other optical element interfere in any undesired way with this approach?

We will add (in Section A), a more detailed description on how data acquisition and training should be conducted to avoid any problems. Since data will be publicly provided, the meta-data will allow readers and future users to look up all acquisition details.

c. How do experimental settings affect MicroSplit? Different pixel sizes/zoom level, different type of detectors, different types of noise (which appear to be ignored in this work) contribute differently to fluorescence microscopy images. The authors should comment or prove effectiveness of the approach under varying experimental settings.

As before, parameters such as pixel size, zoom level, or detector type will change between training and prediction. Hence, none of these properties will make any difference. We have experimented with 9 different datasets, and with two datasets, we have explored multiple noise levels and different mixing ratios, and so we can say that MicroSplit has the ability to perform well across all these setups. That said, we want to reiterate that we do not intend to use the same trained network to be used across all possible configurations. (Please note that this is not a special assumption we take, but something we share with a plethora of other methods used on and with microscopy image data!)

The remark about different types of noise is interesting. MicroSplit’s unsupervised denoising is most effective for pixel noises (noises that are independent per pixel given the signal), such as Gaussian or Poisson noise.

5. Training modes and results:

a. The authors state in reference to training mode I “We generate mixed input images by pixel-wise summation of multiplexed image channels. These input images closely resemble what can later be acquired in a single acquisition” this statement may be incorrect depending on the type of fluorophores, excitation wavelength and emission filters used, owing to the different excitation and emission spectra of fluorophores.

We are not sure we grasp the full extent of the reviewer’s concern. We do agree, the factors mentioned above must be chosen such that training data created by summing multiplexed channels and the data the trained model is later applied do match (note that this test-data can either be acquired with the same fluorescent labels and multiple excitation lasers and filters being applied, or after labeling those structures with the same fluorophore and imaging the superposition truly as a single fluorescent channel). We don’t see how this depends on different excitation and emission spectra of the used fluorophores as the reviewer seems to conclude. As long as the labeled structures in the training and test data end up being roughly in the same relative intensity range (and even this “requirement” can easily be softened by using intensity augmentations during training), MicroSplit can be used without any restrictions. (We believe the reviewer might be misguided by considerations that hold true in the context of spectral unmixing, as we have also pointed out in other locations in this rebuttal.

The puncta removal task is maybe the most “extreme” example where we used Training Mode I. We still see good performance when the network trained using Training Mode I was used on input images acquired in a single channel. Moreover, in the supplementary section C.3, we have quantified the performance degradation when the network gets trained using Training Mode I. We argue that while there is a degradation in performance, the results still look reasonably good to perform downstream tasks.

b. In one of the notes the authors state the “ground truth” used is the “target images” or images acquired by the instrument. These images are intrinsically still spectrally overlapping, hence suggesting the reference image used by the authors is already “spectrally mixed”. The authors should clarify if this was the intent, as these images are not “unmixed”.

We suspect the reviewer might have misunderstood something. For most of the datasets we worked with, one cannot easily see any bleedthrough. Hence, there is in fact no significant “spectral overlap”. The “target images” we talk about that we use as ground truth are the image channels we acquired via temporal multiplexing, including suitable high-, low-, or band-pass filtering so that these channels do not contain overlapping spectra.

For some few tasks, one can indeed see some bleedthrough. However, we emphasize that it does not say anything about MicroSplit. If the target structures will have bleedthrough, the network will be trained to predict structures with bleedthrough.

c. Training mode III seems to have a more realistic acquisition using 3 illuminations and collecting one single channel. These more realistic samples yield the lowest performance of the method presented (Table 1). The authors should acquire more realistic data (e.g. using same fluorophore for 3 targets) and in larger quantity to prove more convincingly the effectiveness of the approach.

This is a misconception that we would like to clarify. The performance is dependent on the complexity of the structures to be unmixed, the similarity of the structures to be unmixed, and the SNR of the data. For example, one can observe task XXVII to have a much higher PSNR than task XXIII. Both these tasks use Training Mode III. Between these two tasks, the only difference is acquisition duration. While task XXIII is using lower-SNR data, task XXVII is conducted on a higher-SNR version of the same data.

d. Most of the datasets used in this work utilize Alexa dyes, which are bright and generally difficult to photobleach when imaged with efficient experimental settings. The authors should clarify this choice as opposed to a broad variety of commonly used genetically encoded fluorophores. Based on supplementary material it seems (though it is not clearly described) that Pavia-P23 dataset uses eGFP. Assuming Pavia-P24 is using the same samples described in Pavia-P23, the results reported in Table 1 for this sample substantially score lower than the Alexa samples. The authors should clarify the performance of the system against less-bright fluorophores or state more clearly the limitation of this approach to bright samples.

MicroSplit's operations do not change with the brightness or photostability of fluorophores, as long as the structures are visible in the training and test data. The final performance does naturally depend on various factors, and SNR is one of them (see also above). The primary reason why SNR and not absolute brightness dictates the quality of results is that MicroSplit operates on normalized input images, meaning that we feed inputs where absolute intensity information is normalized out (note that this is a common thing to do and that denormalization of network predictions are bringing intensities back into the range of the original data).

With Pavia-P24, the issue lies in the fact that target and the input images show soft and “hazy” structures that are not crisp (object outlines visually not “well defined”). As an example, inspect Figure S28 and appreciate that both target channels are “hazy”. When target images are “crisp”, we observe that semantic unmixing tends to lead to better results, even when the network sees only a smaller region of the input. However, when high-frequency details cannot differentiate the two targets, the task at hand becomes more difficult since more spatial context needs to be integrated. If this observation and argument is helpful, we are happy to integrate it, for example, in the discussion section.

6. Segmentation results (Figure 3):

a. Figures 3B and 3C report high DICE scores, but visual overlays show significant discrepancies between predictions and ground truth. Particularly, 3B shows substantial amount of under-prediction or over prediction for A3 and A2 respectively.

We have double checked the DICE formulation, and confirm that it is correct. If x & y are two boolean arrays, we compute dice coefficient as $2 \cdot \text{SUM}(x * y) / (\text{SUM}(X) + \text{SUM}(Y))$. Note that while DICE score is computed over multiple full frames, what is shown is a smaller crop from a single frame. We have now shared full frames, their predictions, and the corresponding code for DICE computation (https://github.com/CAREamics/MicroSplit-reproducibility/tree/rebuttals/examples/segmentation_results), so readers can convince themselves and browse the data and our results in detail by themselves.

b. The reported DICE values do not seem to match the visible alignment between images. It is unclear whether the DICE scores are computed over thresholded regions (positive binary) or across the entire image. The authors should clarify how these values were computed and justify their accuracy.

Thanks for the feedback and suggestions. The requested information is given as part of the previous answer.

7. Improving photon budget: the authors claim “imaging two structures in a single channel can free up photon budget, which then becomes available for imaging at a higher frame rate, imaging at a higher signal-to-noise ratio, or imaging more labeled structures in additional image channels”.

a. What assumption is used for this statement? The imaged channel has a broader spectral band? If labels are different, due to the stochastic emission of fluorescence, shaped as the spectrum of the fluorophore, collecting one channel or collecting multiple channels (if done efficiently) will not change the total number of photons emitted by the fluorophores or detected by the detectors. The authors should clarify and ideally demonstrate this statement.

At least three arguments support our claim that MicroSplit can help to free up photon budget.

(1) In a typical multi-color setup, one must use an emission filter that rejects all wavelength not exclusively (or at least predominantly) used by the fluorescent label currently imaged. However, if a single channel is to be used, then this constraint can be relaxed and one can use a broader emission filter (or even avoid any filtering if all structures are labeled with the same fluorophore). Hence, with the same laser power and exposure duration, one would collect more photons (broader filter).

(2) With a bit of mathematical reasoning, one can see that aggregation of the photons emitted by multiple structures, the overall SNR will improve. Still, even if we assume that the same number of photons as previously collected into two separate channels are now collectively

imaged in one, the SNR of this single channel acquisition is higher. If one chooses to work with superimposed images that are having an SNR similar to individual acquisitions, one has now the freedom to reduce the exposure duration and/or laser power appropriately. Next, we provide the mathematical argument supporting our claim.

Gaussian and Poisson are the two dominant noise sources in microscopy. If X and Y follow the Poisson distribution with parameters a and b , then mean/stdev of $X+Y$ is $(a+b)/\sqrt{a+b} = \sqrt{a+b}$. The average SNR for X and Y are $(\sqrt{a} + \sqrt{b})/2$, which is strictly less than $\sqrt{a+b}$. Note that both $\sqrt{a} < \sqrt{a+b}$ and $\sqrt{b} < \sqrt{a+b}$. So, the above inequality holds. If X and Y follow the Gaussian distribution with parameters (μ_1, σ) and (μ_2, σ) , the SNR of $X+Y$ would be $(\mu_1 + \mu_2)/(\sqrt{2} \cdot \sigma) > (\mu_1/\sigma + \mu_2/\sigma)/2$.

We can see this effect visually as well. In Figure 2.b, for example, compare the noise levels of the target images with the noise level of the input image. The input appears less noisy than the target. (Note that the laser power and acquisition duration was left unchanged when imaging the target and the input.)

(3) Our method, next to splitting structures, also conducts unsupervised denoising. While denoising does not improve the SNR at acquisition time, it certainly does so post-hoc (and thereby makes downstream analysis easier). This allows users to acquire data at lower SNR, and hence, saving photon budget.

To better illustrate this, we have now conducted an analysis which set out to explore MicroSplit's photon efficiency (via denoising). In the table we show below, we use the 3 channel data (Nucleus, Microtubules, Kinetocore) of the 500ms exposure subset of HT-LIF24 data as ground truth. We then took the shorter exposure times data subsets of the same data and computed their similarity to this "ground truth". We do this to contrast the numbers in the table below with the corresponding values for Tasks XXIII-XXVI in Table 1 (main table in main text). For example, for channel 1 (C1), in terms of PSNR, our prediction on 2ms superimposed input (see Task XXIII in Table 1) is better than the C1 obtained directly from the microscope with a 20ms acquisition duration (last row of the table here below). Similar to previous deep-learning based denoising setups like CARE [3], the denoising capability of MicroSplit opens up the possibility of using the available photon budget more effectively.

Acq. Duration	PSNR			MicroMS-SSIM		
	C1	C2	C3	C1	C2	C3
2ms	23.3	25.1	26.1	0.839	0.772	0.869
3ms	23.6	26.0	27.2	0.842	0.780	0.871
5ms	24.6	28.3	29.9	0.857	0.817	0.875
20ms	30.0	35.6	38.16	0.920	0.942	0.914

8. Removing unwanted imaging artifacts: authors state that “any subsequent analysis can be performed with the images predicted by MicroSplit”. The results shown in figure 2 suggest a creation of additional signal in the images as well as a blurring artifact in the prediction. This is echoed by the results in figure 3 and 4 where intensity distributions are seemingly altered in a gaussian blur, altering any intensity-based analysis the predicted images. The data presented by the authors contradicts this statement from the authors. The authors should correct the statement to align with their results.

We would first like to address the claim that our predictions are subject to “Gaussian blur”. This is not the case. Instead, the absence of high-frequency pixel noise makes denoised images appear blurred. We believe that our results show images that are just as blurry as the resolution of the microscope permits. (Note that all MMSE images we show are slightly more blurry since those images also average over the differences between posterior samples, meaning that the data uncertainty is additionally translated into “blurriness”.)

Our sense of downstream applications include segmentation, tracking, detecting presence/absence of some structures, counting the structures, estimating the dimensions of structures (like radius, length, etc.). In all these cases and more, our predictions can be used. However, we would be cautious about using MicroSplit for downstream tasks which are sensitive to minor pixel intensity differences. For example, we would not recommend using MicroSplit to estimate the width of very thin structures, especially in cases where the raw input data is very noisy. Intensity quantifications can, potentially, be conducted on the original raw data. If this is not the case, this kind of downstream processing is indeed not possible. We have clarified this in Section 2.5.

9. In the limitations section the authors suggest that “a higher degree of skew (unequal channel intensities) makes it difficult for MicroSplit to pick up the weaker of the two signals. For users to balance the intensity emissions, the laser power for each fluorophore (in most of the experimental settings described by the authors in supplementary) need to be adjusted. By looking at each emission channel independently. This contradicts the motivation described at the beginning of the manuscript, of collecting a single channel. The authors should clarify this statement.

As the reviewer correctly points out, we quantitatively demonstrate that a higher degree of skew causes MicroSplit predictions to be of lesser quality. We pointed this out as a limitations of our approach. We do, however, not see how this leads to any contradictions. It simply means that in cases where the various structures that are imaged in one go are of vastly different intensities our method should be used with caution because it might fail to correctly predict the dim structure(s).

10. In limitation section the authors state “although we observe blurriness in predictions for the weaker channel in a skewed HT-P23A derived task” this is true also for the results in figure 1, 2, 3, 4, S4, S10, S11, S12, S13, S14, S15, S16, S18, S19, S20. The authors should update their

statement accordingly or explain the visible blurriness that appears in majority of the predicted images.

Please refer to our response to question 8. We believe that the results produced by MicroSplit does not suffer from blurriness to the extent that the utility of the prediction is reduced. PSNR and SSIM values of Table 1 supports our claim quantitatively.

With this being said, even higher PSNR predictions (less blurry) can be achieved by slightly modifying the loss used in Microsplit. Not that this will reduce the trained network's ability to conduct sampling, calibration, and in some cases even the ability to fully denoise. Our default configuration is a general purpose setup which yields useful predictions in all different cases and can be used as out-of-the-box configuration for training MicroSplit. However, if one desires the highest possible PSNR at the expense of sampling, calibration, and denoising, the following small modification in the notebook (01_train.ipynb) will enable that: the uSplit loss needs to be strengthened, in the most extreme case by setting $w=1.0$. Note that the default value used in MicroSplit training is $w=0.9$.

```
1 experiment_params["data_stats"] = data_stats
2
3 # setting up training losses and model config (using default parameters)
4 loss_config = get_loss_config(**experiment_params)
5
6 # set the balance between the musplit and denoisplit loss components.
7 w = 0.99
8 loss_config.musplit_weight = w
9 loss_config.denoisplit_weight = 1 - w
✓ 0.0s
```

11. In limitations, in reference to changing scale of images (Figures S2, S3) authors state that “MicroSplit can successfully unmix the structures even when the smallest nonzero scaling factor is applied”. The results in S2 Magnification factor 1 Prediction Ch1, Ch2 show very evident artifacts. Magnification factor 1.125, Prediction Ch2, cells on the right and bottom side show evident clipping artifacts, similarly in Mag factor 1.25. The authors should correct this statement or clarify the differences.

Semantic unmixing is an ill posed problem from a mathematical point of view. We do not claim otherwise. So, when the problem becomes harder, in this case by reducing the magnification factor closer to 1, the prediction quality deteriorates. However, we believe that the predictions made for even magnification factor 1.125 can be useful for downstream applications like counting, segmentation and tracking. We have now added two more entries for even lower magnification factors (1.0625, 1.03125).

12. The authors propose using a self-referential calibration approach (Figure 2) where the same sample is used to evaluate its own error.

a. The authors should better discuss this approach with respect to tautological validation.

We are afraid we cannot follow how this applies to Figure 2.

What we use to estimate the uncertainty of predictions is the variability between posterior samples of our variational network. The core idea is that a posterior model like ours will generate posterior samples that vary roughly “in line” with how uncertain the network is about its own predictions. (At least in cases where the network is calibrated, which we show empirically for all models we trained.)

b. Would this estimation of calibration only reflect how confident the model is in itself, rather than how accurate it actually is? The results of this calibration appear in contrast with the empirical results shown in most of the figures, reporting a frequent alteration of intensities in the predicted channels respect to the target channels. This change in intensities would considerably alter results of any intensity-based analysis, affecting its reliability.

The calibration methodology employed in this work involves a linear transformation of pixelwise uncertainty estimates derived from multiple posterior samples generated by the network. This transformation aims to align the adjusted pixelwise uncertainty as closely as possible with the actual error. A notable advantage of this approach is that the quality of calibration can be assessed a priori by examining the calibration plot. For instance, in cases such as Figure 2b (last column), where the root mean squared error (RMSE) versus root mean variance (RMV) closely follows the $y = x$ line, the uncertainty estimates are likely well-calibrated with respect to the true error. Conversely, significant deviations from the $y = x$ line serve as a clear indication that the model’s uncertainty estimates should not be relied upon.

It is important to note that uncertainty quantification remains an open challenge, with multiple competing methodologies often presenting contradictory perspectives. For further discussion on this topic, the reviewer may refer to [1] (reference provided at the end). While the proposed approach may not be universally applicable, its utility lies in the ability to determine—through simple inspection of the calibration plot—whether the uncertainty estimates can be trusted in a given scenario.

13. Quantification methods used by the authors seem to not match visual assessment. Example, the maps shown in Figure 3 “overlays” suggest a much lower visual performance than the DICE score reported. This may be due to the averaging over the entire image (mostly dark), instead of the regions with information content. Example in task 1, Analyst 3 overlay shows a substantial under-prediction, while the DICE score shows over .75. The authors could show more maps of estimated error and correlate those better with the single-value results (e.g. MicroSSIM or DICE) to better prove the method and increase the robustness of the presentation.

As mentioned in our response to the the Question 6a, we have double checked our numbers and they are correct. Still, we have now made raw data, predictions, and segmentation outputs public together with an analysis notebook for everyone to work and play with. The notebook is

hosted at https://github.com/CAREamics/MicroSplit-reproducibility/tree/rebuttals/examples/segmentation_results.

14. The differences between the combination of μ Split with denoiSplit and the current method are not well described. The authors should better describe MicroSplit with respect to μ Split and denoiSplit to position it as a new method rather than an incremental improvement.

In Section “A Model Architecture and Training” we have gone into the technical details of how MicroSplit has technical advancements over μ Split and denoiSplit. We briefly enumerate them here. (a) μ Split was GPU-efficient and tailored for high-SNR datasets. denoiSplit offered denoising and uncertainty estimation. MicroSplit is GPU-efficient and at the same time performs denoising and uncertainty estimation. To enable this, we needed to modify the KL-divergence loss formulation of denoiSplit. (b) μ Split and denoiSplit worked with 2D images but with MicroSplit, we can now work with 3D volumes. It not only speeds up the inference, it also yields superior performance (>3.5db PSNR). (c) The loss formulation used in MicroSplit enables a way to control between preserving high-frequency details and getting better denoising and uncertainty estimation with just one hyper-parameter. See section A.1 for more details. Additionally we have added an efficient 3D implementation and have applied MicroSplit to 24 total tasks, describe an artifact removal task for the very first time, and showcase downstream segmentation advantages. Last but not least: in this manuscript we are clearly targeting biologists and microscopists and not, as before, ML experts. For this reason we have also gone through extensive efforts to make the code-base clean and usable (e.g. via the reproducibility notebooks).

15. Presentation and readability of the manuscript

a. Text clarity and accessibility should be considerably improved. The manuscript seems to be written for a highly technical journal, more in line with ML style, requiring readers to refer to multiple external sources to fully grasp the methodology and prior work, necessitating significant background reading, which may limit accessibility for a broader audience. The authors may consider refining the presentation to improve clarity and readability, particularly if aiming for a journal like Nature Methods, which caters to a diverse readership spanning different scientific disciplines.

We thank the reviewer for the comment. While we consciously avoided technical details in the main manuscript and relegated them to the supplement, we have gone through both the main manuscript and the supplement and have adjusted the content in-line with all explicit feedback from reviewers. We remain open to further suggestions for how to improve the main text and supplement, either by the reviewers or the editorial team.

b. Regarding data presentation and organization: the way results are reported deviates from common conventions for this type of journal, making interpretation more challenging. For example, Table 1 presents multiple tasks and performance metrics in a highly condensed

format, requiring cross-referencing with the additional supplemental material to fully understand the comparisons and tasks performed. Error measurements for this same table are reported in a separate table (Supplementary Table ST9), making it difficult to quickly locate and interpret key findings. A more structured and intuitive layout (or plot) could enhance the readability and accessibility of the data.

Also here we are open for concrete and actionable feedback from the editorial team and all reviewers.

c. Dataset information: the description of the datasets used in the study could be more clearly structured. While the manuscript states that datasets are shared, which is great, a comprehensive summary table detailing dataset characteristics, such as image size, pixel resolution, microscopy settings, bit depth, and time points, would improve transparency. Some of this information is included for certain datasets, but for others, it is either difficult to find or not explicitly stated. Providing a standardized dataset summary would make it easier for readers to assess the scope and applicability of the study.

We thank the reviewer for this suggestion. We have added the table ST8 in supplement. Among other things, the table also contains the column Pixel count, that represents the total number of pixels (across training, validation, and test sets) for each task with 1 million as the unit. Since a 1000×1000 sized image corresponds to 1M pixels, so, Pixel Count column serves as a ballpark estimate of the number of images needed for training MicroSplit.

Reviewer #1 (Remarks on code availability):

The MicroSplit code is available and includes installation instructions, but it lacks explicit examples on how to apply the method to real data. While the provided documentation describes how to set up the environment, there are no practical examples demonstrating how to run the model on fluorescence microscopy images. This structure caters more toward a computer science-oriented audience, making it less accessible to researchers without a good programming background. For those familiar with machine learning workflows, the code is understandable, but a step-by-step guide with example datasets would greatly improve and facilitate usability and reproducibility. Enhancing the documentation with practical use cases would make the resource more accessible to a broader community. A simple UI would also facilitate adoption.

We have already worked on a more comprehensive set of notebooks and example code when we submitted this manuscript the first time. After reading this feedback we decided to delay the re-submission until we finished this work (originally we hoped this could be done in parallel to reviewing, revising, and other steps towards publication).

More example notebooks and, most importantly, generic “use your own data” examples in 2D and 3D can now be found here:

<https://github.com/CAREamics/MicroSplit-reproducibility/tree/rebuttals/examples>

Reviewer #2 (Remarks to the Author):

Summary of the Key Results:

The authors present MicroSplit, a deep-learning-based method for computational multiplexing in fluorescence microscopy. This technique enables multiple cellular structures to be imaged within a single fluorescent channel and subsequently unmixed computationally, reducing photon exposure while maintaining imaging quality. MicroSplit employs a variational splitting encoder-decoder (VSE) network, leveraging posterior sampling to estimate prediction uncertainty. The manuscript demonstrates that MicroSplit can effectively separate up to four superimposed structures, denoise images, and provide uncertainty quantification at a quality level suitable for downstream tasks.

Originality and Significance:

While the concept of computational multiplexing is not new, with prior work such as PICASSO addressing similar problems, MicroSplit distinguishes itself by requiring only a single superimposed image as input. This eliminates the need for spectral overlap measurements, which presents a significant advantage. The supplemental material compares MicroSplit to PICASSO effectively, though moving this comparison into the main text would strengthen the manuscript. Overall, the method has the potential for broad adoption and significant impact in fluorescence imaging.

Data & Methodology:

The channel segmentation methodology is well-reasoned, but the Error Estimation, Uncertainty, and Calibration section requires substantial revision and clarification. The authors state: "We therefore estimated the true error by computing the pixel error between the MMSE prediction of MicroSplit and the ground truth target images we derived from the available training data, and plotted this 'true error' against the RMSE errors we described above."

We apologize for the mistake in the above statement. The correct statement would have been the following: "We therefore generate the true error (RMSE) by computing the pixel wise error between the MMSE prediction of MicroSplit and the ground truth target images we derived from the available training data, and plotted this 'true error' against the scaled RMV (estimated RMSE errors) we described above."

However, the calibration curve axes are labeled RMV vs. RMSE, yet RMV is not explicitly defined in the manuscript, making interpretation difficult. Additionally, the claim that RMSE can be calculated solely from posterior samples suggests that the authors are computing RMSE between the posterior samples and the posterior MMSE image. Since RMSE can be expressed as the square root of the sum of variance and squared bias, and given that the MMSE image is an unbiased estimator, pixel-wise RMSE should correspond to the standard deviation of the posterior. RMV represents a similar metric and would be expected to exhibit a nearly linear correlation with RMSE.

We adopted the uncertainty and calibration methodology from denoiSplit. Here, we would like to clarify the terms RMV and RMSE. We added supplementary section D that explains it in sufficient detail. Pixelwise squared error is computed between the groundtruth image pixels and MMSE image pixels. Next, we compute the pixelwise variance in the prediction using different posterior samples. We can do it because, for one pixel location, each posterior sample has possibly a different value. We then aggregate these pixelwise variances into 30 bins to yield 30 RMV (root mean variance) values, one value per bin. For these bins we aggregate the squared error to get RMSE (root mean squared error) values. Using two scalars per channel, one as the multiplicative factor and other as the offset, we scale the RMV values to optimally match RMSE.

If the goal is to demonstrate that low posterior variance correlates with low error and can serve as a proxy for uncertainty, I would recommend the following modifications:

1. Increase the number of posterior samples beyond 50 to allow for a more robust estimation of pixel-wise variance.
2. Compute the pixel-wise ‘true error’ using the posterior MMSE image and plot it against the posterior variance, providing a quantitative correlation metric.
3. In Figure 2, replace the Sample 1 - Sample 2 heat map with a posterior variance heat map, and replace the RMSE column with the true error heat map, enabling readers to visually assess the relationship between variance and error.
4. Explicitly provide equations for all calculations related to calibration and uncertainty estimation.

Partly due to the unclear explanation given in the main manuscript, there was some misconception about how our uncertainty and calibration module works, which apparently led to these suggestions. With adequate explanation, we believe that the reviewer would be satisfied with what we have done. We have covered suggestion 2. and 4 in our previous answer. We picked the number 50 for posterior samples because we did not find a significant change beyond 50.

Validity of Approach, Quality of Data, and Presentation:

The methodology is generally well-structured, with the exception of the Error Estimation, Uncertainty, and Calibration section. Several figures require revisions for clarity:

- Figure 1.A: The axis labels are difficult to read, and a legend is necessary. Additionally, the figure caption refers to emission filters, whereas the main text describes them as excitation filters—this discrepancy should be resolved.
- Figure 2.A: The second lateral context patch should be explicitly labeled for clarity.
- Figure 2.B: The term RMV is undefined in both the calibration plots and the main text, making interpretation challenging. See further comments above
- Supplemental Figure 5: The x-axis should have consistent spacing, either in linear or logarithmic scale, or alternatively, the figure could be reformatted as a bar graph for improved readability.

We thank the reviewer for these concrete suggestions. Figure 1.A is a schematic representation, and not a qualitative figure (additionally, we do not see any mention of excitation filters). Hence the axis labels and the legend are not needed and we will remove them to give the figure a cartoonish flavor. We have made all the other suggested changes (Updated Figure 2A, updated caption for Figure 2B and updated supplemental Figure 5). For RMV definition, we provided an intuitive definition in the caption and have added section D in the supplement that explains the uncertainty quantification and calibration methodology in detail.

Regarding data quality, the authors have taken care to collect and describe three distinct datasets. However, the manuscript's clarity could be improved by more precisely distinguishing between traditional sequential multiplexing (multichannel acquisition) and the proposed simultaneous multiplexing (multiple structures within a single channel). Introducing explicit terminology for these concepts would reduce ambiguity.

References:

The manuscript cites an adequate body of prior work, and I did not identify any major missing references. However, the claim that 32.53 PSNR and 0.886 MicroMS-SSIM are sufficient for downstream analysis would benefit from additional supporting literature.

The validity of this claim is supported by prior works in supervised and unsupervised denoising [2,3,4], which report PSNR and SSIM values within a comparable range. However, it is important to note a caveat regarding PSNR: its magnitude is influenced by the intensity range of the ground-truth data. Specifically, ground-truth images with higher pixel values generally yield higher PSNR values. Nevertheless, the referenced denoising methods remain relevant for comparison, as they were also evaluated on microscopy data and achieved similarly competitive PSNR results.

Final Recommendation:

The manuscript presents a promising computational multiplexing method with significant potential impact. However, revisions are necessary, particularly in the Error Estimation, Uncertainty, and Calibration section, as well as in figure clarity and definitions. I recommend major revisions to improve methodological transparency, ensure rigorous uncertainty quantification, and enhance clarity in data presentation. With these improvements, MicroSplit has the potential to contribute meaningfully to the field of fluorescence microscopy.

- Eric Markley

Reviewer #3 (Remarks to the Author):

MicroSplit combines the advantage of uSplit and denoiSplit, also with extension to volumetric images, to tackle the semantic unmixing problem for fluorescence microscopy. Overall, this work is both theoretically novel and practically useful, supported not only by experiments on extensive and diverse datasets / applications, but also by comprehensive discussion regarding the performance, caveats, practical considerations, etc. I would definitely endorse for publication, but with some minor suggestions.

A few strengths I really want to highlight:

- ***Supper flexible training settings (different modes in Section 2.1).***
I think this is really useful in practice, since different applications / use cases may have different data availability and the proposed framework can still be applicable in a lot of different situations. Importantly, comprehensive discussion and caveats are provided for better understanding of different training modes. This will serve as a useful guide for users to design their assay accordingly for their needs.
- ***The variational model as the backend supports informative error estimation.***
I really appreciate that the variational model backend, which fundamentally supports error estimations. The calibration step confirms the effectiveness. The error estimation and calibration make this method, not just something looking cool, but equipped with a handle for users to take full control of the trustworthy, which is important for deep learning based image predictions.
- ***Evaluation beyond PSNR and SSIM.***
By showing MicroMS-SSIM, the evaluation already provides a comprehensive view from an image perspective. But, it is still not enough to represent its impact on downstream analysis or its impact for answering the underlying biological questions. I really appreciate the additional experiments in Section 2.4, which provides application-appropriate validation to make sure the prediction is biologically valid for the specific application.

Besides the highlight above, I can confirm, to the best of my knowledge, the described methods are sound, the quality of presentation is good, easy to understand, and the use of statistics / evaluation metrics are appropriate. There are only some minor suggestions:

(1) Section A in Supp. can be significantly improved.

In general, the method proposed in this work combines previous work in uSplit and denoiSplit into a single setup. But, it is not quite clear how exactly this “final product” looks. In theory, I could check the previous publications for details. But, this may not be straightforward, since this “final product” is not a simple copy-and-paste from previous works. Therefore, I would highly recommend to provide more details in the Supp. so that, in theory, people should be able to reproduce the exact method based on the description in this paper. Here are some examples of what are not quite clear to me in the technical details.

We thank the reviewer for the suggestion. To this end, we have added content in Supplementary Section A.

A. In Supp. page 1, “In Figure S1, we present the architecture of MicroSplit, which has, LC inputs and Noise models, all integrated into a single setup.” But, when you look at Figure S1, it is not clear at all, where the Noise model is and how it is integrated exactly.

In Figure S1, we have focused on the Encoder-decoder aspect of our setup which was depicted in Figure 2a. The noise models are used in the loss computation and therefore do not form part of the encoder-decoder architecture. We have rephrased the caption of S1 so that it is clear.

B. In Figure S1, what is the “Input Branch”? Any computation within the “Input Branch”? I assume it is just some kind of connection in the neural network, correct?

The reviewer has assessed correctly. It is essentially a series of neural network layers. We will mention this in the Supplement A.

C. Also in Figure S1, the white squares in the network figure are getting smaller through layers, is it the result of pooling or unpadding convolution? If it is pooling, then the size is not halved.

As the reviewer predicts, it is indeed the case that the white squares get smaller due to pooling. White squares represent the content coming from the “primary input patch”. Similar to what happens in a U-Net, the spatial dimensions reduce by a factor of 2 at each consecutive hierarchy level due to a pooling operation. However, in MicroSplit, the resultant embedding is zero-padded after pooling and is subsequently concatenated with the embedding coming from one of the lateral contextualization (LC) inputs. Since the embedding coming from the LC input does not undergo pooling, the final embedding has the same spatial dimensions (gray squares) at each hierarchy level. We have explain this in more detail in the Supplementary Section A.

D. Again in Figure S1, the first part of all three types of models are three white squares. Are these three consecutive convolution layers? Or something else?

E. In Figure S1, there are center crops in Lean-LC. Is there anything similar in Regular-LC and Deep-LC?

No, there is no center-cropping applied to Regular-LC and Deep-LC on the output of the encoder. Their decoder takes in the entire uncropped embedding which contains the full information from the primary input and all the LC inputs. This lack of center-cropping is exactly what differentiates Lean-LC with Regular-LC and Deep-LC. However, as the reviewer may know, in the U-Net, while the encoder has pooling layers which do downsampling, the decoder has layers to perform upsampling. This combination of pooling and upsampling results in the final output having the same spatial dimensions as the original input. In the setup for Deep-LC and Regular-LC, since the size of the embeddings coming out from the encoder does not reduce, we centercropped the resultant embedding after each upsampling in the decoder.

I know a lot of these details can be discovered from the released code (great! Open-source is important!), but it is still good to explain the technical details in the paper as a top-tier publication.

Note: the missing details mentioned above did not affect my understanding of the key idea of the method.

(2) Is there a discussion on why “up to four structures”? I am not sure if I miss this somewhere in the paper. But, in general, what is the biggest trouble if we make an attempt on five or six structures? If we use Deep-LC or even a Deeper-LC, will five or six structures become possible?

While there is no theoretical limitation to our approach to be extended beyond four structures, there are both practical limitations and the limitations from the learning side. With most off-the-shelf microscopy solutions it is relatively easy to image 1-4 fluorescent channels. Beyond that, filter sets and enabling hardware are not commonly available (mainly for live cell imaging where multiple rounds of imaging are not an option) and even the imaged sample must be permissive to the required labeling and light exposure. Hence, Training Mode I and III are not directly permitting to train a network to split more than 4 structures. Beyond this practical limitation, as we increase the number of structures by other means, at some point the quality of predictions will likely drop (see Supplementary Section B.3), and for this work we chose to opt for applicability above pushing the limitations of the presented method. Additionally, even with the presented four channel splitting, the theoretical limitation on off-the-shelf microscopes lies at four parallel 4-channel semantic unmixing tasks, hence at 16 biological structures — undoubtedly a huge advantage for suitable projects where such a strategy would apply.

(3) Will the model be released on Bioimage Model Zoo? I don't think this will affect the evaluation of this paper. This is more of my own wish from a user's perspective.

Yes, we will upload the models to the Bioimage Model Zoo.

(4) There are some other really minor things in terms of presentation. You could even ignore this part.

- In Table 1, it would be better to use 1, 2, 3, ... or T1, T2, T3, ... as indices for tasks, rather than I, II, III.
- Would it be possible to provide a vision overview of all different training modes? I can appreciate the high flexibility of the propose method. A summary table or some kind of visual abstract / overview would be really helpful.
- Personally, I would really appreciate a few sentences (maybe in the Supp.) describing how the variational models allow error estimation using plain English. I believe these few sentences would be super helpful for the users without deep knowledge of machine learning to understand where this error comes from.

We thank the reviewer for the above mentioned suggestions. We added a supplementary Table ST12 giving an overview of the different training modes, which we also refer to from the main manuscript. We also added a new Supplementary Section D, where we describe in detail how

error estimation is performed. We have added an introductory paragraph which explains the overall idea without getting technical.

Reviewer #3 (Remarks on code availability):

I didn't run the code, but I read the code on Github, very clearly written scripts and software. Currently, the code is organized in a way focusing on reproducing the results in this paper. This is fair. But, from a user's perspective, I would like to see a clear instruction on what to do if I want to run training mode 1, training mode 2, etc., with my own data, and how should I organize my own data (2D or 3D). I have no doubt the documentation will get better.

We have already worked on a more comprehensive set of notebooks and example code when we submitted this manuscript the first time. After reading this feedback we decided to delay the re-submission until we finished this work (originally we hoped this could be done in parallel to reviewing, revising, and other steps towards publication).

More example notebooks and, most importantly, generic “use your own data” examples in 2D and 3D can now be found here: <https://github.com/CAREamics/MicroSplit-reproducibility/tree/rebuttals/examples>.

References:

[1] Reexamining the Aleatoric and Epistemic Uncertainty Dichotomy | ICLR Blogposts 2025. <https://iclr-blogposts.github.io/2025/blog/reexamining-the-aleatoric-and-epistemic-uncertainty-dichotomy/>. Accessed 4 June 2025.

[2] Weigert, Martin, et al. "Content-Aware Image Restoration: Pushing the Limits of Fluorescence Microscopy." *Nature Methods*, vol. 15, no. 12, Dec. 2018, pp. 1090–97. www.nature.com, <https://doi.org/10.1038/s41592-018-0216-7>.

[3] Prakash, Mangal, et al. Interpretable Unsupervised Diversity Denoising and Artefact Removal, ICLR 2022. [arXiv.org](https://arxiv.org), <https://openreview.net/forum?id=DfMqIB0PXjM>.

Latest round of Responses to Reviewers

(Note: comments to first rounds of reviews can be found below this section)

Our response to Reviewers 1 and 2 (see below);

We have now integrated all changes we addressing the remaining reviewer comments (as we discussed previously via email).

1. Clarification on dataset-specific training

We fully agree with the reviewer that the current manuscript could communicate more clearly that MicroSplit models are trained separately for each dataset or experimental setup. We now explicitly state this in several places from abstract to methods.

2. Photon efficiency quantification

We appreciate the reviewer’s suggestion to quantify photon-budget savings. We have now included quantitative estimates of the photon-efficiency gains across representative imaging setups (Extended Data Figure 7) and discuss/mention this results in the main manuscript and the supplement.

3. Loss of high-frequency details and applicability to downstream tasks

We agree that the discussion of high-frequency content can be made more precise. We now make it explicit in the main text, that MicroSplit—like all deep learning models today—tends to smooth fine structural details, mostly when input SNR is low. We also cite the perception–distortion trade-off work that will be the right place for interested readers to deepen their knowledge about this aspect of AI.

Minor points (4–6) raised by Reviewer 1

All minor corrections (grammar, terminology consistency, dataset naming) have been implemented as suggested to the best of our abilities.

Overall, we are grateful for these constructive remarks of our reviewers. They pointed us at several actionable and meaningful improvements that now make the paper clearer and more transparent for readers.

Reviewer #1 (Remarks to the Author):

The authors provided extensive responses, often detailed and technical, and I commend them for their work. They clearly invested effort into explaining their rationale, correcting terminology, and adding supplementary material. The expanded Limitations section is an improvement on

the clarity of the technique, though it lacks some fundamental points made by authors in rebuttal, which would greatly benefit readers.

Some of the responses shift the burden of proof back onto the reviewers (e.g., suggesting that some objections are “misconceptions” and that readers/reviewers can find information in the metadata of images). Typically, such reviewer concerns would motivate concrete improvements to the manuscript itself, yet in this case many of these points remain unresolved in the main text. This style weakens the persuasiveness of the rebuttal.

For example, reviewer comments on figures remain somewhat unaddressed, despite multiple reviewers pointing to unclear representations (e.g. Figure 1a, which typically aligns with the experimental settings, the figure could be updated or it should be made clear in the caption that these are cartoon images and not actual experimental settings).

A well written manuscript would provide, in a self-contained setting, all essential information for a clear, not-misguided interpretation of the work. Manuscript clarity improvements rest with the authors, which can refer to the large literature in Nature Methods publications for practical examples on how to present methods and results for the broad readership of this journal.

The manuscript could, with proper improvements and proper address of reviewers’ concerns, be suitable for publication.

Major concerns:

1. A central limitation of MicroSplit is that the network must be trained from scratch for each dataset or experiment, rather than functioning as a broadly generalizable model. This is stated unambiguously in the rebuttal: “MicroSplit is meant to be trained from scratch for every semantic splitting task, effectively eliminating the risk of overfitting” and “MicroSplit will be trained on data from one optical setup and the trained model later applied to more data from the same setup.” These clarifications make it evident that the approach is task-specific and dataset-specific.

However, the main manuscript does not present this point with sufficient clarity. In the current version, this limitation is only implicit in the training sections (e.g., where the authors describe “each splitting task” or report results on a per-dataset basis). By contrast, the abstract states: “We demonstrate the robustness of MicroSplit across various datasets and noise levels and show its utility to image more, to image faster, and to improve downstream analysis.” This phrasing could easily be interpreted as implying that a single generalized network was trained and applied across multiple datasets, which is not the case. Readers may therefore be misled regarding the scope and generalizability of the approach.

To address this issue, the authors should clearly state in the main text, and explicitly in both the Abstract and the Methods/Approach/Limitations sections, that MicroSplit must be trained separately for each dataset/task. This clarification is important for text clarity and will prevent readers from misunderstanding the intended scope/application of the method.

2. With regard to photobleaching and photon efficiency. The authors rebuttal helps understanding the reasoning behind this claim, provide a valid theoretical explanation.

It is clear that, a broad band emission detection will yield a higher photon throughput compared to a narrow band (or sum of narrow bands). This fact however, based on the statements of authors, is independent of MicroSplit. A photon-efficient experimental setting would minimize the emission detection gap between channels, already improving the photon efficiency.

Based on imaging settings reported, a sufficient empirical demonstration is currently reported as HT-LIF24 (ST12) the authors should quantify the improvement (e.g. x-fold based on the split acquisition) in support of the method.

3. The HT-LIF24 data reports (ST12) and Fig s59 are quite helpful. Impressive how the Exposure = 2ms in Prediction Ch3 and corresponding target image (S59) have a MicroMS-SSIM of 0.869 (Table ST12) despite losing almost all high-frequency details, as reported by the authors and as evident by visual inspection. This is a known challenge of SSIM that accounts primarily for global structural statistics and may explain the intensity mismatches reported in the work. For the scope of the MicroSplit downstream applications (“segmentation, tracking, detecting the presence or absence of certain structures, counting structures, and estimating dimensional properties such as radius or length of structures of interest”) this may be ok, provided that high-frequency details do not compose the structures being counted. The authors updated the limitations section to state “However, while not entirely impossible, caution is advised when applying MicroSplit to downstream tasks that rely on precise pixel intensities.”, which is a useful statement, though still a little broad.

I would suggest clarifying this section to specify that high-frequency features (for example those in S59 Prediction Ch3 2ms, 3ms and 5ms) will provide high amount of error in tracking, counting, estimating dimensions and similar downstream applications, narrowing down the application of MicroSplit to more reasonable scenarios.

Minor comments:

4. Spell check “input branch does not has pooling”
5. Figure S2, S3 “Unmixing similar.. “ should be “Semantic unmixing of similar.. “ to keep consistency with updated terminology in the rest of the manuscript
6. HT-LIF24 is both referenced to as HTLIF24 and HT-LIF24, pick one for consistency

Remarks on code availability:

The code appears appropriate, the descriptions (README and in code) provided for installation are sufficient to reproduce the work with examples on how to run the code.

Reviewer #3 (Remarks to the Author):

The authors have addressed all my concerns. Great work!

Remarks on code availability:

the code is significantly improved.

First round of Responses to Reviewers

We want to start this lengthy document by **thanking the reviewers** for their time and diligent work. We understand you are reviewing our work instead of doing your own, and we truly appreciate the depth at which you dove into our new method and the results it produces.

Below, we respond directly to all your feedback. To make it easier to read this document and also be reminded to your own opinion about our work, we decided to display the `unedited reviewer text` in a dark gray, while giving `our responses` a slightly blue color.

Before addressing the reviewers' specific comments, we would like to highlight a few updates we have made to the manuscript on our own initiative, which were not prompted by the reviewers' suggestions.

We have revised the results of our 3D tasks. While the older results were not incorrect, however, a few hyper-parameters used in their training were specifically tuned for the reported task. Since we do not want to optimally pick the hyper-parameters for each task, but instead are committed to be using **a single set of hyper-parameter across all tasks** of one type, *i.e.* all 3D tasks, and all 2D tasks¹, we have rerun those experiments and updated the reported quantitative evaluations. More specifically, we have updated:

- Table 1, Task I: The older result was achieved when using 9 consecutive frames from the z-stack as input. However, all other 3D semantic unmixing tasks used 5 consecutive frames from the z-stack as input. In Supplementary Table ST3, for completeness, we already did investigate how the performance improves with more z-stack elements as input.
- Table 1, Task XXI and XXII: The older result for these tasks were achieved when using LC inputs. To enable our setup to work also on GPUs with lower memory, we had decided to disable LC for 3D tasks. Still, for these two tasks (XXI and XXII), we had LC turned on by mistake.

We have also **added a new dataset called HHMI-D25**, which has three sub-datasets, from which we derive a new and quite hard task XXIII (see Supplementary Figure S56 and S60). We use this task as an example for how users can debug and fix difficult semantic unmixing tasks (see Section 2.5 and Supplementary Section B.1.1, and the respective Tasks XXXI to XXXVI which can maybe best be compared quantitatively in Supplementary Table ST1).

Additionally, we have also **expanded on an analysis and discussion of the role of SNR in the input data and this influences the performance of semantic splitting**. To this end we used the HT-LIF24 and HHMI-D25 datasets (see Supplementary Section B.1).

Below we comment directly to the reviewer comments we have received.

¹ Except tasks from the SIM dataset Chicago-Sch23, for which the noise properties are quite different from all other datasets.

Reviewer #1 (Remarks to the Author):

In the manuscript “MicroSplit: Semantic Unmixing of Fluorescent Microscopy Data” from Ashesh et al, the authors describe a computational approach aimed at overcoming the limitations of fluorescence multiplexing by predicting multiple fluorescently labeled structures from a reduced number of imaging channels. The proposed method seeks to address issues related to overlapping fluorophore excitation spectra, reducing the number of required exposures in multiplex acquisitions, and overcoming the limits of maximum sampling frequency in live cell imaging scenarios. Their approach builds upon two of their previous publications and introduces a volumetric image processing framework that enables the calibration of the network using the data itself while estimating prediction errors.

We would like to clarify that the volumetric image processing framework does not enable calibration of the network as inferred by the reviewer. Volumetric image processing helps improve the performance and reduces prediction time. We adopted the calibration procedure we have first described in denoiSplit [7], which is a technical precursor to this work.

The claimed novelty:

1. Combining elements from two prior computational approaches (μ Split and denoiSplit) into a single framework, leveraging machine learning to address fluorescence multiplexing challenges.
2. Processing volumetric image data using a highly optimized network architecture designed specifically for 3D fluorescence imaging.
3. Introducing a self-calibration procedure that assesses the quality of the trained network and estimates prediction errors, enhancing reliability in fluorescence reconstruction.

Claim 1 is only referenced in the manuscript, not supported by any practical demonstration, code is made available for readers to review.

We thank the reviewer for bringing to our attention that this was not well articulated in the submitted manuscript. We have updated the supplementary section A. Our line of argument is that both our previous works denoiSplit and μ Split, aimed for a Machine Learning audience, had complementary advantages. μ Split was GPU efficient and provided multiple meta-architecture variations, allowing the user to pick among the three μ Split variants, with each variant striking a different balance between performance and GPU utilization. denoiSplit did not have this feature, but it enabled denoising, posterior sampling, uncertainty quantification and calibration. In the creation of MicroSplit, we adopted the architecture design from μ Split, which makes it GPU efficient and allowed multiple architecture variants, and from denoiSplit adopted KL loss formulation, noise model integration, uncertainty quantification and calibration. Hence, we ended up with a GPU efficiency method that also conducts unsupervised denoising and can do all this also volumetrically (in 3 spatial dimensions) due to a new implementation we added in this work. With results presented in this manuscript, one can verify that MicroSplit performs unsupervised denoising and supports posterior sampling, uncertainty quantification and calibration. In our tutorial notebooks, we are describing that Deep-LC is the default configuration

and will, once fully implemented, describe how the user can switch between the three μ Split variants.

Claim 2 is addressed by mathematical demonstrations in supporting material but not associated with any explicit timing of analysis of 3d fluorescence imaging dataset.

We want to state that processing volumetric data is not about the inference speed. It is about performance. Performance improves when working on volumetric data. See Supplementary Table ST3.

Claim 3 is presented with multiple datasets and plots across the manuscript, with some supporting material.

The manuscript, in its current version, is framed to present an alternative to standard unmixing techniques (i.e. spectral), arguing that MicroSplit achieves efficient reconstruction of fluorescent structures through learned feature extraction rather than intensity-based spectral decomposition. The authors suggest that this enables more photon-efficient multiplex imaging, reducing phototoxicity and improving imaging speed.

The proposed method is designed to address these fluorescence imaging challenges, however, similar problems have been explored in similar studies and/or commercially, using alternative approaches:

1. Machine learning-based fluorescence prediction in both 2D and 3D, such as the work from Guo et al (DOI: 10.7554/eLife.55502) , that of Ounkomol et al (DOI: 10.1038/s41592-018-0111-2) , Wang et al (DOI: 10.1038/s41592-018-0239-0) , Christiansen et al (DOI: 10.1016/j.cell.2018.03.040) to name some, though with recent machine learning improvements the count of applications of machine learning articles focusing on fluorescence prediction is substantially larger

While all of the four mentioned works indeed use machine learning, the tasks they tackle are different in nature to the one we are addressing. One of these works performs a super-resolution task, while the other three perform virtual-staining. In virtual staining, the prediction objective is superficially somewhat similar to our objective: predict (multiple) fluorescence structures (channels) from a given input image. This input, however, is a label-free image, e.g. a brightfield image. With MicroSplit, instead, we propose to split superimposed fluorescent channels (note the pixel-wise sum of our predicted channels is in theory the given input image (modulo noise) — a fact that is by no means true for virtual staining approaches). Since the data itself is different, it is not easily possible to compare our method with virtual staining methods. Additionally, it would not be a fair comparison. We address virtual staining methods in more detail in our response to the question 1b.

2. Commercial systems incorporating spectral unmixing methods are already widely used in fluorescence imaging and can be found in commonly available instruments from Zeiss, Leica, Nikon and Olympus.

While spectral unmixing methods indeed perform image decomposition, our methodology is different from spectral unmixing and is, can, and should also be applied when data is imaged on camera-based systems. Spectral unmixing requires multiple image channels as input, with each image capturing a different frequency band of the overall emission spectrum: that is, it requires dedicated and expensive white lasers and spectral filters/detectors. Instead, our approach works with any standard fluorescence microscope and can, therefore, be immediately used by any lab across the globe.

A key difference between MicroSplit and these prior methods is that the latter approaches often leverage real intensity distributions to calibrate predictions, whereas MicroSplit primarily focuses on structural reconstruction without maintaining intensity fidelity.

Virtual staining approaches predict fluorescence from an entirely different imaging modality and predicted fluorescent intensities are arguably only as trustworthy as the quantity of the biological structures in question can be predicted from the input modality itself (which is often barely possible, leading to untrustworthy and imprecise predictions). MicroSplit, on the other hand, is taking the given input and splitting it into its individual “summands”. Spectral unmixing is a different computational task that naturally has its place in microscopy and in biological imaging. We believe that the same is true for MicroSplit, which can easily be used on off-the-shelf camera-based systems (spectral imaging can not) and in cases where virtual staining will lead to uncertain predictions (see also response to 1b, below).

If the claims stated by the authors are confirmed, the work may represent an advance in the application of machine learning to fluorescence imaging. However, fundamental concerns remain regarding the practical advantages of this method.

The study lacks strong experimental validation to support its claims, particularly regarding photon efficiency and phototoxicity reduction. The multiplexing scenario presented is relatively simple (3-4 channels, 3-4 labels), limiting its applicability to more complex imaging tasks. Additionally, the dataset size is small (24 datasets), raising concerns about overfitting and generalizability compared to larger-scale studies. The terminology surrounding “unmixing” is inconsistent, as MicroSplit does not perform true spectral unmixing but rather predicts structures without preserving intensity relationships.

We have fixed the nomenclature (see response to point 3). Here we want to make an important distinction between “spectral unmixing”, an important technique and method in the imaging field, from the “semantic unmixing” we are proposing. They are complementary techniques that can help microscopists and biologists in different use-cases. We will iterate on this also in other responses below.

The presentation is highly technical, requiring extensive cross-referencing of prior work and supplementary materials, making it difficult to follow. There are also concerns with performance validation, as reported DICE scores do not align well with visual assessments, and potential blurring artifacts may compromise downstream image analysis. While the method may have potential in fluorescence imaging, its motivation and practical advantages over existing techniques remain unclear.

I ask the authors to address the following concerns.

Major comments

1. The primary motivation of this work is to overcome challenges in fluorescence multiplexing, reducing the number of required exposures in multiplex acquisitions, and overcoming the limits of maximum sampling frequency in live cell imaging scenarios.

a. In the definition used in this manuscript, a three-channel fluorescence microscopy dataset is considered multiplexing. Three-channel/three-labels fluorescence is a rather simplistic multiplexing scenario. The authors should clarify why such simple scenario is considered here, as opposed to some more complex scenario (10-15 labels).

Even imaging just two fluorescent channels typically requires multiplexing, and surpassing four channels becomes challenging without multiple rounds of introducing and removing fluorescent markers. This limitation is especially pronounced in live-cell imaging. As detailed in Supplementary Section B, the difficulty depends on several factors. In Section B.3 ("Similarity of Structures to Be Unmixed"), we quantitatively demonstrate that as the structural similarity between targets increases, semantic unmixing becomes progressively harder for the network. This was further validated in μ Split using a synthetic dataset of simple sine-wave-generated structures.

Notably, Table 1(c) includes a successful four-channel semantic unmixing result, suggesting that in favorable cases, biologists could group up to four fluorescent structures into a single acquisition channel. By extending this approach across multiple multiplexed channels, it may be possible to image well over 10 distinct structures—even on a standard spinning-disk microscope in live-cell experiments. We are not aware of other methods in existence that could enable this.

b. The work here presented appears to require triple labeling with different fluorophores. The authors should clarify how this triple-labeling approach is advantageous compared to the approaches for machine learning-based fluorescence prediction referenced above (e.g. DOI: 10.7554/eLife.55502)

Virtual staining is a different task than the semantic unmixing. Here, we would like to raise few additional points in support of our approach. While we agree that virtual staining has indeed proven useful in several cases, we argue from a conceptual point of view that it has the following two issues not faced by our approach: (a) Incomplete information: Some of the structural details, particularly the high-frequency details in the fluorescence target structures often are absent in the brightfield input image. So, a virtual staining network needs to predict those missing details. In our setup, all the details present in the fluorescence target images are present in the input image. Hence, the network does not need to invent any structural details. It just needs to *decompose* the input into the predictions. (b) Structural Noise: Unwanted sample details present in the brightfield input image would act as noise, thereby complicating the prediction task. In our setup, input is, by construction the superposition of only those structures

which we wish to disentangle. We therefore argue that our approach can predict high frequency details and dim structures better than what a virtual staining task would do.

As emphasized earlier, virtual staining is fundamentally distinct from the unmixing task we present. To further support our approach, we highlight two key advantages over virtual staining:

1. *Complete Structural Information:* In virtual staining, high-frequency details of fluorescent structures are often missing in the brightfield input, forcing the network to predict absent features. In contrast, our input image inherently contains all structural details of the target fluorescence signals—the network only needs to decompose them, not “invent” them.
2. *Minimized Structural Noise:* Brightfield images include extraneous sample details that act as noise, complicating virtual staining. Our input, by design, consists only of the superposition of structures to be disentangled, eliminating this source of interference.

We therefore argue that our approach can predict high frequency details and even dim structures better than a virtual staining approach would do from a brightfield acquisition (since it is simply a much harder task).

c. The authors should demonstrate how this approach overcomes the limits of maximum sampling frequency in live cell imaging scenarios while reducing photo-bleaching. Detecting fluorescence emission while exciting with 3 lasers as reported by the authors (“we acquire an additional image channel by exciting all used labels at once and collecting the entirety of the emitted light”) is known to cause photo-physical effects that increase non-linearly the photobleaching speed (DOI: 10.1038/nmeth.3891, DOI: 10.1038/nmeth.3405).

Temporal multiplexing is conducted in virtually every imaging facility on a daily basis. In all such cases, each channel is acquired after one other sequentially and imaging k channels is about k times slower than imaging a single-channel. Hence, improving the temporal sampling frequency is trivially achieved by MicroSplit since it allows to acquire a single channel containing multiple structures that can later be unmixed.

Regarding photobleaching, we agree with the reviewer that using multiple excitation lasers will cause faster bleaching. However, since photobleaching depends upon the total laser power and exposure duration, the unsupervised denoising capability of MicroSplit helps to save some photon budget. In this work, we acquired the HTLIF24 datasets with varying levels of exposure duration, and acquired images with exposure duration being as low as 2ms. We did this to demonstrate that our method, even with noisy short exposure input images, provides useful denoised results (see Fig. S12, for results on 2ms acquisition duration).

Additionally, we also propose to adopt imaging protocols where the same fluorescent marker is used to label multiple structures. In this case, a single excitation laser will be needed and the photon exposure would be multi-fold reduced.

2. One of the claimed advantages of MicroSplit is that it enables more photon-efficient multiplex imaging, reducing phototoxicity and improving imaging speed.

a. Neither photon-efficiency nor reduced phototoxicity are demonstrated in this work. The authors should prove experimentally these claims.

We would like to draw the reviewer's attention to the fact that denoising isn't just about cleaner images-it's also a tool to rethink the required photon-budget. Similar to previous deep-learning based denoising setups, the denoising capability of MicroSplit opens up the possibility of using a smaller photon budget to get images having the SNR of images acquired with higher photon budget. To this end, we added the table ST11 in the supplement to demonstrate this. In the table we treat the 3 channel data (Nucleus, Microtubules and Kinetocore) of 500ms acquisition duration sub-dataset in HTLIF24 dataset as ground truth and estimate their similarity with sub-datasets of HTLIF24 having lower acquisition duration. The utility of this analysis is to enable comparing these metric values with those provided for Tasks XXIII-XXVI presented in Table 1. For example, for channel 1 (C1), our prediction on 2ms superimposed input (Task XXIII of Table 1) is better than the 20ms groundtruth (last row of the tableST11). In simple terms we compare two situations: (a) acquire individual channels with 20ms acquisition duration, and (b) acquire a single channel containing the superposition of three structures with 2ms as the acquisition duration and use our Microsplit network to unmix. By comparing the quantitative metrics, we find that the second approach is better.

b. The settings depicted in Figure 1A suggest an inefficient microscopy configuration, with surprisingly narrow emission bands, where a significant portion of the emission signal is lost. Using broader emission bands would collect a large number of fluorescent photons, increasing the efficiency of the system likely 2-fold and resolving, in a more practical approach, the motivation of this work. The authors should clarify how the choice of more efficient microscopy settings would affect the improvements of this approach.

We ask the reviewer to appreciate that Figure 1A is a cartoon optimized for visibility. The concrete imaging setup for all datasets was optimized by the microscopists who created the respective data and was by no means influenced by us. The only reason to image with less than ideal microscope configurations was when we wanted to test the limitations of our method.

c. Most samples used in this work are fixed samples labeled with Alexas dyes. The authors should clarify how phototoxicity has any effect on this type of samples.

Phototoxicity is not a concern on fixed samples. It is on live specimen though, where MicroSplit can help to reduce the photon exposure in the ways explained above and in the manuscript.

d. Regarding live cell imaging: the authors should demonstrate and experimentally prove the reduced phototoxicity in live samples, ideally in 3D considering the claim 2 of optimized analysis of 3d fluorescence imaging dataset.

We believe that the reduction in light exposure by either imaging multiple structures in one channel or by relying on MicroSplit's unsupervised denoising capability are obvious.

If the editors share the reviewer's opinion, we will of course conduct this experiment and accept the cost and delay this decision will cause.

3. The authors freely intermix the terms unmixing, semantic unmixing, computational multiplexing. This may be misleading to readers as the term “unmixing” is traditionally associated with spectral unmixing, where signals are mathematically decomposed into component intensities. MicroSplit does not perform spectral unmixing but instead predicts structures without preserving spectral intensity relationships. I would suggest the authors clarify the terms or intent of the approach. Perhaps a more appropriate term would be “semantic splitting” or “computational multiplexing.”

Now, we consistently use the term “semantic unmixing” and use computational multiplexing only when introducing the very concept of semantic unmixing. We hope this addresses the issue of the reviewer sufficiently well.

4. This work appears to use a total of 24 datasets.

9 datasets, 29 tasks which are derived from them. For the supplement, we have now added one more dataset and a new task to shed more light into one of the limitations of MicroSplit. See supplementary section B.5.

a. This number appears to be substantially small, in comparison to other works. For example, Christiansen et al uses ~200,000 images. The authors should comment on the risk of overfitting to specific training conditions, given the limited number of datasets.

MicroSplit is meant to be trained from scratch for every semantic splitting task, effectively eliminating the risk of overfitting. (If MicroSplit would overfit, the validation loss during training would clearly show this.)

To show how little data is sufficient to train a well working model, we conducted a series of experiments using the HTLIF24 dataset (20ms). We progressively reduced the amount of training data and consistently tested the trained network on the entire validation data. The plot below illustrates the resulting PSNR (Y-axis) as a function of the fraction of training data used (X-axis). The results show that performance plateaus well before utilizing the full training dataset of 90 images of size 1608x1608. However, we note that the amount of data required can vary across tasks, depending on the complexity and variability of the underlying structures.

Additionally, we believe that 24 total tasks to be acquired and shared with the world for this manuscript is a good number we are in fact rather proud of!

b. How do the multiple optical elements affect the network? (e.g. different lenses, different beamsplitters/filters)? How robust is the approach to different optics/instruments? The authors should validate the generalizability of MicroSplit.

As mentioned in our response to 4.a, we have assessed the performance of MicroSplit across multiple microscopy setups without problems. We are actually not entirely sure what the reviewer is suggesting us to do/test. MicroSplit will be trained on data from one optical setup and the trained model later applied to more data from the same setup. In what way will a lens of beamsplitter or other optical element interfere in any undesired way with this approach?

We will add (in Section A), a more detailed description on how data acquisition and training should be conducted to avoid any problems. Since data will be publicly provided, the meta-data will allow readers and future users to look up all acquisition details.

c. How do experimental settings affect MicroSplit? Different pixel sizes/zoom level, different type of detectors, different types of noise (which appear to be ignored in this work) contribute differently to fluorescence microscopy images. The authors should comment or prove effectiveness of the approach under varying experimental settings.

As before, parameters such as pixel size, zoom level, or detector type will change between training and prediction. Hence, none of these properties will make any difference. We have experimented with 9 different datasets, and with two datasets, we have explored multiple noise levels and different mixing ratios, and so we can say that MicroSplit has the ability to perform well across all these setups. That said, we want to reiterate that we do not intend to use the same trained network to be used across all possible configurations. (Please note that this is not a special assumption we take, but something we share with a plethora of other methods used on and with microscopy image data!)

The remark about different types of noise is interesting. MicroSplit’s unsupervised denoising is most effective for pixel noises (noises that are independent per pixel given the signal), such as Gaussian or Poisson noise.

5. Training modes and results:

a. The authors state in reference to training mode I “We generate mixed input images by pixel-wise summation of multiplexed image channels. These input images closely resemble what can later be acquired in a single acquisition” this statement may be incorrect depending on the type of fluorophores, excitation wavelength and emission filters used, owing to the different excitation and emission spectra of fluorophores.

We are not sure we grasp the full extent of the reviewer’s concern. We do agree, the factors mentioned above must be chosen such that training data created by summing multiplexed channels and the data the trained model is later applied do match (note that this test-data can either be acquired with the same fluorescent labels and multiple excitation lasers and filters being applied, or after labeling those structures with the same fluorophore and imaging the superposition truly as a single fluorescent channel). We don’t see how this depends on different excitation and emission spectra of the used fluorophores as the reviewer seems to conclude. As long as the labeled structures in the training and test data end up being roughly in the same relative intensity range (and even this “requirement” can easily be softened by using intensity augmentations during training), MicroSplit can be used without any restrictions. (We believe the reviewer might be misguided by considerations that hold true in the context of spectral unmixing, as we have also pointed out in other locations in this rebuttal.

The puncta removal task is maybe the most “extreme” example where we used Training Mode I. We still see good performance when the network trained using Training Mode I was used on input images acquired in a single channel. Moreover, in the supplementary section C.3, we have quantified the performance degradation when the network gets trained using Training Mode I. We argue that while there is a degradation in performance, the results still look reasonably good to perform downstream tasks.

b. In one of the notes the authors state the “ground truth” used is the “target images” or images acquired by the instrument. These images are intrinsically still spectrally overlapping, hence suggesting the reference image used by the authors is already “spectrally mixed”. The authors should clarify if this was the intent, as these images are not “unmixed”.

We suspect the reviewer might have misunderstood something. For most of the datasets we worked with, one cannot easily see any bleedthrough. Hence, there is in fact no significant “spectral overlap”. The “target images” we talk about that we use as ground truth are the image channels we acquired via temporal multiplexing, including suitable high-, low-, or band-pass filtering so that these channels do not contain overlapping spectra.

For some few tasks, one can indeed see some bleedthrough. However, we emphasize that it does not say anything about MicroSplit. If the target structures will have bleedthrough, the network will be trained to predict structures with bleedthrough.

c. Training mode III seems to have a more realistic acquisition using 3 illuminations and collecting one single channel. These more realistic samples yield the lowest performance of the method presented (Table 1). The authors should acquire more realistic data (e.g. using same fluorophore for 3 targets) and in larger quantity to prove more convincingly the effectiveness of the approach.

This is a misconception that we would like to clarify. The performance is dependent on the complexity of the structures to be unmixed, the similarity of the structures to be unmixed, and the SNR of the data. For example, one can observe task XXVII to have a much higher PSNR than task XXIII. Both these tasks use Training Mode III. Between these two tasks, the only difference is acquisition duration. While task XXIII is using lower-SNR data, task XXVII is conducted on a higher-SNR version of the same data.

d. Most of the datasets used in this work utilize Alexa dyes, which are bright and generally difficult to photobleach when imaged with efficient experimental settings. The authors should clarify this choice as opposed to a broad variety of commonly used genetically encoded fluorophores. Based on supplementary material it seems (though it is not clearly described) that Pavia-P23 dataset uses eGFP. Assuming Pavia-P24 is using the same samples described in Pavia-P23, the results reported in Table 1 for this sample substantially score lower than the Alexa samples. The authors should clarify the performance of the system against less-bright fluorophores or state more clearly the limitation of this approach to bright samples.

MicroSplit's operations do not change with the brightness or photostability of fluorophores, as long as the structures are visible in the training and test data. The final performance does naturally depend on various factors, and SNR is one of them (see also above). The primary reason why SNR and not absolute brightness dictates the quality of results is that MicroSplit operates on normalized input images, meaning that we feed inputs where absolute intensity information is normalized out (note that this is a common thing to do and that denormalization of network predictions are bringing intensities back into the range of the original data).

With Pavia-P24, the issue lies in the fact that target and the input images show soft and “hazy” structures that are not crisp (object outlines visually not “well defined”). As an example, inspect Figure S28 and appreciate that both target channels are “hazy”. When target images are “crisp”, we observe that semantic unmixing tends to lead to better results, even when the network sees only a smaller region of the input. However, when high-frequency details cannot differentiate the two targets, the task at hand becomes more difficult since more spatial context needs to be integrated. If this observation and argument is helpful, we are happy to integrate it, for example, in the discussion section.

6. Segmentation results (Figure 3):

a. Figures 3B and 3C report high DICE scores, but visual overlays show significant discrepancies between predictions and ground truth. Particularly, 3B shows substantial amount of under-prediction or over prediction for A3 and A2 respectively.

We have double checked the DICE formulation, and confirm that it is correct. If x & y are two boolean arrays, we compute dice coefficient as $2 \cdot \text{SUM}(x * y) / (\text{SUM}(X) + \text{SUM}(Y))$. Note that while DICE score is computed over multiple full frames, what is shown is a smaller crop from a single frame. We have now shared full frames, their predictions, and the corresponding code for DICE computation (https://github.com/CAREamics/MicroSplit-reproducibility/tree/rebuttals/examples/segmentation_results), so readers can convince themselves and browse the data and our results in detail by themselves.

b. The reported DICE values do not seem to match the visible alignment between images. It is unclear whether the DICE scores are computed over thresholded regions (positive binary) or across the entire image. The authors should clarify how these values were computed and justify their accuracy.

Thanks for the feedback and suggestions. The requested information is given as part of the previous answer.

7. Improving photon budget: the authors claim “imaging two structures in a single channel can free up photon budget, which then becomes available for imaging at a higher frame rate, imaging at a higher signal-to-noise ratio, or imaging more labeled structures in additional image channels”.

a. What assumption is used for this statement? The imaged channel has a broader spectral band? If labels are different, due to the stochastic emission of fluorescence, shaped as the spectrum of the fluorophore, collecting one channel or collecting multiple channels (if done efficiently) will not change the total number of photons emitted by the fluorophores or detected by the detectors. The authors should clarify and ideally demonstrate this statement.

At least three arguments support our claim that MicroSplit can help to free up photon budget.

(1) In a typical multi-color setup, one must use an emission filter that rejects all wavelength not exclusively (or at least predominantly) used by the fluorescent label currently imaged. However, if a single channel is to be used, then this constraint can be relaxed and one can use a broader emission filter (or even avoid any filtering if all structures are labeled with the same fluorophore). Hence, with the same laser power and exposure duration, one would collect more photons (broader filter).

(2) With a bit of mathematical reasoning, one can see that aggregation of the photons emitted by multiple structures, the overall SNR will improve. Still, even if we assume that the same number of photons as previously collected into two separate channels are now collectively

imaged in one, the SNR of this single channel acquisition is higher. If one chooses to work with superimposed images that are having an SNR similar to individual acquisitions, one has now the freedom to reduce the exposure duration and/or laser power appropriately. Next, we provide the mathematical argument supporting our claim.

Gaussian and Poisson are the two dominant noise sources in microscopy. If X and Y follow the Poisson distribution with parameters a and b , then mean/stdev of $X+Y$ is $(a+b)/\sqrt{a+b} = \sqrt{a+b}$. The average SNR for X and Y are $(\sqrt{a} + \sqrt{b})/2$, which is strictly less than $\sqrt{a+b}$. Note that both $\sqrt{a} < \sqrt{a+b}$ and $\sqrt{b} < \sqrt{a+b}$. So, the above inequality holds. If X and Y follow the Gaussian distribution with parameters (μ_1, σ) and (μ_2, σ) , the SNR of $X+Y$ would be $(\mu_1 + \mu_2)/(\sqrt{2} \cdot \sigma) > (\mu_1/\sigma + \mu_2/\sigma)/2$.

We can see this effect visually as well. In Figure 2.b, for example, compare the noise levels of the target images with the noise level of the input image. The input appears less noisy than the target. (Note that the laser power and acquisition duration was left unchanged when imaging the target and the input.)

(3) Our method, next to splitting structures, also conducts unsupervised denoising. While denoising does not improve the SNR at acquisition time, it certainly does so post-hoc (and thereby makes downstream analysis easier). This allows users to acquire data at lower SNR, and hence, saving photon budget.

To better illustrate this, we have now conducted an analysis which set out to explore MicroSplit's photon efficiency (via denoising). In the table we show below, we use the 3 channel data (Nucleus, Microtubules, Kinetocore) of the 500ms exposure subset of HT-LIF24 data as ground truth. We then took the shorter exposure times data subsets of the same data and computed their similarity to this "ground truth". We do this to contrast the numbers in the table below with the corresponding values for Tasks XXIII-XXVI in Table 1 (main table in main text). For example, for channel 1 (C1), in terms of PSNR, our prediction on 2ms superimposed input (see Task XXIII in Table 1) is better than the C1 obtained directly from the microscope with a 20ms acquisition duration (last row of the table here below). Similar to previous deep-learning based denoising setups like CARE [3], the denoising capability of MicroSplit opens up the possibility of using the available photon budget more effectively.

Acq. Duration	PSNR			MicroMS-SSIM		
	C1	C2	C3	C1	C2	C3
2ms	23.3	25.1	26.1	0.839	0.772	0.869
3ms	23.6	26.0	27.2	0.842	0.780	0.871
5ms	24.6	28.3	29.9	0.857	0.817	0.875
20ms	30.0	35.6	38.16	0.920	0.942	0.914

8. Removing unwanted imaging artifacts: authors state that “any subsequent analysis can be performed with the images predicted by MicroSplit”. The results shown in figure 2 suggest a creation of additional signal in the images as well as a blurring artifact in the prediction. This is echoed by the results in figure 3 and 4 where intensity distributions are seemingly altered in a gaussian blur, altering any intensity-based analysis the predicted images. The data presented by the authors contradicts this statement from the authors. The authors should correct the statement to align with their results.

We would first like to address the claim that our predictions are subject to “Gaussian blur”. This is not the case. Instead, the absence of high-frequency pixel noise makes denoised images appear blurred. We believe that our results show images that are just as blurry as the resolution of the microscope permits. (Note that all MMSE images we show are slightly more blurry since those images also average over the differences between posterior samples, meaning that the data uncertainty is additionally translated into “blurriness”.)

Our sense of downstream applications include segmentation, tracking, detecting presence/absence of some structures, counting the structures, estimating the dimensions of structures (like radius, length, etc.). In all these cases and more, our predictions can be used. However, we would be cautious about using MicroSplit for downstream tasks which are sensitive to minor pixel intensity differences. For example, we would not recommend using MicroSplit to estimate the width of very thin structures, especially in cases where the raw input data is very noisy. Intensity quantifications can, potentially, be conducted on the original raw data. If this is not the case, this kind of downstream processing is indeed not possible. We have clarified this in Section 2.5.

9. In the limitations section the authors suggest that “a higher degree of skew (unequal channel intensities) makes it difficult for MicroSplit to pick up the weaker of the two signals. For users to balance the intensity emissions, the laser power for each fluorophore (in most of the experimental settings described by the authors in supplementary) need to be adjusted. By looking at each emission channel independently. This contradicts the motivation described at the beginning of the manuscript, of collecting a single channel. The authors should clarify this statement.

As the reviewer correctly points out, we quantitatively demonstrate that a higher degree of skew causes MicroSplit predictions to be of lesser quality. We pointed this out as a limitations of our approach. We do, however, not see how this leads to any contradictions. It simply means that in cases where the various structures that are imaged in one go are of vastly different intensities our method should be used with caution because it might fail to correctly predict the dim structure(s).

10. In limitation section the authors state “although we observe blurriness in predictions for the weaker channel in a skewed HT-P23A derived task” this is true also for the results in figure 1, 2, 3, 4, S4, S10, S11, S12, S13, S14, S15, S16, S18, S19, S20. The authors should update their

statement accordingly or explain the visible blurriness that appears in majority of the predicted images.

Please refer to our response to question 8. We believe that the results produced by MicroSplit does not suffer from blurriness to the extent that the utility of the prediction is reduced. PSNR and SSIM values of Table 1 supports our claim quantitatively.

With this being said, even higher PSNR predictions (less blurry) can be achieved by slightly modifying the loss used in Microsplit. Not that this will reduce the trained network's ability to conduct sampling, calibration, and in some cases even the ability to fully denoise. Our default configuration is a general purpose setup which yields useful predictions in all different cases and can be used as out-of-the-box configuration for training MicroSplit. However, if one desires the highest possible PSNR at the expense of sampling, calibration, and denoising, the following small modification in the notebook (01_train.ipynb) will enable that: the uSplit loss needs to be strengthened, in the most extreme case by setting $w=1.0$. Note that the default value used in MicroSplit training is $w=0.9$.

```
1 experiment_params["data_stats"] = data_stats
2
3 # setting up training losses and model config (using default parameters)
4 loss_config = get_loss_config(**experiment_params)
5
6 # set the balance between the musplit and denoisplit loss components.
7 w = 0.99
8 loss_config.musplit_weight = w
9 loss_config.denoisplit_weight = 1 - w
✓ 0.0s
```

11. In limitations, in reference to changing scale of images (Figures S2, S3) authors state that “MicroSplit can successfully unmix the structures even when the smallest nonzero scaling factor is applied”. The results in S2 Magnification factor 1 Prediction Ch1, Ch2 show very evident artifacts. Magnification factor 1.125, Prediction Ch2, cells on the right and bottom side show evident clipping artifacts, similarly in Mag factor 1.25. The authors should correct this statement or clarify the differences.

Semantic unmixing is an ill posed problem from a mathematical point of view. We do not claim otherwise. So, when the problem becomes harder, in this case by reducing the magnification factor closer to 1, the prediction quality deteriorates. However, we believe that the predictions made for even magnification factor 1.125 can be useful for downstream applications like counting, segmentation and tracking. We have now added two more entries for even lower magnification factors (1.0625, 1.03125).

12. The authors propose using a self-referential calibration approach (Figure 2) where the same sample is used to evaluate its own error.

a. The authors should better discuss this approach with respect to tautological validation.

We are afraid we cannot follow how this applies to Figure 2.

What we use to estimate the uncertainty of predictions is the variability between posterior samples of our variational network. The core idea is that a posterior model like ours will generate posterior samples that vary roughly “in line” with how uncertain the network is about its own predictions. (At least in cases where the network is calibrated, which we show empirically for all models we trained.)

b. Would this estimation of calibration only reflect how confident the model is in itself, rather than how accurate it actually is? The results of this calibration appear in contrast with the empirical results shown in most of the figures, reporting a frequent alteration of intensities in the predicted channels respect to the target channels. This change in intensities would considerably alter results of any intensity-based analysis, affecting its reliability.

The calibration methodology employed in this work involves a linear transformation of pixelwise uncertainty estimates derived from multiple posterior samples generated by the network. This transformation aims to align the adjusted pixelwise uncertainty as closely as possible with the actual error. A notable advantage of this approach is that the quality of calibration can be assessed a priori by examining the calibration plot. For instance, in cases such as Figure 2b (last column), where the root mean squared error (RMSE) versus root mean variance (RMV) closely follows the $y = x$ line, the uncertainty estimates are likely well-calibrated with respect to the true error. Conversely, significant deviations from the $y = x$ line serve as a clear indication that the model’s uncertainty estimates should not be relied upon.

It is important to note that uncertainty quantification remains an open challenge, with multiple competing methodologies often presenting contradictory perspectives. For further discussion on this topic, the reviewer may refer to [1] (reference provided at the end). While the proposed approach may not be universally applicable, its utility lies in the ability to determine—through simple inspection of the calibration plot—whether the uncertainty estimates can be trusted in a given scenario.

13. Quantification methods used by the authors seem to not match visual assessment. Example, the maps shown in Figure 3 “overlays” suggest a much lower visual performance than the DICE score reported. This may be due to the averaging over the entire image (mostly dark), instead of the regions with information content. Example in task 1, Analyst 3 overlay shows a substantial under-prediction, while the DICE score shows over .75. The authors could show more maps of estimated error and correlate those better with the single-value results (e.g. MicroSSIM or DICE) to better prove the method and increase the robustness of the presentation.

As mentioned in our response to the the Question 6a, we have double checked our numbers and they are correct. Still, we have now made raw data, predictions, and segmentation outputs public together with an analysis notebook for everyone to work and play with. The notebook is

hosted at https://github.com/CAREamics/MicroSplit-reproducibility/tree/rebuttals/examples/segmentation_results.

14. The differences between the combination of μ Split with denoiSplit and the current method are not well described. The authors should better describe MicroSplit with respect to μ Split and denoiSplit to position it as a new method rather than an incremental improvement.

In Section “A Model Architecture and Training” we have gone into the technical details of how MicroSplit has technical advancements over μ Split and denoiSplit. We briefly enumerate them here. (a) μ Split was GPU-efficient and tailored for high-SNR datasets. denoiSplit offered denoising and uncertainty estimation. MicroSplit is GPU-efficient and at the same time performs denoising and uncertainty estimation. To enable this, we needed to modify the KL-divergence loss formulation of denoiSplit. (b) μ Split and denoiSplit worked with 2D images but with MicroSplit, we can now work with 3D volumes. It not only speeds up the inference, it also yields superior performance (>3.5db PSNR). (c) The loss formulation used in MicroSplit enables a way to control between preserving high-frequency details and getting better denoising and uncertainty estimation with just one hyper-parameter. See section A.1 for more details. Additionally we have added an efficient 3D implementation and have applied MicroSplit to 24 total tasks, describe an artifact removal task for the very first time, and showcase downstream segmentation advantages. Last but not least: in this manuscript we are clearly targeting biologists and microscopists and not, as before, ML experts. For this reason we have also gone through extensive efforts to make the code-base clean and usable (e.g. via the reproducibility notebooks).

15. Presentation and readability of the manuscript

a. Text clarity and accessibility should be considerably improved. The manuscript seems to be written for a highly technical journal, more in line with ML style, requiring readers to refer to multiple external sources to fully grasp the methodology and prior work, necessitating significant background reading, which may limit accessibility for a broader audience. The authors may consider refining the presentation to improve clarity and readability, particularly if aiming for a journal like Nature Methods, which caters to a diverse readership spanning different scientific disciplines.

We thank the reviewer for the comment. While we consciously avoided technical details in the main manuscript and relegated them to the supplement, we have gone through both the main manuscript and the supplement and have adjusted the content in-line with all explicit feedback from reviewers. We remain open to further suggestions for how to improve the main text and supplement, either by the reviewers or the editorial team.

b. Regarding data presentation and organization: the way results are reported deviates from common conventions for this type of journal, making interpretation more challenging. For example, Table 1 presents multiple tasks and performance metrics in a highly condensed

format, requiring cross-referencing with the additional supplemental material to fully understand the comparisons and tasks performed. Error measurements for this same table are reported in a separate table (Supplementary Table ST9), making it difficult to quickly locate and interpret key findings. A more structured and intuitive layout (or plot) could enhance the readability and accessibility of the data.

Also here we are open for concrete and actionable feedback from the editorial team and all reviewers.

c. Dataset information: the description of the datasets used in the study could be more clearly structured. While the manuscript states that datasets are shared, which is great, a comprehensive summary table detailing dataset characteristics, such as image size, pixel resolution, microscopy settings, bit depth, and time points, would improve transparency. Some of this information is included for certain datasets, but for others, it is either difficult to find or not explicitly stated. Providing a standardized dataset summary would make it easier for readers to assess the scope and applicability of the study.

We thank the reviewer for this suggestion. We have added the table ST8 in supplement. Among other things, the table also contains the column Pixel count, that represents the total number of pixels (across training, validation, and test sets) for each task with 1 million as the unit. Since a 1000×1000 sized image corresponds to 1M pixels, so, Pixel Count column serves as a ballpark estimate of the number of images needed for training MicroSplit.

Reviewer #1 (Remarks on code availability):

The MicroSplit code is available and includes installation instructions, but it lacks explicit examples on how to apply the method to real data. While the provided documentation describes how to set up the environment, there are no practical examples demonstrating how to run the model on fluorescence microscopy images. This structure caters more toward a computer science-oriented audience, making it less accessible to researchers without a good programming background. For those familiar with machine learning workflows, the code is understandable, but a step-by-step guide with example datasets would greatly improve and facilitate usability and reproducibility. Enhancing the documentation with practical use cases would make the resource more accessible to a broader community. A simple UI would also facilitate adoption.

We have already worked on a more comprehensive set of notebooks and example code when we submitted this manuscript the first time. After reading this feedback we decided to delay the re-submission until we finished this work (originally we hoped this could be done in parallel to reviewing, revising, and other steps towards publication).

More example notebooks and, most importantly, generic “use your own data” examples in 2D and 3D can now be found here:

<https://github.com/CAREamics/MicroSplit-reproducibility/tree/rebuttals/examples>

Reviewer #2 (Remarks to the Author):

Summary of the Key Results:

The authors present MicroSplit, a deep-learning-based method for computational multiplexing in fluorescence microscopy. This technique enables multiple cellular structures to be imaged within a single fluorescent channel and subsequently unmixed computationally, reducing photon exposure while maintaining imaging quality. MicroSplit employs a variational splitting encoder-decoder (VSE) network, leveraging posterior sampling to estimate prediction uncertainty. The manuscript demonstrates that MicroSplit can effectively separate up to four superimposed structures, denoise images, and provide uncertainty quantification at a quality level suitable for downstream tasks.

Originality and Significance:

While the concept of computational multiplexing is not new, with prior work such as PICASSO addressing similar problems, MicroSplit distinguishes itself by requiring only a single superimposed image as input. This eliminates the need for spectral overlap measurements, which presents a significant advantage. The supplemental material compares MicroSplit to PICASSO effectively, though moving this comparison into the main text would strengthen the manuscript. Overall, the method has the potential for broad adoption and significant impact in fluorescence imaging.

Data & Methodology:

The channel segmentation methodology is well-reasoned, but the Error Estimation, Uncertainty, and Calibration section requires substantial revision and clarification. The authors state: "We therefore estimated the true error by computing the pixel error between the MMSE prediction of MicroSplit and the ground truth target images we derived from the available training data, and plotted this 'true error' against the RMSE errors we described above."

We apologize for the mistake in the above statement. The correct statement would have been the following: "We therefore generate the true error (RMSE) by computing the pixel wise error between the MMSE prediction of MicroSplit and the ground truth target images we derived from the available training data, and plotted this 'true error' against the scaled RMV (estimated RMSE errors) we described above."

However, the calibration curve axes are labeled RMV vs. RMSE, yet RMV is not explicitly defined in the manuscript, making interpretation difficult. Additionally, the claim that RMSE can be calculated solely from posterior samples suggests that the authors are computing RMSE between the posterior samples and the posterior MMSE image. Since RMSE can be expressed as the square root of the sum of variance and squared bias, and given that the MMSE image is an unbiased estimator, pixel-wise RMSE should correspond to the standard deviation of the posterior. RMV represents a similar metric and would be expected to exhibit a nearly linear correlation with RMSE.

We adopted the uncertainty and calibration methodology from denoiSplit. Here, we would like to clarify the terms RMV and RMSE. We added supplementary section D that explains it in sufficient detail. Pixelwise squared error is computed between the groundtruth image pixels and MMSE image pixels. Next, we compute the pixelwise variance in the prediction using different posterior samples. We can do it because, for one pixel location, each posterior sample has possibly a different value. We then aggregate these pixelwise variances into 30 bins to yield 30 RMV (root mean variance) values, one value per bin. For these bins we aggregate the squared error to get RMSE (root mean squared error) values. Using two scalars per channel, one as the multiplicative factor and other as the offset, we scale the RMV values to optimally match RMSE.

If the goal is to demonstrate that low posterior variance correlates with low error and can serve as a proxy for uncertainty, I would recommend the following modifications:

1. Increase the number of posterior samples beyond 50 to allow for a more robust estimation of pixel-wise variance.
2. Compute the pixel-wise ‘true error’ using the posterior MMSE image and plot it against the posterior variance, providing a quantitative correlation metric.
3. In Figure 2, replace the Sample 1 - Sample 2 heat map with a posterior variance heat map, and replace the RMSE column with the true error heat map, enabling readers to visually assess the relationship between variance and error.
4. Explicitly provide equations for all calculations related to calibration and uncertainty estimation.

Partly due to the unclear explanation given in the main manuscript, there was some misconception about how our uncertainty and calibration module works, which apparently led to these suggestions. With adequate explanation, we believe that the reviewer would be satisfied with what we have done. We have covered suggestion 2. and 4 in our previous answer. We picked the number 50 for posterior samples because we did not find a significant change beyond 50.

Validity of Approach, Quality of Data, and Presentation:

The methodology is generally well-structured, with the exception of the Error Estimation, Uncertainty, and Calibration section. Several figures require revisions for clarity:

- Figure 1.A: The axis labels are difficult to read, and a legend is necessary. Additionally, the figure caption refers to emission filters, whereas the main text describes them as excitation filters—this discrepancy should be resolved.
- Figure 2.A: The second lateral context patch should be explicitly labeled for clarity.
- Figure 2.B: The term RMV is undefined in both the calibration plots and the main text, making interpretation challenging. See further comments above
- Supplemental Figure 5: The x-axis should have consistent spacing, either in linear or logarithmic scale, or alternatively, the figure could be reformatted as a bar graph for improved readability.

We thank the reviewer for these concrete suggestions. Figure 1.A is a schematic representation, and not a qualitative figure (additionally, we do not see any mention of excitation filters). Hence the axis labels and the legend are not needed and we will remove them to give the figure a cartoonish flavor. We have made all the other suggested changes (Updated Figure 2A, updated caption for Figure 2B and updated supplemental Figure 5). For RMV definition, we provided an intuitive definition in the caption and have added section D in the supplement that explains the uncertainty quantification and calibration methodology in detail.

Regarding data quality, the authors have taken care to collect and describe three distinct datasets. However, the manuscript's clarity could be improved by more precisely distinguishing between traditional sequential multiplexing (multichannel acquisition) and the proposed simultaneous multiplexing (multiple structures within a single channel). Introducing explicit terminology for these concepts would reduce ambiguity.

References:

The manuscript cites an adequate body of prior work, and I did not identify any major missing references. However, the claim that 32.53 PSNR and 0.886 MicroMS-SSIM are sufficient for downstream analysis would benefit from additional supporting literature.

The validity of this claim is supported by prior works in supervised and unsupervised denoising [2,3,4], which report PSNR and SSIM values within a comparable range. However, it is important to note a caveat regarding PSNR: its magnitude is influenced by the intensity range of the ground-truth data. Specifically, ground-truth images with higher pixel values generally yield higher PSNR values. Nevertheless, the referenced denoising methods remain relevant for comparison, as they were also evaluated on microscopy data and achieved similarly competitive PSNR results.

Final Recommendation:

The manuscript presents a promising computational multiplexing method with significant potential impact. However, revisions are necessary, particularly in the Error Estimation, Uncertainty, and Calibration section, as well as in figure clarity and definitions. I recommend major revisions to improve methodological transparency, ensure rigorous uncertainty quantification, and enhance clarity in data presentation. With these improvements, MicroSplit has the potential to contribute meaningfully to the field of fluorescence microscopy.

- Eric Markley

Reviewer #3 (Remarks to the Author):

MicroSplit combines the advantage of uSplit and denoiSplit, also with extension to volumetric images, to tackle the semantic unmixing problem for fluorescence microscopy. Overall, this work is both theoretically novel and practically useful, supported not only by experiments on extensive and diverse datasets / applications, but also by comprehensive discussion regarding the performance, caveats, practical considerations, etc. I would definitely endorse for publication, but with some minor suggestions.

A few strengths I really want to highlight:

- ***Supper flexible training settings (different modes in Section 2.1).***
I think this is really useful in practice, since different applications / use cases may have different data availability and the proposed framework can still be applicable in a lot of different situations. Importantly, comprehensive discussion and caveats are provided for better understanding of different training modes. This will serve as a useful guide for users to design their assay accordingly for their needs.
- ***The variational model as the backend supports informative error estimation.***
I really appreciate that the variational model backend, which fundamentally supports error estimations. The calibration step confirms the effectiveness. The error estimation and calibration make this method, not just something looking cool, but equipped with a handle for users to take full control of the trustworthy, which is important for deep learning based image predictions.
- ***Evaluation beyond PSNR and SSIM.***
By showing MicroMS-SSIM, the evaluation already provides a comprehensive view from an image perspective. But, it is still not enough to represent its impact on downstream analysis or its impact for answering the underlying biological questions. I really appreciate the additional experiments in Section 2.4, which provides application-appropriate validation to make sure the prediction is biologically valid for the specific application.

Besides the highlight above, I can confirm, to the best of my knowledge, the described methods are sound, the quality of presentation is good, easy to understand, and the use of statistics / evaluation metrics are appropriate. There are only some minor suggestions:

(1) Section A in Supp. can be significantly improved.

In general, the method proposed in this work combines previous work in uSplit and denoiSplit into a single setup. But, it is not quite clear how exactly this “final product” looks. In theory, I could check the previous publications for details. But, this may not be straightforward, since this “final product” is not a simple copy-and-paste from previous works. Therefore, I would highly recommend to provide more details in the Supp. so that, in theory, people should be able to reproduce the exact method based on the description in this paper. Here are some examples of what are not quite clear to me in the technical details.

We thank the reviewer for the suggestion. To this end, we have added content in Supplementary Section A.

A. In Supp. page 1, “In Figure S1, we present the architecture of MicroSplit, which has, LC inputs and Noise models, all integrated into a single setup.” But, when you look at Figure S1, it is not clear at all, where the Noise model is and how it is integrated exactly.

In Figure S1, we have focused on the Encoder-decoder aspect of our setup which was depicted in Figure 2a. The noise models are used in the loss computation and therefore do not form part of the encoder-decoder architecture. We have rephrased the caption of S1 so that it is clear.

B. In Figure S1, what is the “Input Branch”? Any computation within the “Input Branch”? I assume it is just some kind of connection in the neural network, correct?

The reviewer has assessed correctly. It is essentially a series of neural network layers. We will mention this in the Supplement A.

C. Also in Figure S1, the white squares in the network figure are getting smaller through layers, is it the result of pooling or unpadded convolution? If it is pooling, then the size is not halved.

As the reviewer predicts, it is indeed the case that the white squares get smaller due to pooling. White squares represent the content coming from the “primary input patch”. Similar to what happens in a U-Net, the spatial dimensions reduce by a factor of 2 at each consecutive hierarchy level due to a pooling operation. However, in MicroSplit, the resultant embedding is zero-padded after pooling and is subsequently concatenated with the embedding coming from one of the lateral contextualization (LC) inputs. Since the embedding coming from the LC input does not undergo pooling, the final embedding has the same spatial dimensions (gray squares) at each hierarchy level. We have explain this in more detail in the Supplementary Section A.

D. Again in Figure S1, the first part of all three types of models are three white squares. Are these three consecutive convolution layers? Or something else?

E. In Figure S1, there are center crops in Lean-LC. Is there anything similar in Regular-LC and Deep-LC?

No, there is no center-cropping applied to Regular-LC and Deep-LC on the output of the encoder. Their decoder takes in the entire uncropped embedding which contains the full information from the primary input and all the LC inputs. This lack of center-cropping is exactly what differentiates Lean-LC with Regular-LC and Deep-LC. However, as the reviewer may know, in the U-Net, while the encoder has pooling layers which do downsampling, the decoder has layers to perform upsampling. This combination of pooling and upsampling results in the final output having the same spatial dimensions as the original input. In the setup for Deep-LC and Regular-LC, since the size of the embeddings coming out from the encoder does not reduce, we centercropped the resultant embedding after each upsampling in the decoder.

I know a lot of these details can be discovered from the released code (great! Open-source is important!), but it is still good to explain the technical details in the paper as a top-tier publication.

Note: the missing details mentioned above did not affect my understanding of the key idea of the method.

(2) Is there a discussion on why “up to four structures”? I am not sure if I miss this somewhere in the paper. But, in general, what is the biggest trouble if we make an attempt on five or six structures? If we use Deep-LC or even a Deeper-LC, will five or six structures become possible?

While there is no theoretical limitation to our approach to be extended beyond four structures, there are both practical limitations and the limitations from the learning side. With most off-the-shelf microscopy solutions it is relatively easy to image 1-4 fluorescent channels. Beyond that, filter sets and enabling hardware are not commonly available (mainly for live cell imaging where multiple rounds of imaging are not an option) and even the imaged sample must be permissive to the required labeling and light exposure. Hence, Training Mode I and III are not directly permitting to train a network to split more than 4 structures. Beyond this practical limitation, as we increase the number of structures by other means, at some point the quality of predictions will likely drop (see Supplementary Section B.3), and for this work we chose to opt for applicability above pushing the limitations of the presented method. Additionally, even with the presented four channel splitting, the theoretical limitation on off-the-shelf microscopes lies at four parallel 4-channel semantic unmixing tasks, hence at 16 biological structures — undoubtedly a huge advantage for suitable projects where such a strategy would apply.

(3) Will the model be released on Bioimage Model Zoo? I don't think this will affect the evaluation of this paper. This is more of my own wish from a user's perspective.

Yes, we will upload the models to the Bioimage Model Zoo.

(4) There are some other really minor things in terms of presentation. You could even ignore this part.

- In Table 1, it would be better to use 1, 2, 3, ... or T1, T2, T3, ... as indices for tasks, rather than I, II, III.
- Would it be possible to provide a vision overview of all different training modes? I can appreciate the high flexibility of the propose method. A summary table or some kind of visual abstract / overview would be really helpful.
- Personally, I would really appreciate a few sentences (maybe in the Supp.) describing how the variational models allow error estimation using plain English. I believe these few sentences would be super helpful for the users without deep knowledge of machine learning to understand where this error comes from.

We thank the reviewer for the above mentioned suggestions. We added a supplementary Table ST12 giving an overview of the different training modes, which we also refer to from the main manuscript. We also added a new Supplementary Section D, where we describe in detail how

error estimation is performed. We have added an introductory paragraph which explains the overall idea without getting technical.

Reviewer #3 (Remarks on code availability):

I didn't run the code, but I read the code on Github, very clearly written scripts and software. Currently, the code is organized in a way focusing on reproducing the results in this paper. This is fair. But, from a user's perspective, I would like to see a clear instruction on what to do if I want to run training mode 1, training mode 2, etc., with my own data, and how should I organize my own data (2D or 3D). I have no doubt the documentation will get better.

We have already worked on a more comprehensive set of notebooks and example code when we submitted this manuscript the first time. After reading this feedback we decided to delay the re-submission until we finished this work (originally we hoped this could be done in parallel to reviewing, revising, and other steps towards publication).

More example notebooks and, most importantly, generic “use your own data” examples in 2D and 3D can now be found here: <https://github.com/CAREamics/MicroSplit-reproducibility/tree/rebuttals/examples>.

References:

[1] Reexamining the Aleatoric and Epistemic Uncertainty Dichotomy | ICLR Blogposts 2025. <https://iclr-blogposts.github.io/2025/blog/reexamining-the-aleatoric-and-epistemic-uncertainty-dichotomy/>. Accessed 4 June 2025.

[2] Weigert, Martin, et al. “Content-Aware Image Restoration: Pushing the Limits of Fluorescence Microscopy.” *Nature Methods*, vol. 15, no. 12, Dec. 2018, pp. 1090–97. www.nature.com, <https://doi.org/10.1038/s41592-018-0216-7>.

[3] Prakash, Mangal, et al. Interpretable Unsupervised Diversity Denoising and Artefact Removal, ICLR 2022. [arXiv.org](https://arxiv.org), <https://openreview.net/forum?id=DfMqIB0PXjM>.